# Stability and Generalization of the Decentralized Stochastic Gradient Descent Ascent Algorithm

**Miaoxi Zhu**[1]   **Li Shen**[2*]   **Bo Du**[1*]   **Dacheng Tao**[3]

[1] School of Computer Science, National Engineering Research Center for Multimedia Software,
Institute of Artificial Intelligence and Hubei Key Laboratory of Multimedia
and Network Communication Engineering, Wuhan University, China
[2] JD Explore Academy, China    [3] The University of Sydney, Australia
{zhumx,dubo}@whu.edu.cn, {mathshenli,dacheng.tao}@gmail.com

## Abstract

The growing size of available data has attracted increasing interest in solving minimax problems in a decentralized manner for various machine learning tasks. Previous theoretical research has primarily focused on the convergence rate and communication complexity of decentralized minimax algorithms, with little attention given to their generalization. In this paper, we investigate the primal-dual generalization bound of the decentralized stochastic gradient descent ascent (D-SGDA) algorithm using the approach of algorithmic stability under both convex-concave and nonconvex-nonconcave settings. Our theory refines the algorithmic stability in a decentralized manner and demonstrates that the decentralized structure does not destroy the stability and generalization of D-SGDA, implying that it can generalize as well as the vanilla SGDA in certain situations. Our results analyze the impact of different topologies on the generalization bound of the D-SGDA algorithm beyond trivial factors such as sample sizes, learning rates, and iterations. We also evaluate the optimization error and balance it with the generalization gap to obtain the optimal population risk of D-SGDA in the convex-concave setting. Additionally, we perform several numerical experiments which validate our theoretical findings.

## 1 Introduction

Minimax problems have shown extensive applications in machine learning, such as adversarial robustness [26, 16], GAN [11], the zero-sum game [27], multi-agent reinforcement learning [33], AUC maximization [41]. Alongside this, as the use of large-scale models has become widespread, distributed learning algorithms have emerged as a noteworthy approach for handling massive amounts of data and model parameters [5, 1]. Without a parameter server [20] aggregating all data from each local agent, decentralized algorithms that do not rely on the central structure can be advantageous when network bandwidth is low or latency is high, and they can also protect data privacy[21]. In this work, we consider the following decentralized minimax stochastic optimization problems:

$$\min_{\boldsymbol{x}\in\mathcal{X}} \max_{\boldsymbol{y}\in\mathcal{Y}} F(\boldsymbol{x},\boldsymbol{y}) := \frac{1}{m}\sum_{i=1}^{m} F_i(\boldsymbol{x},\boldsymbol{y}) := \frac{1}{m}\sum_{i=1}^{m} \mathbb{E}_{\xi_i\sim\mathcal{D}_i}[f_i(\boldsymbol{x},\boldsymbol{y};\xi_i)] \tag{1}$$

where $m$ denotes the number of agents, $F_i$ is the local loss function, $\xi_i$ represents local data stored on agent $i$, and $\mathcal{X}\subseteq\mathbb{R}^{d_{\boldsymbol{x}}}$, $\mathcal{Y}\subseteq\mathbb{R}^{d_{\boldsymbol{y}}}$. Note that the data distributions $\mathcal{D}_i$ may differ across the agents.

The most straightforward algorithm for solving the above stochastic minimax optimization problem is to apply Stochastic Gradient Descent Ascent (SGDA) [14, 22] in a decentralized manner, named

---

*Corresponding authors.

37th Conference on Neural Information Processing Systems (NeurIPS 2023).

Table 1: Main results on different cases: SC-SC, C-C, and NC-NC represent strongly-convex-strongly-concave, convex-concave, and nonconvex-nonconcave, respectively. $\widetilde{\mathcal{O}}$ means it contains the logarithmic function. $C_\lambda$ is a constant concerning about the spectral gap $1 - \lambda$ of different topology which is defined in Thm. 2. $T$ represents iterations. $L$ is Lipschitz constant. $\mu$ represents the strong convexity and strong concavity parameter. $n$ denotes the sample size in each node. $m$ denotes the number of nodes and $0 < c \leq 1$ is configurable constant.

| Cases | Measure | Bound |
|---|---|---|
| SC-SC | strong/weak primal-dual generalization gap | $\widetilde{\mathcal{O}}\left(\dfrac{C_\lambda}{T^{\frac{L}{L+\mu}}} + \dfrac{T^{1-c}}{n}\right)$ [Thm. 2] |
| | strong/weak primal-dual population risk | $\mathcal{O}\left(\dfrac{C_\lambda}{T^{\min\{\frac{1}{2}, \frac{L}{L+\mu}\}}} + \dfrac{1}{n}\right)$ [Thm. 3] |
| C-C | weak primal-dual generalization gap | $\mathcal{O}\left(\dfrac{1}{(1-\lambda)T} + \dfrac{1}{n}\right)$ [Thm. 4] |
| | weak primal-dual population risk | $\mathcal{O}\left(\dfrac{1}{(1-\lambda)T^{\frac{1}{3}}} + \dfrac{T^{\frac{1}{3}}}{n}\right)$ [Thm. 5] |
| NC-NC | weak primal-dual generalization gap | $\mathcal{O}\left((C_\lambda T^L)^{\frac{1}{c+L}}(\frac{m}{n})^{1-\frac{1}{c+L}}\right)$ [Thm. 6] |

D-SGDA. Many algorithms [36, 7, 10, 43, 35, 38, 25, 12, 24, 34, 2] have been proposed to solve Problem (1). As for the theoretical part, they mainly focus on analyzing the convergence behavior and communication complexity of their proposed algorithms. Due to the inaccessibility of the data distribution $\mathcal{D}_i$, they approximate the expectation value by averaged sum on the training dataset $\mathcal{S} = \{\mathcal{S}_1, ..., \mathcal{S}_m\}$ with local samples $\xi_{i,l_i}$ stored in local dataset $\mathcal{S}_i = \{\xi_{i,l_i}\}_{1 \leq l_i \leq n}$:

$$\min_{\boldsymbol{x} \in \mathcal{X}} \max_{\boldsymbol{y} \in \mathcal{Y}} F_{\mathcal{S}}(\boldsymbol{x}, \boldsymbol{y}), \text{ with } F_{\mathcal{S}}(\boldsymbol{x}, \boldsymbol{y}) = \frac{1}{m} \sum_{i=1}^{m} F_{\mathcal{S}_i}(\boldsymbol{x}, \boldsymbol{y}) = \frac{1}{m} \sum_{i=1}^{m} \frac{1}{n} \sum_{l_i=1}^{n} f_i(\boldsymbol{x}, \boldsymbol{y}; \xi_{i,l_i}) \tag{2}$$

However, it is insufficient to evaluate the stochastic algorithm not to consider the generalization performance, which is roughly the gap between Eq. (1) and Eq. (2). Generally speaking, saddle point of $F_{\mathcal{S}}(\boldsymbol{x}, \boldsymbol{y})$ may not be the optimal solution of $\min_{\boldsymbol{x}} \max_{\boldsymbol{y}} F(\boldsymbol{x}, \boldsymbol{y})$. As a result, the model learned by Eq. (2) may not perform well on the test dataset. In fact, the generalization gap is a crucial criterion for us to foresee the performance of the trained model on the unknown dataset. Furthermore, it is quite necessary for us to make a trade-off between the optimization error and the generalization gap to obtain models with optimal population risk (see Eq. (1)).

Concerning the stability and generalization of the minimax problem, several works [19, 30, 9, 42] have studied the generalization gap and population risk of some algorithms, including SGDA, SGDmax, PPM, and AGDA. However, these results cannot be directly extended to the decentralized case due to the additional communication step during the training process. Intuitively, the number of nodes and communication topology in decentralized training may exert a potential influence on the model's generalizability. Note that even for the decentralized minimization problem, the generalization and stability of decentralized SGD are adversely affected by an extra non-vanishing term [32], and the stability usually suffers from a constant term $\lambda^2$ [44], compared to vanilla SGD. Building upon these findings, we argue that it is worthy to investigate the generalization and stability of D-SGDA for decentralized minimax problems, where there do exist more newly unveiled problems.

To mitigate this theoretical deficiency, we present the first comprehensive analysis of the stability and generalization of D-SGDA for the decentralized minimax problem in this paper. Specifically, we develop a refined stability analysis in a decentralized manner and derive the generalization gap and population risk for D-SGDA under different settings. The main theoretical results are summarized in Table 1. And our main contributions are summarized as follows:

- *First work on the stability and generalization of D-SGDA for decentralized minimax problem.* We extend the concepts of algorithmic stability, which includes argument stability and weak stability, to the decentralized setting. And we establish a universal connection between argument stability and different measures of generalization gap in the framework of decentralization. We propose a subtle technique to distribute the "different" samples in the neighboring datasets among agents by methods of permutation and combination.
- *New theoretical results.* Our theoretical results reveal that decentralized structure does not hurt the stability and generalizability of D-SGDA compared with SGDA and explain how topology of the communication network influences the performance in stongly-convex-strongly-concave,

convex-concave, and nonconvex-nonconcave conditions (see Table 1,2). We also evaluate the optimization error and leverage it with generalization gap to obtain the optimal population risk.

- *Experiments.* We provide several numerical experiments on AUC maximization (C-C) and adversarial learning (NC-NC) in which we vary different factors to support our theoretical findings. The preliminary experimental results align with our theoretical insights.

## 2   Related Work

**Decentralized minimax problem.** Existing works mainly focus on improving the convergence rate and communication complexity. Liu et al. [23] propose DPOSG, which is firstly applied in case of nonconvex-nonconcave, i.e., GAN training, and they prove $\mathcal{O}(\epsilon^{-12})$ computational complexity and $\mathcal{O}(log(1/\epsilon))$ communication complexity on the busiest node. Xian et al. [36] propose DM-HSGD with convergence rate of $\mathcal{O}(\kappa^3\epsilon^{-3})$ and Chen et al. [7] propose DREAM with communication rounds of $\mathcal{O}(\kappa^2\epsilon^{-2}/\sqrt{1-\lambda_2})$ in nonconvex-strongly-concave condition. Chen et al. [6] propose SPIDER-GDA and achieve stochastic first-order oracle of $\mathcal{O}((n + \sqrt{n}\kappa_x\kappa_y^2)log(1/\epsilon))$ under two-sided PL condition. Rogozin et al. [31] propose a Mirror-prox based algorithm with $\mathcal{O}(\epsilon^{-1})$ communication complexity in C-C setting. Huang [15], Luo and Ye [25] accelerates by variance reduction. Beznosikov et al. [3] considers time-varying networks with heterogeneous data, Kovalev et al. [18] provides a rigorous complexity for decentralized variational inequalities.

**Stability and generalization.** There are mainly two approaches to investigating the generalization: algorithm-independent generalization, which is also called uniform convergence generalization, and algorithm-dependent generalization respectively. Where the former may degrade to a vacuous conclusion in [28] and we adopt the latter method in our paper which can better explain the generalization behavior of a detailed algorithm. Bousquet and Elisseeff [4] come up with algorithmic stability, Elisseeff et al. [8] extend the concept to randomized algorithms. Hardt et al. [13] further develop the framework by connecting algorithmic stability with the generalization gap. Sun et al. [32] and Zhu et al. [44] extend the generalization and stability analysis to D-SGD. In the minimax problem, Zhang et al. [42] focus on argument stability and prove $\mathcal{O}(1/n)$ weak and strong generalization bounds for the SC-SC condition; Farnia and Ozdaglar [9] analyze the uniform stability and generalization gap of GDA, GDmax, and PPM (proximal point method) in the case of NC-NC and Lei et al. [19] summarize the connection between different measures of stability and generalization gap and further develop the corresponding high-probability results. Xing et al. [37] specify the generalization gap for adversarial training and Yang et al. [39] investigate the stability-based generalization of SGDA with differential privacy constraints. Ozdaglar et al. [30] propose a new metric to better evaluate the generalization performance even in the case when the existing metric fails.

## 3   Problem Formulation

In this section, we provide the necessary assumptions, notations, terminologies of population risk, generalization gap, and algorithmic stability in decentralized minimax problems.

### 3.1   Basic Assumptions

**Notations.** We use bold lower case to denote vectors and bold upper case to denote matrices. $\|\cdot\|_2$ means $\ell_2$ norm for vectors and $\|\cdot\|_F$ means Frobinius norm for matrices, and we will omit the subscript when the type of norm is clear from the context. $\mathbb{1}_n \in \mathbb{R}^n$ denotes the all-one vector and $\lambda_i(\cdot)$ represents the $i$-th largest eigenvalue of a matrix. $[n] := \{1, 2, ..., n\}$.

**Assumption 1** (**Lipschitz continuous**). Each local function $f_i$ is differentiable and there exists $G > 0$ that $f_i$ is $G$-Lipschitz continuous with respect to both $\boldsymbol{x}$ and $\boldsymbol{y}$ on any given sample $\xi_i$, i.e.,

$$|f_i(\boldsymbol{x}, \boldsymbol{y}; \xi_i) - f_i(\boldsymbol{x}', \boldsymbol{y}'; \xi_i)| \leq G \left\| \begin{pmatrix} \boldsymbol{x} - \boldsymbol{x}' \\ \boldsymbol{y} - \boldsymbol{y}' \end{pmatrix} \right\|_2.$$

**Assumption 2** (**Lipschitz smooth**). Each local function $f_i$ is differentiable and there exists $L > 0$ that $f_i$ is $L$-Lipschitz smooth with respect to both $\boldsymbol{x}$ and $\boldsymbol{y}$ on any given sample $\xi_i$, i.e.,

$$\left\| \begin{pmatrix} \nabla_{\boldsymbol{x}} f_i(\boldsymbol{x}, \boldsymbol{y}; \xi_i) - \nabla_{\boldsymbol{x}} f_i(\boldsymbol{x}', \boldsymbol{y}'; \xi_i) \\ \nabla_{\boldsymbol{y}} f_i(\boldsymbol{x}, \boldsymbol{y}; \xi_i) - \nabla_{\boldsymbol{y}} f_i(\boldsymbol{x}', \boldsymbol{y}'; \xi_i) \end{pmatrix} \right\| \leq L \left\| \begin{pmatrix} \boldsymbol{x} - \boldsymbol{x}' \\ \boldsymbol{y} - \boldsymbol{y}' \end{pmatrix} \right\|_2.$$

**Definition 1** (**Convexity-Concavity**). For each local loss function $f_i(\boldsymbol{x}, \boldsymbol{y}; \xi_i)$, we say that $f_i$ is $\mu_{\boldsymbol{x}}$-strongly convex on $\boldsymbol{x}$ if for any given $\boldsymbol{y}$ and on any given sample $\xi_i$, there holds:

$$f_i(\boldsymbol{x}', \boldsymbol{y}; \xi_i) \geq f_i(\boldsymbol{x}, \boldsymbol{y}; \xi_i) + \nabla_{\boldsymbol{x}} f_i(\boldsymbol{x}, \boldsymbol{y}; \xi_i)^T (\boldsymbol{x}' - \boldsymbol{x}) + \frac{\mu_x}{2} \|\boldsymbol{x}' - \boldsymbol{x}\|^2, \mu_{\boldsymbol{x}} \geq 0, \forall \boldsymbol{x}, \boldsymbol{x}'.$$

we say that $f_i$ is $\mu_{\boldsymbol{y}}$-strongly concave on $\boldsymbol{y}$ if for any given $\boldsymbol{x}$ and on any given sample $\xi_i$, there holds:

$$f_i(\boldsymbol{x}, \boldsymbol{y}'; \xi_i) \leq f_i(\boldsymbol{x}, \boldsymbol{y}; \xi_i) + \nabla_{\boldsymbol{y}} f_i(\boldsymbol{x}, \boldsymbol{y}; \xi_i)^T (\boldsymbol{y}' - \boldsymbol{y}) - \frac{\mu_y}{2} \|\boldsymbol{y}' - \boldsymbol{y}\|^2, \mu_{\boldsymbol{y}} \geq 0, \forall \boldsymbol{y}, \boldsymbol{y}'.$$

We can call it is convex w.r.t. $\boldsymbol{x}$ when $\mu_{\boldsymbol{x}} = 0$ and concave w.r.t. $\boldsymbol{y}$ when $\mu_{\boldsymbol{y}} = 0$.

**Remark 1.** Assumptions about the Lipschitz continuity and smoothness are commonly used in the context of decentralized minimax optimization problems [42, 9, 32].

## 3.2 Decentralized Stochastic Gradient Descent Ascent (D-SGDA)

In decentralized setting, each node will exchange information alternatively and we represent the communication network between nodes as $\mathcal{G} = (\mathcal{V}, \mathcal{E})$, which is a connected graph with node set $\mathcal{V} = \{1, 2, ..., m\}$ and edge set $\mathcal{E} \subseteq \mathcal{V} \times \mathcal{V}$. Specifically, $(i, l) \in \mathcal{E}$ indicates that agent $l$ can receive information from agent $i$ and therefore we symbolize the *in* and *out* neighbors as $\mathcal{N}^{in}(i) \triangleq \{l \in \mathcal{V}, (l, i) \in \mathcal{E}\}$ and $\mathcal{N}^{out}(i) \triangleq \{l \in \mathcal{V}, (i, l) \in \mathcal{E}\}$ respectively. In an undirected graph, there is no consideration about the order, thus $(i, l) \in \mathcal{E}$ implies $(l, i) \in \mathcal{E}$ and the *in* and *out* neighbors are identical which we will abbreviate as $\mathcal{N}$ for brief. In our work, we focus on undirected graphs. The communication graph is associated with an adjacency matrix, which is also called a mixing matrix, $\boldsymbol{W} = [\omega_{ij}] \in \mathbb{R}^{m \times m}$. It implies the connection between $m$ agents that $\omega_{ij} > 0$ if and only if $(j, i) \in \mathcal{E}$, otherwise $\omega_{ij} = 0$. And there are some basic assumptions about the mixing matrix which is commonly used in decentralized settings [17, 21, 23, 32].

**Assumption 3** (**Mixing matrix**). We assume the mixing matrix $\boldsymbol{W} = [\omega_{ik}] \in [0, 1]^{m \times m}$ defined on the graph $\mathcal{G} = (\mathcal{V}, \mathcal{E})$ is a symmetric doubly stochastic matrix, which holds the property that $\boldsymbol{W}^T = \boldsymbol{W}$ and $\boldsymbol{W} \mathbb{1}_m = \mathbb{1}_m, \mathbb{1}_m^T \boldsymbol{W} = \mathbb{1}_m^T$. Besides, we assume $\lambda := \max\{|\lambda_2|, |\lambda_m(\boldsymbol{W})|\} \in (0, 1)$.

For a symmetric doubly stochastic matrix, $\boldsymbol{W}$ holds the property that: $\lambda_1 = 1$. For different topologies, $\lambda \to 1$ implies the sparsity while $\lambda \to 0$ implies the complete connection. Nedić et al. [29] and Ying et al. [40] list upper bounds for the spectral gap $1 - \lambda$ over the commonly communication network. More knowledge on decentralized optimization is placed in **Appendix** A.

In this paper, we study the decentralized minimax problem solved via D-SGDA (see Algorithm 1). We use the superscript to denote the $t$-th iteration and the subscript to denote the $i$-th local agent. During iteration, each client first computes its local gradient approximation by $\nabla_{\boldsymbol{x}} f_i(\boldsymbol{x}_i^t, \boldsymbol{y}_i^t; \xi_{i, j_t(i)})$ and $\nabla_{\boldsymbol{y}} f_i(\boldsymbol{x}_i^t, \boldsymbol{y}_i^t; \xi_{i, j_t(i)})$ respectively where $j_t(i)$ is randomly chosen from $[n]$. Then each client communicates with its neighbor $\mathcal{N}(i)$ and updates by SGDA.

---

**Algorithm 1** D-SGDA

**Initialize:** $\boldsymbol{x}_i^0 = 0; \boldsymbol{y}_i^0 = 0, i = 1, ..., m$
  **for** $t = 1, 2, \cdots, T$ **do**

$$\boldsymbol{x}_i^{t+1} = P_{\mathcal{X}}\left(\sum_{k \in \mathcal{N}(i)} \omega_{ik} \boldsymbol{x}_k^t - \eta_{\boldsymbol{x}, t} \nabla_{\boldsymbol{x}} f_i(\boldsymbol{x}_i^t, \boldsymbol{y}_i^t; \xi_{i, j_t(i)})\right)$$

$$\boldsymbol{y}_i^{t+1} = P_{\mathcal{Y}}\left(\sum_{k \in \mathcal{N}(i)} \omega_{ik} \boldsymbol{y}_k^t + \eta_{\boldsymbol{y}, t} \nabla_{\boldsymbol{y}} f_i(\boldsymbol{x}_i^t, \boldsymbol{y}_i^t; \xi_{i, j_t(i)})\right)$$

  **end for**
**Output:** $\boldsymbol{x}^t = \frac{1}{m} \sum_{i=1}^m \boldsymbol{x}_i^t; \boldsymbol{y}^t = \frac{1}{m} \sum_{i=1}^m \boldsymbol{y}_i^t$

---

## 3.3 Generalization Gap

In a sense, we obtain the result by minimaxing the empirical one $F_{\mathcal{S}}$ in Eq. (2), which differs from the population one $F$ in Eq. (1). So we can not guarantee the same performance on the unknown distribution as on the training dataset. And therefore the gap between the empirical one and the population one reflects the ability of generalization. Unlike the standard learning theory which only contains a single variable that can directly define the population risk and empirical risk by the objective function[4]. Owing to the structure of minimax, there are different methods to define the population and empirical risk as concluded in [19], where primal-dual measure starts from the idea of duality gap in optimization. And we first introduce two types of population risks as follows.

**Definition 2** (**Population risk**). For a randomized model $(\boldsymbol{x}, \boldsymbol{y})$, we define the population risk as:

1. *Weak primal-dual population risk*: $\Delta^w(\boldsymbol{x}, \boldsymbol{y}) = \sup_{\boldsymbol{y}' \in \mathcal{Y}} \mathbb{E}[F(\boldsymbol{x}, \boldsymbol{y}')] - \inf_{\boldsymbol{x}' \in \mathcal{X}} \mathbb{E}[F(\boldsymbol{x}', \boldsymbol{y})].$

2. *Strong primal-dual population risk*: $\Delta^s(\boldsymbol{x}, \boldsymbol{y}) = \mathbb{E}[\sup_{\boldsymbol{y}' \in \mathcal{Y}} F(\boldsymbol{x}, \boldsymbol{y}') - \inf_{\boldsymbol{x}' \in \mathcal{X}} F(\boldsymbol{x}', \boldsymbol{y})].$

Here the expectation is taken over the randomness of the model. By replacing function $F$ with function $F_{\mathcal{S}}$ (see Eq. (2)) in Def. 2 when considering the empirical risk, we obtain the corresponding *weak primal-dual empirical risk* $\Delta_{\mathcal{S}}^w(\boldsymbol{x}, \boldsymbol{y}) = \sup_{\boldsymbol{y}' \in \mathcal{Y}} \mathbb{E}[F_{\mathcal{S}}(\boldsymbol{x}, \boldsymbol{y}')] - \inf_{\boldsymbol{x}' \in \mathcal{X}} \mathbb{E}[F_{\mathcal{S}}(\boldsymbol{x}', \boldsymbol{y})]$ and *strong primal-dual empirical risk* $\Delta_{\mathcal{S}}^s(\boldsymbol{x}, \boldsymbol{y}) = \mathbb{E}[\sup_{\boldsymbol{y}' \in \mathcal{Y}} F_{\mathcal{S}}(\boldsymbol{x}, \boldsymbol{y}') - \inf_{\boldsymbol{x}' \in \mathcal{X}} F_{\mathcal{S}}(\boldsymbol{x}', \boldsymbol{y})]$ respectively. Subtracting the empirical risk from population risk, we can define the generalization gap as follows.

**Definition 3 (Generalization gap).** For a randomized model $(\boldsymbol{x}, \boldsymbol{y})$, we define the corresponding generalization gap as:

1. *Weak primal-dual generalization gap*: $\epsilon_{gen}^w(\boldsymbol{x}, \boldsymbol{y}) = \Delta^w(\boldsymbol{x}, \boldsymbol{y}) - \Delta_{\mathcal{S}}^w(\boldsymbol{x}, \boldsymbol{y}).$
2. *Strong primal-dual generalization gap*: $\epsilon_{gen}^s(\boldsymbol{x}, \boldsymbol{y}) = \Delta^s(\boldsymbol{x}, \boldsymbol{y}) - \Delta_{\mathcal{S}}^s(\boldsymbol{x}, \boldsymbol{y}).$

**Remark 2.** Notice that we revise the name as generalization gap to avoid misunderstanding since "generalization error" usually refers to the empirical risk in learning theory (see [4]). Our primal target (see Problem (1)) is to obtain small population risk (Def. 2) which can be considered as a summation like $\Delta^w(\boldsymbol{x}, \boldsymbol{y}) = \epsilon_{gen}^w(\boldsymbol{x}, \boldsymbol{y}) + \Delta_{\mathcal{S}}^w(\boldsymbol{x}, \boldsymbol{y})$, where generalization gap reflects how well the model generalizes and empirical risk reflects the optimization performance. The strong primal-dual risk is stronger than the weak one due to $\Delta^w(\boldsymbol{x}, \boldsymbol{y}) \leq \Delta^s(\boldsymbol{x}, \boldsymbol{y})$ according to Jensen's inequality. But in some cases, it is sufficient to bound weak primal-dual population risk such as the MDP [42].

## 3.4 Algorithmic Stability

Inspired by [13] that the generalization gap of an $\epsilon$-stable algorithm can be bounded by $\epsilon$. And this connection between stability and generalization for minimax problem is furthermore established in [19]. For a randomized algorithm $\mathcal{A}$ solving the problem (2), we use $\mathcal{A}(\mathcal{S}) = (\mathcal{A}_{\boldsymbol{x}}(\mathcal{S}), \mathcal{A}_{\boldsymbol{y}}(\mathcal{S}))$ to denote the output of applying algorithm $\mathcal{A}$ on dataset $\mathcal{S} = \{\mathcal{S}_1, ..., \mathcal{S}_m\}$. Let $\mathbb{E}_{\mathcal{S}}$ and $\mathbb{E}_{\mathcal{A}}$ denote taking expectation on the randomness of the algorithm $\mathcal{A}$ and the dataset $\mathcal{S}$ respectively. Sometimes we omit the subscript as $\mathbb{E}[\cdot]$ when it is clear from the context. Next, we first refine the definitions of algorithmic stability in a decentralized manner and then provide a connection between algorithmic stability and generalization gap in the framework of decentralization.

**Definition 4 (Decentralized neighboring dataset).** We call $\mathcal{S}, \mathcal{S}'$ the decentralized neighboring datasets when there are at most one different sample in each local dataset, where $\mathcal{S} = \{\mathcal{S}_1, \mathcal{S}_2, ..., \mathcal{S}_m\}, \mathcal{S}' = \{\mathcal{S}_1', \mathcal{S}_2', ..., \mathcal{S}_m'\}$ and each $\mathcal{S}_i$ and $\mathcal{S}_i'$ differs by at most one sample.

**Definition 5 (Decentralized algorithmic stability).** For a randomized algorithm $\mathcal{A}$, we say:

1. $\mathcal{A}$ is $\boldsymbol{\epsilon}$-**argument stable** if there holds for any neighboring datasets $\mathcal{S}, \mathcal{S}'$:

$$\mathbb{E}_{\mathcal{A}} \left\| \left( \begin{array}{c} \mathcal{A}_{\boldsymbol{x}}(\mathcal{S}) - \mathcal{A}_{\boldsymbol{x}}(\mathcal{S}') \\ \mathcal{A}_{\boldsymbol{y}}(\mathcal{S}) - \mathcal{A}_{\boldsymbol{y}}(\mathcal{S}') \end{array} \right) \right\|_2 \leq \epsilon.$$

2. $\mathcal{A}$ is $\boldsymbol{\epsilon}$-**weakly stable** if there holds for any neighboring datasets $\mathcal{S}, \mathcal{S}'$:

$$\sup_{\boldsymbol{\xi}} \left[ \sup_{\boldsymbol{y}' \in \mathcal{Y}} \mathbb{E}_{\mathcal{A}}[\boldsymbol{f}(\mathcal{A}_{\boldsymbol{x}}(\mathcal{S}), \boldsymbol{y}'; \boldsymbol{\xi}) - \boldsymbol{f}(\mathcal{A}_{\boldsymbol{x}}(\mathcal{S}'), \boldsymbol{y}'; \boldsymbol{\xi})] + \sup_{\boldsymbol{x}' \in \mathcal{X}} \mathbb{E}_{\mathcal{A}}[\boldsymbol{f}(\boldsymbol{x}', \mathcal{A}_{\boldsymbol{y}}(\mathcal{S}); \boldsymbol{\xi}) - \boldsymbol{f}(\boldsymbol{x}', \mathcal{A}_{\boldsymbol{y}}(\mathcal{S}'); \boldsymbol{\xi})] \right] \leq \epsilon.$$

where $\boldsymbol{\xi} \triangleq \{\xi_1, ..., \xi_m\}$ denotes sample index with $\xi_i \in \mathcal{D}_i$ and $\boldsymbol{f}(\boldsymbol{x}, \boldsymbol{y}; \boldsymbol{\xi}) \triangleq \frac{1}{m} \sum_{i=1}^m f_i(\boldsymbol{x}, \boldsymbol{y}; \xi_i).$

And we further specify the stability error as $\epsilon_{sta}^{arg}(\mathcal{A})$ and $\epsilon_{sta}^w(\mathcal{A})$ respectively.

**Remark 3.** The definition of neighboring datasets in the decentralized setting can degenerate to the traditional neighboring datasets where there is at most a single different sample between $\mathcal{S}$ and $\mathcal{S}'$. And the refined concepts of algorithmic stability are also fit for classic stability without decentralization in [19]. These facts validate that our definitions above are well-defined. Notice that the argument stability can imply weak stability because of the property of Lipschitz continuity (see Assumption 1). Specifically speaking, when algorithm $\mathcal{A}$ is $\epsilon$-argument stable, then it is $\sqrt{2}G\epsilon$-weakly stable. So we will mainly focus on the argument stability in the rest part.

**Theorem 1 (Connection).** *For an $\epsilon$-argument stable decentralized algorithm $\mathcal{A}$, under Assumption 1, we have the following different measures of generalization gap:*
*a. Weak primal-dual generalization gap:* $\epsilon_{gen}^w(\mathcal{A}_{\boldsymbol{x}}(\mathcal{S}), \mathcal{A}_{\boldsymbol{y}}(\mathcal{S})) \leq \sqrt{2}G\epsilon.$
*b. Strong primal-dual generalization gap holds under extra Assumption 2 when $f_i$ is $\mu_{\boldsymbol{x}}SC$-$\mu_{\boldsymbol{y}}SC$:*

$$\epsilon_{gen}^s(\mathcal{A}_{\boldsymbol{x}}(\mathcal{S}), \mathcal{A}_{\boldsymbol{y}}(\mathcal{S})) \leq G\sqrt{2 + \frac{2L^2}{\mu^2}}\epsilon, \text{ where } \mu \triangleq \min\{\mu_{\boldsymbol{x}}, \mu_{\boldsymbol{y}}\}.$$

**Remark 4.** The complete proof is provided in **Appendix** D. We establish the connection between argument stability and a different measure of generalization gap under different constraints, where weak primal-dual generalization gap does not require any convexity or concavity in part a. and strong primal-dual generalization gap requires both strong convexity and strong concavity in part b.. And this connection is not limited to the single decentralized algorithm D-SGDA, but a universal connection for decentralized minimax algorithms. Actually weak stability is sufficient to prove the weak primal-dual generalization gap in part a. that when algorithm $\mathcal{A}$ is $\epsilon$-weakly stable, we have $\mathbb{E}_{\mathcal{A},\mathcal{S}}[\epsilon_{gen}^w(\mathcal{A}_{\boldsymbol{x}}(\mathcal{S}), \mathcal{A}_{\boldsymbol{y}}(\mathcal{S}))] \leq \epsilon$. And the theorem implies that once we have access to the stability error, we can derive the generalization gap as an accompanying result.

## 4 Theoretical Results on D-SGDA

In this section, we study algorithmic stability and generalization bound in SC-SC, C-C, and NC-NC settings in Sec 4.1, Sec 4.2 and Sec 4.3, respectively. Due to the space limitation, the proofs are placed in the **Appendix** E, F, G respectively.

### 4.1 Results on Strongly-Convex-Strongly-Concave Case

Below, we first characterize the argument stability with fixed and decaying learning rates, respectively.

**Theorem 2** (**Argument Stability**). *Under Assumption 1,2,3 when each $f_i$ is $\mu_{\boldsymbol{x}}$-strongly convex and $\mu_{\boldsymbol{y}}$-strongly concave, we have the argument stability bound for D-SGDA (denoted as $\mathcal{A}$):*

$$\epsilon_{sta}^{arg}(\mathcal{A}) \leq \frac{2G}{n} \sum_{k=0}^{T-1} \eta_k^{max} \prod_{s=k+1}^{T-1} (1-\eta_s^{min}\frac{L\mu}{L+\mu}) + 4GL \sum_{k=1}^{T-1} \left(\eta_k^{max} \sum_{s=0}^{k-1} \eta_s^{max} \lambda^{k-1-s}\right) \prod_{j=k+1}^{T-1}(1-\eta_j^{min}\frac{L\mu}{L+\mu}).$$

*where $\eta_t^{max} \triangleq \max\{\eta_{\boldsymbol{x},t}, \eta_{\boldsymbol{y},t}\}$, $\eta_t^{min} \triangleq \min\{\eta_{\boldsymbol{x},t}, \eta_{\boldsymbol{y},t}\}$, $\mu = \min\{\mu_{\boldsymbol{x}}, \mu_{\boldsymbol{y}}\}$. Furthermore,*

*a. for fixed learning rates, $\epsilon_{sta}^{arg}(\mathcal{A}) \leq 2G\frac{L+\mu}{\eta^{min}L\mu}(\frac{2(\eta^{max})^2 L}{1-\lambda} + \frac{\eta^{max}}{n})$.*

*b. for decaying learning rates with $\eta_t^{min} = \frac{1}{\mu(t+1)}$ and $\eta_t^{max} = \frac{1}{\mu(t+1)^c}, c \leq 1$, we have:*

$$\epsilon_{sta}^{arg}(\mathcal{A}) \leq \frac{2G}{\mu n T^{\frac{L}{L+\mu}}} \sum_{k=0}^{T-1} \frac{1}{(k+1)^{c-\frac{L}{L+\mu}}} + \frac{4GL}{\mu^2 T^{\frac{L}{L+\mu}}} \sum_{k=1}^{T-1} \frac{1}{(k+1)^{c-\frac{L}{L+\mu}}}\frac{C_\lambda}{k^c}.$$

*where $C_\lambda \triangleq \frac{(k/e)^c}{\lambda(\ln\frac{1}{\lambda})^c} + \frac{2e^{-1}}{\lambda \ln\frac{1}{\lambda}} + \frac{2^c}{\lambda \ln\frac{1}{\lambda}}$.*

**Remark 5. (i) Bound analysis and comparison.** For case a.with fixed learning rates, the argument stability is bounded by $\mathcal{O}(\frac{\eta}{1-\lambda} + \frac{1}{n})$, which can reach $\mathcal{O}(\frac{1}{(1-\lambda)T} + \frac{1}{n})$ when $\eta \sim \frac{1}{T}$. For case b. with decaying learning rates, we should require $2c \geq \frac{L}{L+\mu}+1$ otherwise the bound can tend to infinity, then we have $\epsilon_{sta}^{arg}(\mathcal{A}) \leq \frac{2G}{\mu(1-c+\frac{L}{L+\mu})}\frac{T^{1-c}}{n} + \frac{4GLC_\lambda}{\mu^2 T^{\frac{L}{L+\mu}}}(\frac{\mathbf{1}_{2c>L/(L+\mu)+1}}{2c-\frac{L}{L+\mu}-1} + \ln T \cdot \mathbf{1}_{2c=L/(L+\mu)+1})$. Both can be bounded by $\widetilde{\mathcal{O}}(\frac{T^{1-c}}{n} + \frac{C_\lambda}{T^{\frac{L}{L+\mu}}})$ with $\widetilde{\mathcal{O}}$ containing the logarithmic function into consideration, which matches the corresponding results for SGDA (see Thm.2.(e) in [19]) except an extra multiplication factor $C_\lambda$: $\widetilde{\mathcal{O}}(\frac{1}{\sqrt{T}} + \frac{1}{N})$ with decaying learning rates and $N$ the total sample size. **(ii) Factor influence.** It is apparent that a smaller stability bound is achieved with larger sample size $n$, and smaller learning rates $\eta$ (which is associated with larger $T$). And notice that the decaying learning rates may slightly underperform than the fixed one, it is easy to explain that at the beginning decaying learning rates are too large and therefore result in weaker stability. The influence of these factors is consistent with vanilla SGDA. **(iii) Effect of topology.** So the major difference lies in $C_\lambda$ and the number of nodes $m$. For $C_\lambda = \frac{(k/e)^c}{\lambda(\ln\frac{1}{\lambda})^c} + \frac{2e^{-1}}{\lambda \ln\frac{1}{\lambda}} + \frac{2^c}{\lambda \ln\frac{1}{\lambda}}$, when $\lambda \to 1$, it is bounded by $\mathcal{O}(\frac{1}{\lambda \ln\frac{1}{\lambda}})$ and when $\lambda \to 0$, $C_\lambda$ is bounded by $\mathcal{O}(\frac{1}{\lambda(\ln\frac{1}{\lambda})^c})$. We list some common topology with estimated value of $\lambda$ along with the upper bound of $C_\lambda$ in Table 2. We can conclude that topology with a denser connection (larger spectral gap $1-\lambda$) will have a smaller stability error, i.e., be more stable. In extreme conditions, the fully connected network will behave as well as vanilla SGDA, and the stability error of the disconnected network will diverge. Furthermore, under the same topology, fewer nodes will result in better stability.

**Generalization gap.** According to Thm. 1, we can directly derive the weak and strong primal-dual generalization gap of D-SGDA as $\sqrt{2}G\epsilon_{sta}^{arg}$ and $G\sqrt{2 + \frac{2L^2}{\mu^2}}\epsilon_{sta}^{arg}$ respectively. Therefore, we hold the same analysis as stability in above Remark 5.

Next, we will first derive the optimization error and then provide the population risk by decomposition $\Delta^s(\boldsymbol{x}, \boldsymbol{y}) = \epsilon_{gen}^s(\boldsymbol{x}, \boldsymbol{y}) + \Delta_{\mathcal{S}}^s(\boldsymbol{x}, \boldsymbol{y})$. Population risk is an important evaluation for the performance of a stochastic learning algorithm, which will evaluate how our model obtained by training dataset behave over the whole distribution. Notice that we will use the average output instead of the last iterate in analyzing the optimization errors. We denote that:

Table 2: $\lambda$ value of different topology. Here 0 means the extra term will disappear and N/A means the term will diverge.

| Topology | $\lambda$([40]) | $C_\lambda$ | $\frac{1}{1-\lambda}$ |
|---|---|---|---|
| fully connected | 0 | 0 | 0 |
| exponential | $1 - \frac{2}{1+\ln m}$ | $\mathcal{O}(\ln m)$ | $\mathcal{O}(\ln m)$ |
| grid | $1 - \frac{1}{m \ln m}$ | $\mathcal{O}(m \ln m)$ | $\mathcal{O}(m \ln m)$ |
| ring | $1 - \frac{16\pi^2}{3m^2}$ ([21]) | $\mathcal{O}(m^2)$ | $\mathcal{O}(m^2)$ |
| star | $1 - \frac{1}{m^2}$ | $\mathcal{O}(m^2)$ | $\mathcal{O}(m^2)$ |
| disconnected | 1 | N/A | N/A |

$$\boldsymbol{x}_{ave}^T \triangleq \frac{\sum_{t=0}^{T-1} \eta_{\boldsymbol{x},t}\boldsymbol{x}^t}{\sum_{t=0}^{T-1} \eta_{\boldsymbol{x},t}}, \quad \boldsymbol{y}_{ave}^T \triangleq \frac{\sum_{t=0}^{T-1} \eta_{\boldsymbol{y},t}\boldsymbol{y}^t}{\sum_{t=0}^{T-1} \eta_{\boldsymbol{y},t}}. \tag{3}$$

**Theorem 3** (**Strong primal-dual population risk**). *Under Assumption 1,2,3, when each $f_i$ is $\mu_{\boldsymbol{x}}$SC-$\mu_{\boldsymbol{y}}$SC, we have the strong primal-dual population risk as follows, where $\eta_t^{max} \triangleq \max\{\eta_{\boldsymbol{x},t}, \eta_{\boldsymbol{y},t}\}$, $\eta_t^{min} \triangleq \min\{\eta_{\boldsymbol{x},t}, \eta_{\boldsymbol{y},t}\}$, $\mu = \min\{\mu_{\boldsymbol{x}}, \mu_{\boldsymbol{y}}\}$, and $(\boldsymbol{x}_{ave}^T, \boldsymbol{y}_{ave}^T)$ is defined in Eq. (3):*

*a. for fixed learning rates,*

$$\Delta^s(\boldsymbol{x}_{ave}^T, \boldsymbol{y}_{ave}^T) \leq G\sqrt{2 + \frac{2L^2}{\mu^2}}\left(2G\frac{L+\mu}{\eta^{min}L\mu}\left(\frac{2(\eta^{max})^2 L}{1-\lambda} + \frac{\eta^{max}}{n}\right)\right) + \frac{C_{\boldsymbol{x}}^2 + C_{\boldsymbol{y}}^2}{2\eta^{min}T}$$

$$+ \eta^{max}G^2 + \frac{4(C_{\boldsymbol{x}}+C_{\boldsymbol{y}})GL\eta^{max}}{1-\lambda} + \frac{2(C_{\boldsymbol{x}}+C_{\boldsymbol{y}})G}{\sqrt{T}}.$$

*b. for decaying learning rates that $\eta_t^{min} = \frac{1}{\mu(t+1)}$ and $\eta_t^{max} = \frac{1}{\mu(t+1)^c}$ with $c \leq 1$ and $2c \geq \frac{L}{L+\mu}+1$,*

$$\Delta^s(\boldsymbol{x}_{ave}^T, \boldsymbol{y}_{ave}^T)$$

$$\leq G\sqrt{2 + \frac{2L^2}{\mu^2}}\left(\frac{2G}{\mu(1-c+\frac{L}{L+\mu})}\frac{T^{1-c}}{n} + \frac{4GLC_\lambda}{\mu^2 T^{\frac{L}{L+\mu}}}\left(\frac{\mathbf{1}_{2c \neq L/(L+\mu)+1}}{2c - \frac{L}{L+\mu}-1} + \ln T \cdot \mathbf{1}_{2c=L/(L+\mu)+1}\right)\right)$$

$$+ \frac{2G(C_{\boldsymbol{x}}+C_{\boldsymbol{y}})}{\sqrt{T}} + \frac{G^2}{2\mu}\left(\frac{1+\ln T}{T} + \frac{\mathbf{1}_{c \neq 1}}{(1-c)T^c} + \frac{(1+\ln T)\mathbf{1}_{c=1}}{T}\right) + \frac{4GLC_\lambda(C_{\boldsymbol{x}}+C_{\boldsymbol{y}})}{\mu T^c}\left(\frac{\mathbf{1}_{c \neq 1}}{1-c} + \ln T \cdot \mathbf{1}_{c=1}\right).$$

**Remark 6.** By Jensen's inequality (see Remark 2), we can conclude that weak primal-dual population risk also satisfies the conclusions above. **(i) Bound analysis and comparison.** For fixed learning rates, it is interesting to see that we should choose $\eta \sim 1/\sqrt{T}$ to get the optimal population risk of $\mathcal{O}(\frac{1}{n} + \frac{1}{(1-\lambda)\sqrt{T}})$, although when $\eta \sim 1/T$ we can get optimal generalization performance but the optimization error will not converge. While for the decaying learning rates, the population risk bound is $\widetilde{\mathcal{O}}(T^{1-c}/n + C_\lambda/T^{\min\{\frac{1}{2}, \frac{L}{L+\mu}\}})$, which matches the corresponding results for SGDA (see Thm.3.(c) in [19]) of $\mathcal{O}(\ln N/N\mu)$ when $n \sim T^{\min\{\frac{1}{2}, \frac{L}{L+\mu}\}}$. **(ii) Topology influence.** We omit the trivial factor influence analysis here (or see Remark 5). And the effect of topology is captured quantitatively by $C_\lambda$ and $\frac{1}{1-\lambda}$ which have been discussed in Remark 5 and Table 2.

## 4.2 Results on Convex-Concave Case

In this section, we provide the argument stability and weak primal-dual population risk of D-SGDA algorithm for the NC-NC condition in the following theorems with proof in Appendix F.

**Theorem 4** (**Argument Stability**). *Under Assumption 1,2,3, when each $f_i$ is convex-concave, we have the argument stability bound for D-SGDA (denoted as $\mathcal{A}$):*

$$\epsilon_{sta}^{arg}(\mathcal{A}) \leq \frac{2G}{n}\sum_{k=0}^{T-1} \eta_k^{max} + 4GL\sum_{k=1}^{T-1}\left(\eta_k^{max}\sum_{s=0}^{k-1} \eta_s^{max}\lambda^{k-1-s}\right).$$

**Remark 7. (i) Bound analysis and comparison.** When there is no strong convexity or strong concavity, we can no longer choose the decaying learning rates, otherwise the argument stability error may not converge. For fixed learning rates, $\epsilon_{sta}^{arg}$ is upper bounded by $\mathcal{O}(\frac{\eta T}{n} + \frac{\eta^2 T}{1-\lambda})$, which is slightly looser than SC-SC condition. While we can still choose $\eta \sim 1/T$ to obtain optimal result $\mathcal{O}(\frac{1}{n} + \frac{1}{(1-\lambda)T})$. Besides, compared with the corresponding result for SGDA (see Thm.2.(b) in [19]) of $\mathcal{O}(\frac{\sqrt{T}}{N} + \frac{1}{\sqrt{N}})$, we can approach it when $\eta \sim 1/T^{3/4}$ and $n \sim T^{3/4}$. **(ii) Topology influence.** In C-C condition, the effect of topology on the stability is quantified by $\frac{1}{1-\lambda}$ which has been discussed in Table 2. And we can conclude that denser topology is more stable and fewer nodes will increase stability under the same topology.

**Generalization gap.** Thm. 1 implies $\epsilon_{gen}^w(\mathcal{A}_{\boldsymbol{x}}(\mathcal{S}), \mathcal{A}_{\boldsymbol{y}}(\mathcal{S})) \leq \sqrt{2}G\epsilon_{sta}^{arg}(\mathcal{A})$. So the generalization gap holds with the same quantitative analysis as stability above. Analogously we can present the weak primal-dual population risk in the following theorem.

**Theorem 5** (**Weak primal-dual population risk**). *Under Assumption 1,2,3, when each $f_i$ C-C, we have the weak primal-dual population risk as follows, where $\eta_t^{max} \triangleq \max\{\eta_{\boldsymbol{x},t}, \eta_{\boldsymbol{y},t}\}$, $\eta_t^{min} \triangleq \min\{\eta_{\boldsymbol{x},t}, \eta_{\boldsymbol{y},t}\}$, $\mu = \min\{\mu_{\boldsymbol{x}}, \mu_{\boldsymbol{y}}\}$, and $(\boldsymbol{x}_{ave}^T, \boldsymbol{y}_{ave}^T)$ is defined in Eq. (3):*

$$\Delta^w(\boldsymbol{x}_{ave}^T, \boldsymbol{y}_{ave}^T) \leq \sqrt{2}G\left(\frac{2G\eta^{max}T}{n} + \frac{4GL(\eta^{max})^2 T}{1-\lambda}\right) + \frac{C_{\boldsymbol{x}}^2 + C_{\boldsymbol{y}}^2}{2\eta^{min}T}$$
$$+ \eta^{max}G^2 + \frac{4(C_{\boldsymbol{x}} + C_{\boldsymbol{y}})GL\eta^{max}}{1-\lambda} + \frac{2(C_{\boldsymbol{x}} + C_{\boldsymbol{y}})G}{\sqrt{T}}.$$

**Remark 8.** The weak primal-dual population risk attains optimal of $\mathcal{O}(\frac{T^{1/3}}{n} + \frac{1}{(1-\lambda)T^{1/3}})$ when we choose $\eta^{max} = \eta^{min} \sim 1/T^{\frac{2}{3}}$. Note that we select $\eta \sim 1/T$ to obtain optimal generalization performance (see Remark 7), but the optimization error will diverge in that case. Compared with the result of SGDA: $\mathcal{O}(N^{-1/2})$ (see Thm.3.(b) in [19]), our result can approach it by $n^{1/2} \sim T^{1/3}$. Then the effect of topology and number of nodes on the population risk is reflected by $\frac{1}{1-\lambda}$, which has been discussed in Remark 5 and Table 2.

### 4.3 Results on Nonconvex-Nonconcave Case

In this section, we present the weak stability and weak primal-dual generalization gap of D-SGDA algorithm for the NC-NC problem in the following theorem with proof in Appendix G.

**Theorem 6** (**Weak stability**). *Under Assumption 1,2,3, denoting D-SGDA algorithm as $\mathcal{A}$, we have the following weak stability bound when $\eta_t^{max} \triangleq \max\{\eta_{\boldsymbol{x},t}, \eta_{\boldsymbol{y},t}\}$, and $\eta_t^{min} \triangleq \min\{\eta_{\boldsymbol{x},t}, \eta_{\boldsymbol{y},t}\}$:*

*a. for fixed learning rates that $\eta_t^{max} = \eta^{max}$, and $\eta_t^{min} = \eta^{min}$,*

$$\epsilon_{sta}^w(\mathcal{A}) \leq 2\sqrt{2}G^2\left(\frac{\eta^{max}T}{n} + \frac{2L\eta^{max2}T}{1-\lambda}\right).$$

*b. for decaying learning rates that $\eta_t^{min} = \frac{1}{t+1}$, and $\eta_t^{max} = \frac{1}{(t+1)^c}, c \leq 1$,*

$$\epsilon_{sta}^w(\mathcal{A}) \leq (c+L)(c+L-1)^{\frac{1}{c+L}}\left(\frac{2\sqrt{2}G^2T^L}{(c+L-1)n} + \frac{4\sqrt{2}G^2LC_\lambda T^L}{2c+L-1}\right)^{\frac{1}{c+L}}\left(\frac{Bm}{n}\right)^{1-\frac{1}{c+L}}.$$

**Weak primal-dual generalization gap.** According to Thm. 1, we can derive the weak primal-dual generalization gap as $\epsilon_{gen}^w(\mathcal{A}_{\boldsymbol{x}}(\mathcal{S}), \mathcal{A}_{\boldsymbol{y}}(\mathcal{S})) \leq \sqrt{2}G\epsilon_{sta}^w(\mathcal{A})$ for D-SGDA in NC-NC condition.

**Remark 9.** For case a. with fixed learning rates, the weak stability and weak primal-dual generalization gap is bounded by $\mathcal{O}(\frac{\eta T}{n} + \frac{\eta^2 T}{1-\lambda})$, which can reach $\mathcal{O}(\frac{1}{n} + \frac{1}{(1-\lambda)T})$ when $\eta \sim \frac{1}{T}$. For case b. with decaying learning rates, the stability and generalization gap is bounded by $\mathcal{O}((C_\lambda)^{\frac{1}{c+L}}T^{\frac{L}{c+L}}(\frac{m}{n})^{1-\frac{1}{c+L}})$. It is evident to analyze the influence of factors that, larger sample size and fewer nodes will result in a smaller stability error and generalization gap, which coincides with results in (S)C-(S)C conditions (see Remark 5). Approaching to the weak primal-dual generalization bound of $\mathcal{O}(n^{-\frac{2c\rho+1}{2c\rho+3}}T^{\frac{2c\rho}{2c\rho+3}})$ provided $\rho$-weakly-convex-weakly-concave (see Thm.5 in [19]). $C_\lambda$ reflects the effect of topology on the stability and generalization gap and its value has been discussed in Reamrk 5 and Table 2.

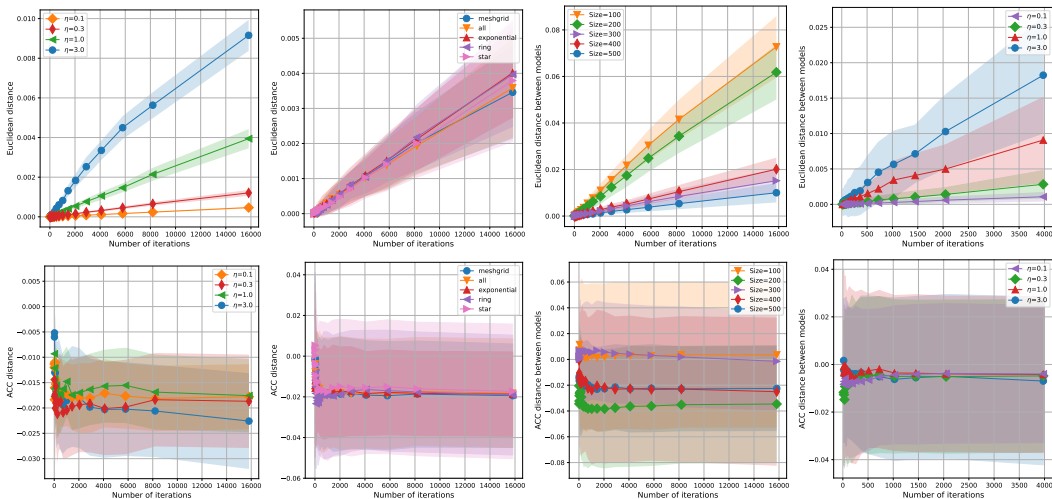

Figure 1: $\Delta$ against the number of iterations, with the first row showing different settings. From left to right, the settings include varying learning rates, communication typologies, sample sizes on the `w5a` dataset, and learning rates(`svmguide3`). The generalization error is displayed at the bottom accordingly.

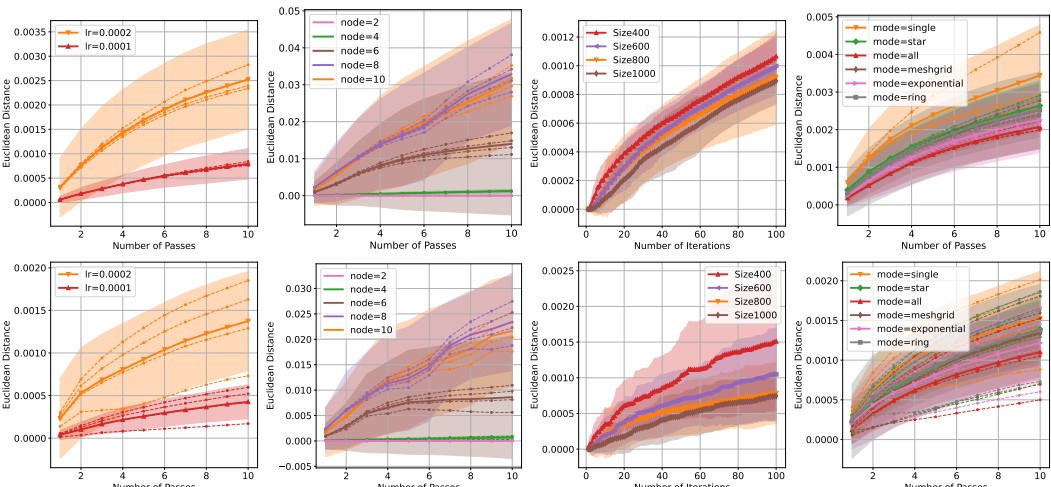

Figure 2: The first row shows the performance of the generator. From left to right, the settings include varying learning rates, the number of nodes, sample sizes, and communication typologies on the `MNIST` dataset. The performance of the discriminator is displayed at the bottom accordingly. The dashes denote different layers.

## 5 Experiments

**Experiments Setup.** We evaluate our theoretical results of the C-C case by adopting the `SOLAM` method [41] to solve the AUC problem on two datasets `svmguide` and `w5a`, and the NC-NC case by solving the generative adversarial network on `MNIST`. We extend the methods for both cases to a decentralized implementation. Our experimental setting follows the way conducted in [13, 19] to study how the stability and generalizability of D-SGDA would behave along the learning process with different factors, including learning rates, typologies, nodes, and sample sizes. We employ the same randomized method to generate two model sequences, one for the original data and another for a one-observation perturbing data, and subsequently calculate the Euclidean distance $\Delta$ between their respective parameter sets. Additional implementation details can be found in the **Appendix**.

**Results analysis:** From Fig. 1 and Fig. 2, we can observe that: (i) faster learning rates, fewer number of nodes and smaller sample size can result in a smaller Euclidean distance between weights and a smaller difference between training dataset and validation dataset; (ii) the performance of different topology on stability: fully connected(all) > exponential > grid ≈ ring ≥ star > disconnected(single).

These validate our theoretical results of the algorithmic stability for D-SGDA (see Remark 5 below Thm. 2). And the impact of different topologies also coincides with our discussion in Table 2.

## 6 Conclusion

In this paper, we provide the first comprehensive analysis for the stability and generalization of D-SGDA for decentralized minimax problems. Our theoretical results show that a decentralized structure does not destroy the stability and generalization of D-SGDA, instead we can leverage between the iterations and the number of nodes, as well as sample size to achieve better population performance. Numerical experiments also validate our theory. Our analysis technique has the potential used for studying the ability and generalization of other decentralized minimax algorithms.

**Limitation&Broader Impacts.** In our analysis, we require the Lipschitz smoothness, which may be further relaxed in future work. In addition, D-SGDA can converge with the heterogeneous data distribution, whose stability and generalization are still unexplored in this work. Since our work focuses on the theoretical understanding of D-SGDA, it does not suffer from negative impacts.

## Acknowledgement

This work was supported in part by the National Natural Science Foundation of China under Grants 62225113.

The authors appreciate Erdun Gao for providing valuable suggestions on the experimental implementation. The authors also thank Tongtian Zhu for pointing out typos as well as useful discussions on the experimental results.

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

# Appendix

## Table of Contents

## A Decentralized Optimization

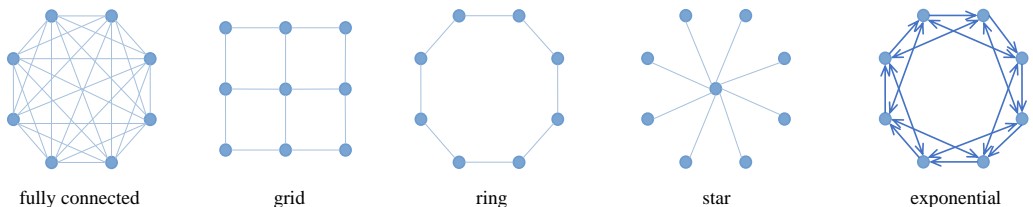

fully connected  grid  ring  star  exponential

Figure 3: Visualization of different typologies. Note that the lines without arrows mean undirected, such as the fully connected, the grid, the ring, and the star graph. While lines with arrows denote the direction, which shows how information flows from one node to another. Our study objective is undirected graph in this paper, with the exponential graph an illustrative example of directed graph.

## B Notation

1. $x \in \mathbb{R}^{d_x}$ and $y \in \mathbb{R}^{d_y}$, we use $\begin{pmatrix} x \\ y \end{pmatrix} \in \mathbb{R}^{d_x + d_y}$ to denote the concatenation.

2. $\boldsymbol{X}^t \triangleq (\boldsymbol{x}_1^t, ..., \boldsymbol{x}_m^t)^T \in \mathbb{R}^{m \times d_{\boldsymbol{x}}}$; $\boldsymbol{Y}^t \triangleq (\boldsymbol{y}_1^t, ..., \boldsymbol{y}_m^t)^T \in \mathbb{R}^{m \times d_{\boldsymbol{y}}}$; and we use $[\boldsymbol{X}, \boldsymbol{Y}]^t \in \mathbb{R}^{m \times (d_{\boldsymbol{x}} + d_{\boldsymbol{y}})}$ to denote the concatenation $\left[ \begin{pmatrix} \boldsymbol{x}_1^t \\ \boldsymbol{y}_1^t \end{pmatrix}, \begin{pmatrix} \boldsymbol{x}_2^t \\ \boldsymbol{y}_2^t \end{pmatrix}, ..., \begin{pmatrix} \boldsymbol{x}_m^t \\ \boldsymbol{y}_m^t \end{pmatrix} \right]^T$.

3. We use $\boldsymbol{\xi}^t$ to denote the samples collected in the $t$-th iteration, i.e., $\{\xi_{1, j_t(1)}, ..., \xi_{m, j_t(m)}\}$.

4. We denote the gradients in the $t$-th iteration as follows:

$$\nabla_{\boldsymbol{x}} \boldsymbol{f}(\boldsymbol{X}^t, \boldsymbol{Y}^t; \boldsymbol{\xi}^t) \triangleq \left( \nabla_{\boldsymbol{x}} f_1(\boldsymbol{x}_1^t, \boldsymbol{y}_1^t; \xi_{1, j_t(1)}), ..., \nabla_{\boldsymbol{x}} f_m(\boldsymbol{x}_m^t, \boldsymbol{y}_m^t; \xi_{m, j_t(m)}) \right)^T \in \mathbb{R}^{m \times d_{\boldsymbol{x}}}$$

$$\nabla_{\boldsymbol{y}} \boldsymbol{f}(\boldsymbol{X}^t, \boldsymbol{Y}^t; \boldsymbol{\xi}^t) \triangleq \left( \nabla_{\boldsymbol{y}} f_1(\boldsymbol{x}_1^t, \boldsymbol{y}_1^t; \xi_{1, j_t(1)}), ..., \nabla_{\boldsymbol{y}} f_m(\boldsymbol{x}_m^t, \boldsymbol{y}_m^t; \xi_{m, j_t(m)}) \right)^T \in \mathbb{R}^{m \times d_{\boldsymbol{y}}}$$

$$\begin{pmatrix} \nabla_{\boldsymbol{x}} \boldsymbol{f}(\boldsymbol{X}^t, \boldsymbol{Y}^t; \boldsymbol{\xi}^t) \\ \nabla_{\boldsymbol{y}} \boldsymbol{f}(\boldsymbol{X}^t, \boldsymbol{Y}^t; \boldsymbol{\xi}^t) \end{pmatrix} \triangleq \left[ \begin{pmatrix} \nabla_{\boldsymbol{x}} f_1(\boldsymbol{x}_1^t, \boldsymbol{y}_1^t; \xi_{1, j_t(1)}) \\ \nabla_{\boldsymbol{y}} f_1(\boldsymbol{x}_1^t, \boldsymbol{y}_1^t; \xi_{1, j_t(1)}) \end{pmatrix}, ..., \begin{pmatrix} \nabla_{\boldsymbol{x}} f_m(\boldsymbol{x}_m^t, \boldsymbol{y}_m^t; \xi_{m, j_t(m)}) \\ \nabla_{\boldsymbol{y}} f_m(\boldsymbol{x}_m^t, \boldsymbol{y}_m^t; \xi_{m, j_t(m)}) \end{pmatrix} \right]^T \in \mathbb{R}^{m \times (d_{\boldsymbol{x}} + d_{\boldsymbol{y}})}.$$

5. We use $\mathbf{P} \in \mathbb{R}^{m \times m}$ to denote the matrix whose elements are all $\frac{1}{m}$, $\mathbb{I}$ to denote the identity matrix, and $\mathbb{1}_m \in \mathbb{R}^m$ to denote the vector with all elements equal to 1.

## C Important Lemmas

In this section, we provide some important lemmas as fundamentals of the following proof.

**Lemma 1.** *We define $G_{g, \eta}$ as*

$$G_{g, \eta} \begin{pmatrix} \boldsymbol{x} \\ \boldsymbol{y} \end{pmatrix} = \begin{pmatrix} P_{\mathcal{X}}(\boldsymbol{x} - \eta_{\boldsymbol{x}} \nabla_{\boldsymbol{x}} g(\boldsymbol{x}, \boldsymbol{y})) \\ P_{\mathcal{Y}}(\boldsymbol{y} + \eta_{\boldsymbol{y}} \nabla_{\boldsymbol{y}} g(\boldsymbol{x}, \boldsymbol{y})) \end{pmatrix}$$

*with $\eta^{max} \triangleq \max\{\eta_{\boldsymbol{x}}, \eta_{\boldsymbol{y}}\}$ and $\eta^{min} \triangleq \min\{\eta_{\boldsymbol{x}}, \eta_{\boldsymbol{y}}\}$. Under the assumption that $g$ is $L$-Lipschitz smooth, then we have:*

a. *$G_{g, \eta}$ is $(1 + \eta^{max} L)$-expansive.*

b. *When $g$ is $\mu_{\boldsymbol{x}}$-strongly-convex w.r.t. $\boldsymbol{x}$ and $\mu_{\boldsymbol{y}}$-strongly-concave w.r.t. $\boldsymbol{y}$, letting $\frac{L + \mu}{2}(\eta^{max})^2 \leq \eta^{min} \leq \frac{L + \mu}{2} \frac{1}{L\mu}$ where $\mu \triangleq \min\{\mu_{\boldsymbol{x}}, \mu_{\boldsymbol{y}}\}$, then $G_{g, \eta}$ is $(1 - \eta^{min} \frac{L\mu}{L + \mu})$-expansive.*

*Proof.* For case a. where we do not require strong convexity or strong concavity, we have:

$$\left\| G_{g, \eta} \begin{pmatrix} \boldsymbol{x} \\ \boldsymbol{y} \end{pmatrix} - G_{g, \eta} \begin{pmatrix} \boldsymbol{x}' \\ \boldsymbol{y}' \end{pmatrix} \right\| \leq \left\| \begin{pmatrix} \boldsymbol{x} - \boldsymbol{x}' - \eta_{\boldsymbol{x}} (\nabla_{\boldsymbol{x}} g(\boldsymbol{x}, \boldsymbol{y}) - \nabla_{\boldsymbol{x}} g(\boldsymbol{x}', \boldsymbol{y}')) \\ \boldsymbol{y} - \boldsymbol{y}' - \eta_{\boldsymbol{y}} (\nabla_{\boldsymbol{y}} g(\boldsymbol{x}', \boldsymbol{y}') - \nabla_{\boldsymbol{y}} g(\boldsymbol{x}, \boldsymbol{y})) \end{pmatrix} \right\|$$

$$= \left\| \begin{pmatrix} \boldsymbol{x} - \boldsymbol{x}' \\ \boldsymbol{y} - \boldsymbol{y}' \end{pmatrix} \right\| + \left\| \begin{pmatrix} \eta_{\boldsymbol{x}} (\nabla_{\boldsymbol{x}} g(\boldsymbol{x}, \boldsymbol{y}) - \nabla_{\boldsymbol{x}} g(\boldsymbol{x}', \boldsymbol{y}')) \\ \eta_{\boldsymbol{y}} (\nabla_{\boldsymbol{y}} g(\boldsymbol{x}', \boldsymbol{y}') - \nabla_{\boldsymbol{y}} g(\boldsymbol{x}, \boldsymbol{y})) \end{pmatrix} \right\|$$

$$\leq (1 + \eta^{max} L) \left\| \begin{pmatrix} \boldsymbol{x} - \boldsymbol{x}' \\ \boldsymbol{y} - \boldsymbol{y}' \end{pmatrix} \right\|$$

When function $g$ is further $\mu_{\boldsymbol{x}}$-strongly convex and $\mu_{\boldsymbol{y}}$-strongly concave in case b., thus we have:

$$\left\| G_{g, \eta} \begin{pmatrix} \boldsymbol{x} \\ \boldsymbol{y} \end{pmatrix} - G_{g, \eta} \begin{pmatrix} \boldsymbol{x}' \\ \boldsymbol{y}' \end{pmatrix} \right\|^2$$

$$= \left\| \begin{pmatrix} \boldsymbol{x} - \boldsymbol{x}' - \eta_{\boldsymbol{x}} (\nabla_{\boldsymbol{x}} g(\boldsymbol{x}, \boldsymbol{y}) - \nabla_{\boldsymbol{x}} g(\boldsymbol{x}', \boldsymbol{y}')) \\ \boldsymbol{y} - \boldsymbol{y}' - \eta_{\boldsymbol{y}} (\nabla_{\boldsymbol{y}} g(\boldsymbol{x}', \boldsymbol{y}') - \nabla_{\boldsymbol{y}} g(\boldsymbol{x}, \boldsymbol{y})) \end{pmatrix} \right\|^2$$

$$= \| \boldsymbol{x} - \boldsymbol{x}' - \eta_{\boldsymbol{x}} (\nabla_{\boldsymbol{x}} g(\boldsymbol{x}, \boldsymbol{y}) - \nabla_{\boldsymbol{x}} g(\boldsymbol{x}', \boldsymbol{y}')) \|^2 + \| \boldsymbol{y} - \boldsymbol{y}' - \eta_{\boldsymbol{y}} (\nabla_{\boldsymbol{y}} g(\boldsymbol{x}', \boldsymbol{y}') - \nabla_{\boldsymbol{y}} g(\boldsymbol{x}, \boldsymbol{y})) \|^2$$

$$= \| \boldsymbol{x} - \boldsymbol{x}' \|^2 + \eta_{\boldsymbol{x}}^2 \| \nabla_{\boldsymbol{x}} g(\boldsymbol{x}, \boldsymbol{y}) - \nabla_{\boldsymbol{x}} g(\boldsymbol{x}', \boldsymbol{y}') \|^2 - 2\eta_{\boldsymbol{x}} \langle \boldsymbol{x} - \boldsymbol{x}', \nabla_{\boldsymbol{x}} g(\boldsymbol{x}, \boldsymbol{y}) - \nabla_{\boldsymbol{x}} g(\boldsymbol{x}', \boldsymbol{y}') \rangle$$

$$+ \| \boldsymbol{y} - \boldsymbol{y}' \|^2 + \eta_{\boldsymbol{y}}^2 \| \nabla_{\boldsymbol{y}} g(\boldsymbol{x}', \boldsymbol{y}') - \nabla_{\boldsymbol{y}} g(\boldsymbol{x}, \boldsymbol{y}) \|^2 - 2\eta_{\boldsymbol{y}} \langle \boldsymbol{y} - \boldsymbol{y}', \nabla_{\boldsymbol{y}} g(\boldsymbol{x}', \boldsymbol{y}') - \nabla_{\boldsymbol{y}} g(\boldsymbol{x}, \boldsymbol{y}) \rangle \tag{4}$$

Recalling the co-coercivity of a $L$-Lipschitz smooth and $\mu$-strongly convex function $f(\boldsymbol{x})$ that [13]:

$$\langle \nabla f(\boldsymbol{x}) - \nabla f(\boldsymbol{y}), \boldsymbol{x} - \boldsymbol{y} \rangle \geq \frac{1}{L + \mu} \| \nabla f(\boldsymbol{x}) - \nabla f(\boldsymbol{y}) \|^2 + \frac{L\mu}{L + \mu} \| \boldsymbol{x} - \boldsymbol{y} \|^2$$

So we can get:

$$\langle \boldsymbol{x} - \boldsymbol{x}', \nabla_{\boldsymbol{x}} g(\boldsymbol{x}, \boldsymbol{y}) - \nabla_{\boldsymbol{x}} g(\boldsymbol{x}', \boldsymbol{y}') \rangle + \langle \boldsymbol{y} - \boldsymbol{y}', \nabla_{\boldsymbol{y}} g(\boldsymbol{x}', \boldsymbol{y}') - \nabla_{\boldsymbol{y}} g(\boldsymbol{x}, \boldsymbol{y}) \rangle$$

$$= \left\langle \left( \begin{array}{c} \nabla_{\boldsymbol{x}} g(\boldsymbol{x}, \boldsymbol{y}) - \nabla_{\boldsymbol{x}} g(\boldsymbol{x}', \boldsymbol{y}') \\ \nabla_{\boldsymbol{y}} g(\boldsymbol{x}', \boldsymbol{y}') - \nabla_{\boldsymbol{y}} g(\boldsymbol{x}, \boldsymbol{y}) \end{array} \right), \left( \begin{array}{c} \boldsymbol{x} - \boldsymbol{x}' \\ \boldsymbol{y} - \boldsymbol{y}' \end{array} \right) \right\rangle$$

$$\geq \frac{1}{L + \mu} \left\| \left( \begin{array}{c} \nabla_{\boldsymbol{x}} g(\boldsymbol{x}, \boldsymbol{y}) - \nabla_{\boldsymbol{x}} g(\boldsymbol{x}', \boldsymbol{y}') \\ \nabla_{\boldsymbol{y}} g(\boldsymbol{x}', \boldsymbol{y}') - \nabla_{\boldsymbol{y}} g(\boldsymbol{x}, \boldsymbol{y}) \end{array} \right) \right\|^2 + \frac{L\mu}{L + \mu} \left\| \left( \begin{array}{c} \boldsymbol{x} - \boldsymbol{x}' \\ \boldsymbol{y} - \boldsymbol{y}' \end{array} \right) \right\|^2 \quad (5)$$

$$= \frac{1}{L + \mu} \| \nabla_{\boldsymbol{x}} g(\boldsymbol{x}, \boldsymbol{y}) - \nabla_{\boldsymbol{x}} g(\boldsymbol{x}', \boldsymbol{y}') \|^2 + \frac{1}{L + \mu} \| \nabla_{\boldsymbol{y}} g(\boldsymbol{x}', \boldsymbol{y}') - \nabla_{\boldsymbol{y}} g(\boldsymbol{x}, \boldsymbol{y}) \|^2$$

$$+ \frac{L\mu}{L + \mu} \| \boldsymbol{x} - \boldsymbol{x}' \|^2 + \frac{L\mu}{L + \mu} \| \boldsymbol{y} - \boldsymbol{y}' \|^2$$

Combining above inequalities (4) and (5), we can get:

$$\left\| G_{g,\eta} \left( \begin{array}{c} \boldsymbol{x} \\ \boldsymbol{y} \end{array} \right) - G_{g,\eta} \left( \begin{array}{c} \boldsymbol{x}' \\ \boldsymbol{y}' \end{array} \right) \right\| \leq (1 - \eta^{min} \frac{L\mu}{L + \mu}) \left\| \left( \begin{array}{c} \boldsymbol{x} - \boldsymbol{x}' \\ \boldsymbol{y} - \boldsymbol{y}' \end{array} \right) \right\|$$

when $\frac{L+\mu}{2}(\eta^{max})^2 \leq \eta^{min} \leq \frac{L+\mu}{2} \frac{1}{L\mu}$ satisfies. $\qquad \square$

**Lemma 2.** *Letting* $\mathbf{P} \in \mathbb{R}^{m \times m}$ *denote the matrix whose elements are all* $\frac{1}{m}$*, following the update rule of D-SGDA (see Algorithm 1), we have:*

$$\left\| (\mathbb{I} - \mathbf{P}) [\boldsymbol{X}, \boldsymbol{Y}]^t \right\| \leq 2\sqrt{m} G \sum_{s=0}^{t-1} \eta_s^{max} \lambda^{t-1-s}$$

*where* $\eta_s^{max} = \max\{\eta_{\boldsymbol{x},s}, \eta_{\boldsymbol{y},s}\}$.

*Proof.* The Lipschitz continuity (see Assumption 1) implies that for any $\boldsymbol{x} \in \mathcal{X}$ and $\boldsymbol{y} \in \mathcal{Y}$ on a given sample $\xi$, the gradient value is bounded by $G$, i.e., $\left\| \left( \begin{array}{c} \eta_{\boldsymbol{x},t} \nabla_{\boldsymbol{x}} \boldsymbol{f}(\boldsymbol{X}^t, \boldsymbol{Y}^t; \boldsymbol{\xi}^t) \\ -\eta_{\boldsymbol{y},t} \nabla_{\boldsymbol{y}} \boldsymbol{f}(\boldsymbol{X}^t, \boldsymbol{Y}^t; \boldsymbol{\xi}^t) \end{array} \right) \right\| \leq$ $\max\{\eta_{\boldsymbol{x},t}, \eta_{\boldsymbol{y},t}\} \sqrt{m} G$. Following the proof of Lemma 8 in [32], we can get the result, which can be expressed as $\left[ \sum_{i=1}^{m} \| \boldsymbol{x}_i^t - \boldsymbol{x}^t \|^2 + \| \boldsymbol{y}_i^t - \boldsymbol{y}^t \|^2 \right]^{1/2} \leq 2\sqrt{m} G \sum_{s=0}^{t-1} \eta_s^{max} \lambda^{t-1-s}$ $\qquad \square$

**Lemma 3.** *When* $0 < \lambda < 1$,

$$\sum_{j=0}^{t-1} \frac{\lambda^{t-1-j}}{(j+1)^k} \leq \frac{C_\lambda}{t^k}, k > 0; \qquad C_\lambda \triangleq \frac{(k/e)^k}{\lambda(\ln \frac{1}{\lambda})^k} + \frac{2e^{-1}}{\lambda \ln \frac{1}{\lambda}} + \frac{2^k}{\lambda \ln \frac{1}{\lambda}}$$

*Proof.*

$$\sum_{j=0}^{t-1} \frac{\lambda^{t-1-j}}{(j+1)^k} = \lambda^{t-1} + \sum_{j=1}^{t-1} \frac{\lambda^{t-1-j}}{(j+1)^k} \leq \lambda^{t-1} + \int_1^t \frac{\lambda^{t-1-x}}{x^k} dx = \lambda^{t-1} + \lambda^{t-1} \int_1^t \frac{\lambda^{-x}}{x^k} dx$$

whereas for the integral we have:

$$\int_1^t \frac{\lambda^{-x}}{x^k} dx = \int_1^{\frac{t}{2}} \frac{\lambda^{-x}}{x^k} dx + \int_{\frac{t}{2}}^t \frac{\lambda^{-x}}{x^k} dx \leq \lambda^{-\frac{t}{2}} \int_1^{\frac{t}{2}} \frac{1}{x^k} dx + \frac{1}{(\frac{t}{2})^k} \int_{\frac{t}{2}}^t \lambda^{-x} dx$$

$$\int_1^{\frac{t}{2}} \frac{1}{x^k} dx \leq \left\{ \begin{array}{ll} ln\frac{t}{2} & k = 1 \\ \frac{(\frac{t}{2})^{1-k}}{1-k} & 0 < k < 1 \end{array} \right.$$

Therefore we have:

$$\sum_{j=0}^{t-1} \frac{\lambda^{t-1-j}}{(j+1)^k} \leq \left\{ \begin{array}{ll} \lambda^{t-1} + \lambda^{\frac{t}{2}-1} ln\frac{t}{2} + \frac{1}{t^k} \frac{2^k}{\lambda ln \frac{1}{\lambda}} & k = 1 \\ \lambda^{t-1} + \lambda^{\frac{t}{2}-1} t^{1-k} + \frac{1}{t^k} \frac{2^k}{\lambda ln \frac{1}{\lambda}} & 0 < k < 1 \end{array} \right.$$

When $0 < k < 1$, $\sum_{j=0}^{t-1} \frac{\lambda^{t-1-j}}{(j+1)^k} \leq \frac{1}{t^k}\left(\lambda^{t-1}t^k + \lambda^{\frac{t}{2}-1}t + \frac{2^k}{\lambda \ln \frac{1}{\lambda}}\right)$, where $\lambda^{t-1}t^k \leq \frac{(k/e)^k}{\lambda(\ln \frac{1}{\lambda})^k}$, $\lambda^{\frac{t}{2}-1}t \leq \frac{2e^{-1}}{\lambda \ln \frac{1}{\lambda}}$. So we can define $C_\lambda = \frac{(k/e)^k}{\lambda(\ln \frac{1}{\lambda})^k} + \frac{2e^{-1}}{\lambda \ln \frac{1}{\lambda}} + \frac{2^k}{\lambda \ln \frac{1}{\lambda}}$ that $\sum_{j=0}^{t-1} \frac{\lambda^{t-1-j}}{(j+1)^k} \leq \frac{C_\lambda}{t^k}$. And for the case $k = 1$ we can roughly consider the logarithm as constants so our discussion can be put together into the above category as $k = 1$. $\qquad\square$

**Remark 10.** Considering function $h(\lambda) = \frac{1}{\lambda \ln \frac{1}{\lambda}}$, it monotonically decrease in interval $(0, \frac{1}{e})$ and monotonically increase in interval $(\frac{1}{e}, 1)$ with minimal value of $e$. Analogously for the function $g(\lambda) = \frac{1}{\lambda(\ln \frac{1}{\lambda})^k}$, which monotonically decrease in interval $(0, \frac{1}{e^k})$ and monotonically increase in interval $(\frac{1}{e^k}, 1)$ with minimal value of $(\frac{e}{k})^k$. Therefore, when $k = 1$, $C_\lambda$ is bounded by $\mathcal{O}(\frac{1}{\lambda \ln \frac{1}{\lambda}})$; otherwise, when $\lambda \to 1$, $C_\lambda$ is bounded by $\mathcal{O}(\frac{1}{\lambda \ln \frac{1}{\lambda}})$; when $\lambda \to 0$, $C_\lambda$ is bounded by $\mathcal{O}(\frac{1}{\lambda(\ln \frac{1}{\lambda})^k})$.

# D   Proof of the Connection

In this section, we will first figure out a fundamental lemma that illustrates the structure of dataset in decentralized setting. Then we will provide proof for the connection between argument stability and primal-dual generalization gap.

**Lemma 4.** *Denoting dataset as $\mathcal{S} = \{\mathcal{S}_1, ..., \mathcal{S}_m\}$, and denoting each local samples stored in $\mathcal{S}_i$ as $\mathcal{S}_i = \{\xi_{i,l_i}\}_{l_i=1,...,n}$. Then we have the decomposition equation of the empirical function:*

$$F_{\mathcal{S}}(\boldsymbol{x}, \boldsymbol{y}) = \frac{1}{n^m} \sum_{l_1=1}^{n} ... \sum_{l_m=1}^{n} \frac{1}{m} \sum_{i=1}^{m} f_i(\boldsymbol{x}, \boldsymbol{y}; \xi_{i,l_i})$$

*Proof.*

$$\frac{1}{n^m} \sum_{l_1=1}^{n} ... \sum_{l_m=1}^{n} \frac{1}{m} \sum_{i=1}^{m} f_i(\boldsymbol{x}, \boldsymbol{y}; \xi_{i,l_i})$$

$$= \frac{1}{m} \sum_{i=1}^{m} \left( \frac{1}{n^{m-1}} \sum_{l_1=1}^{n} ... \sum_{l_{i-1}=1}^{n} \sum_{l_{i+1}=1}^{n} ... \sum_{l_m=1}^{n} \frac{1}{n} \sum_{l_i=1}^{n} f_i(\boldsymbol{x}, \boldsymbol{y}; \xi_{i,l_i}) \right)$$

$$= \frac{1}{m} \sum_{i=1}^{m} \left( \frac{1}{n^{m-1}} \sum_{l_1=1}^{n} ... \sum_{l_{i-1}=1}^{n} \sum_{l_{i+1}=1}^{n} ... \sum_{l_m=1}^{n} F_{\mathcal{S}_i}(\boldsymbol{x}, \boldsymbol{y}) \right)$$

$$= \frac{1}{m} \sum_{i=1}^{m} F_{\mathcal{S}_i}(\boldsymbol{x}, \boldsymbol{y})$$

$$= F_{\mathcal{S}}(\boldsymbol{x}, \boldsymbol{y})$$

$\qquad\square$

*Proof of Theorem 1.* We let $\mathcal{S} = \{\mathcal{S}_1, ..., \mathcal{S}_m\}$ where $\mathcal{S}_i = \{\xi_{i,l}\}_{1 \leq l \leq n}$ and $\mathcal{S}' = \{\mathcal{S}'_1, ..., \mathcal{S}'_m\}$ where $\mathcal{S}'_i = \{\xi'_{i,l}\}_{1 \leq l \leq n}$ be two different datasets while $\mathcal{S}_i$ and $\mathcal{S}'_i$ are drawn from the same distribution $\mathcal{D}_i$. Further we define $\mathcal{S}^{(l)} = \{\mathcal{S}_1^{(l_1)}, \mathcal{S}_2^{(l_2)}, ..., \mathcal{S}_m^{(l_m)}\}$ where $\boldsymbol{l} = (l_1, l_2, ..., l_m)$ and $\mathcal{S}_i^{(l_i)} = \{\xi_{i,1}, ..., \xi_{i,l_i-1}, \xi'_{i,l_i}, \xi_{i,l_i+1}, ..., \xi_{i,n}\}$. So we can say that there is at most one different sample in each local dataset between $\mathcal{S}$ and $\mathcal{S}^{(l)}$. For function $F : \mathcal{X} \times \mathcal{Y} \mapsto \mathbb{R}$, we denote $\boldsymbol{y}^*(\boldsymbol{x}) = arg \max_{\boldsymbol{y} \in \mathcal{Y}} F(\boldsymbol{x}, \boldsymbol{y})$.

For case a. the weak primal-dual generalization gap $\epsilon_{gen}^w(\mathcal{A}_{\boldsymbol{x}}(\mathcal{S}), \mathcal{A}_{\boldsymbol{y}}(\mathcal{S})) = \Delta^w(\mathcal{A}_{\boldsymbol{x}}(\mathcal{S}), \mathcal{A}_{\boldsymbol{y}}(\mathcal{S})) - \Delta_{\mathcal{S}}^w(\mathcal{A}_{\boldsymbol{x}}(\mathcal{S}), \mathcal{A}_{\boldsymbol{y}}(\mathcal{S}))$. We first make some adjustments to the definition:

$$\epsilon_{gen}^w(\mathcal{A}_{\boldsymbol{x}}(\mathcal{S}), \mathcal{A}_{\boldsymbol{y}}(\mathcal{S}))$$

$$=(\sup_{\boldsymbol{y}'\in\mathcal{Y}}\mathbb{E}[F(\mathcal{A}_{\boldsymbol{x}}(\mathcal{S}),\boldsymbol{y}')]-\inf_{\boldsymbol{x}'\in\mathcal{X}}\mathbb{E}[F(\boldsymbol{x}',\mathcal{A}_{\boldsymbol{y}}(\mathcal{S}))])-(\sup_{\boldsymbol{y}'\in\mathcal{Y}}\mathbb{E}[F_{\mathcal{S}}(\mathcal{A}_{\boldsymbol{x}}(\mathcal{S}),\boldsymbol{y}')]-\inf_{\boldsymbol{x}'\in\mathcal{X}}\mathbb{E}[F_{\mathcal{S}}(\boldsymbol{x}',\mathcal{A}_{\boldsymbol{y}}(\mathcal{S}))])$$

$$=\sup_{\boldsymbol{y}'\in\mathcal{Y}}\mathbb{E}[F(\mathcal{A}_{\boldsymbol{x}}(\mathcal{S}),\boldsymbol{y}')]-\sup_{\boldsymbol{y}'\in\mathcal{Y}}\mathbb{E}[F_{\mathcal{S}}(\mathcal{A}_{\boldsymbol{x}}(\mathcal{S}),\boldsymbol{y}')]+\inf_{\boldsymbol{x}'\in\mathcal{X}}\mathbb{E}[F_{\mathcal{S}}(\boldsymbol{x}',\mathcal{A}_{\boldsymbol{y}}(\mathcal{S}))]-\inf_{\boldsymbol{x}'\in\mathcal{X}}\mathbb{E}[F(\boldsymbol{x}',\mathcal{A}_{\boldsymbol{y}}(\mathcal{S}))]$$

$$\leq\sup_{\boldsymbol{y}'\in\mathcal{Y}}(\mathbb{E}_{\mathcal{A},\mathcal{S}}[F(\mathcal{A}_{\boldsymbol{x}}(\mathcal{S}),\boldsymbol{y}')-F_{\mathcal{S}}(\mathcal{A}_{\boldsymbol{x}}(\mathcal{S}),\boldsymbol{y}')])+\sup_{\boldsymbol{x}'\in\mathcal{X}}(\mathbb{E}_{\mathcal{A},\mathcal{S}}[F_{\mathcal{S}}(\boldsymbol{x}',\mathcal{A}_{\boldsymbol{y}}(\mathcal{S}))-F(\boldsymbol{x}',\mathcal{A}_{\boldsymbol{y}}(\mathcal{S}))])$$

where the expectation can be detailed into $\mathbb{E}_{\mathcal{A},\mathcal{S}}$ in this situation.
For the first term, we have

$$\mathbb{E}_{\mathcal{A},\mathcal{S}}[F(\mathcal{A}_{\boldsymbol{x}}(\mathcal{S}),\boldsymbol{y}')-F_{\mathcal{S}}(\mathcal{A}_{\boldsymbol{x}}(\mathcal{S}),\boldsymbol{y}')]$$

$$=\frac{1}{n^m}\sum_{l_1=1}^{n}...\sum_{l_m=1}^{n}\mathbb{E}_{\mathcal{A},\mathcal{S},\mathcal{S}'}[F(\mathcal{A}_{\boldsymbol{x}}(\mathcal{S}^{(l)}),\boldsymbol{y}')]-\mathbb{E}_{\mathcal{A},\mathcal{S},\mathcal{S}'}[F_{\mathcal{S}}(\mathcal{A}_{\boldsymbol{x}}(\mathcal{S}),\boldsymbol{y}')]$$

$$=\frac{1}{n^m}\sum_{l_1=1}^{n}...\sum_{l_m=1}^{n}\mathbb{E}_{\mathcal{A},\mathcal{S},\mathcal{S}'}[\frac{1}{m}\sum_{i=1}^{m}f_i(\mathcal{A}_{\boldsymbol{x}}(\mathcal{S}^l),\boldsymbol{y}';\xi_{i,l_i})-\frac{1}{m}\sum_{i=1}^{m}f_i(\mathcal{A}_{\boldsymbol{x}}(\mathcal{S}),\boldsymbol{y}';\xi_{i,l_i})]$$

$$\leq\frac{1}{n^m}\sum_{l_1=1}^{n}...\sum_{l_m=1}^{n}\mathbb{E}_{\mathcal{A},\mathcal{S},\mathcal{S}'}[\frac{1}{m}\sum_{i=1}^{m}G\|\mathcal{A}_{\boldsymbol{x}}(\mathcal{S}^l)-\mathcal{A}_{\boldsymbol{x}}(\mathcal{S})\|]$$

$$=G\mathbb{E}_{\mathcal{A},\mathcal{S},\mathcal{S}'}\|\mathcal{A}_{\boldsymbol{x}}(\mathcal{S}^l)-\mathcal{A}_{\boldsymbol{x}}(\mathcal{S})\|$$

where the first equation is due to the symmetric distribution between $\mathcal{S}_i$ and $\mathcal{S}_i'$ and there are overall $n^m$ permutations of $\boldsymbol{l}$. And the second equation is due to the independence of sample $\xi_{i,l_i}$ from the training process upon dataset $\mathcal{S}^l$ and Lemma 4.
And we can analyze the second term in a similar way:

$$\mathbb{E}_{\mathcal{A},\mathcal{S}}[F_{\mathcal{S}}(\boldsymbol{x}',\mathcal{A}_{\boldsymbol{y}}(\mathcal{S}))-F(\boldsymbol{x}',\mathcal{A}_{\boldsymbol{y}}(\mathcal{S}))]\leq G\mathbb{E}_{\mathcal{A},\mathcal{S},\mathcal{S}'}\|\mathcal{A}_{\boldsymbol{y}}(\mathcal{S})-\mathcal{A}_{\boldsymbol{y}}(\mathcal{S}^l)\|$$

Therefore we can get the weak primal-dual generalization gap that:

$$\epsilon_{gen}^w(\mathcal{A}_{\boldsymbol{x}}(\mathcal{S}),\mathcal{A}_{\boldsymbol{y}}(\mathcal{S}))\leq\sqrt{2}G\sup_{\mathcal{S},\mathcal{S}'}\mathbb{E}_{\mathcal{A}}\left\|\begin{array}{c}\mathcal{A}_{\boldsymbol{x}}(\mathcal{S})-\mathcal{A}_{\boldsymbol{x}}(\mathcal{S}^l)\\\mathcal{A}_{\boldsymbol{y}}(\mathcal{S})-\mathcal{A}_{\boldsymbol{y}}(\mathcal{S}^l)\end{array}\right\|\leq\sqrt{2}G\epsilon$$

For case b. the strong primal-dual generalization gap $\epsilon_{gen}^s(\mathcal{A}_{\boldsymbol{x}}(\mathcal{S}),\mathcal{A}_{\boldsymbol{y}}(\mathcal{S}))=\Delta^s(\mathcal{A}_{\boldsymbol{x}}(\mathcal{S}),\mathcal{A}_{\boldsymbol{y}}(\mathcal{S}))-\Delta_{\mathcal{S}}^s(\mathcal{A}_{\boldsymbol{x}}(\mathcal{S}),\mathcal{A}_{\boldsymbol{y}}(\mathcal{S}))$.
Observing the structure that:

$$\epsilon_{gen}^s(\mathcal{A}_{\boldsymbol{x}}(\mathcal{S}),\mathcal{A}_{\boldsymbol{y}}(\mathcal{S}))$$

$$=\mathbb{E}[\sup_{\boldsymbol{y}'\in\mathcal{Y}}F(\mathcal{A}_{\boldsymbol{x}}(\mathcal{S}),\boldsymbol{y}')-\inf_{\boldsymbol{x}'\in\mathcal{X}}F(\boldsymbol{x}',\mathcal{A}_{\boldsymbol{y}}(\mathcal{S}))]-\mathbb{E}[\sup_{\boldsymbol{y}'\in\mathcal{Y}}F_{\mathcal{S}}(\mathcal{A}_{\boldsymbol{x}}(\mathcal{S}),\boldsymbol{y}')-\inf_{\boldsymbol{x}'\in\mathcal{X}}F_{\mathcal{S}}(\boldsymbol{x}',\mathcal{A}_{\boldsymbol{y}}(\mathcal{S}))]$$

$$=\mathbb{E}_{\mathcal{A},\mathcal{S}}[\sup_{\boldsymbol{y}'\in\mathcal{Y}}F(\mathcal{A}_{\boldsymbol{x}}(\mathcal{S}),\boldsymbol{y}')-\sup_{\boldsymbol{y}'\in\mathcal{Y}}F_{\mathcal{S}}(\mathcal{A}_{\boldsymbol{x}}(\mathcal{S}),\boldsymbol{y}')]+\mathbb{E}_{\mathcal{A},\mathcal{S}}[\inf_{\boldsymbol{x}'\in\mathcal{X}}F_{\mathcal{S}}(\boldsymbol{x}',\mathcal{A}_{\boldsymbol{y}}(\mathcal{S}))-\inf_{\boldsymbol{x}'\in\mathcal{X}}F(\boldsymbol{x}',\mathcal{A}_{\boldsymbol{y}}(\mathcal{S}))]$$

(6)

where the expectation can be detailed into $\mathbb{E}_{\mathcal{S},\mathcal{A}}$ in this situation.

For the first term,

$$\mathbb{E}_{\mathcal{A},\mathcal{S}}[\sup_{\boldsymbol{y}'\in\mathcal{Y}}F(\mathcal{A}_{\boldsymbol{x}}(\mathcal{S}),\boldsymbol{y}')]\overset{(i)}{=}\frac{1}{n^m}\sum_{l_1=1}^{n}\sum_{l_2=1}^{n}...\sum_{l_m=1}^{n}\mathbb{E}_{\mathcal{A},\mathcal{S},\mathcal{S}'}[\sup_{\boldsymbol{y}'\in\mathcal{Y}}F(\mathcal{A}_{\boldsymbol{x}}(\mathcal{S}^{(l)}),\boldsymbol{y}')]$$

$$\overset{(ii)}{=}\frac{1}{n^m}\sum_{l_1=1}^{n}\sum_{l_2=1}^{n}...\sum_{l_m=1}^{n}\mathbb{E}_{\mathcal{A},\mathcal{S},\mathcal{S}'}[F(\mathcal{A}_{\boldsymbol{x}}(\mathcal{S}^{(l)}),\boldsymbol{y}_{\mathcal{S}^{(l)}}^*)]$$

$$\overset{(iii)}{=}\frac{1}{n^m}\sum_{l_1=1}^{n}\sum_{l_2=1}^{n}...\sum_{l_m=1}^{n}\mathbb{E}_{\mathcal{A},\mathcal{S},\mathcal{S}'}[\frac{1}{m}\sum_{i=1}^{m}f_i(\mathcal{A}_{\boldsymbol{x}}(\mathcal{S}^{(l)}),\boldsymbol{y}_{\mathcal{S}^{(l)}}^*;\xi_{i,l_i})]$$

Since $\mathcal{S}_i$ and $\mathcal{S}_i'$ are drawn from the same distribution, and there are $n^m$ permutations of $\boldsymbol{l}=\{l_1,...,l_m\}$ so we can get equality $(i)$. In equality $(ii)$, we denote $\boldsymbol{y}_{\mathcal{S}^{(l)}}^*$ as $arg\max_{y\in\mathcal{Y}}F(\mathcal{A}_{\boldsymbol{x}}(\mathcal{S}),\boldsymbol{y})$.

While for equality $(iii)$, $\xi_{i,l_i}$ is independent from the training process under dataset $\mathcal{S}^l$ for each local agent respectively.

Using the property of Lipschitz continuous (see Assumption 1), we can further get:

$$f_i(\mathcal{A}_{\boldsymbol{x}}(\mathcal{S}^{(l)}), \boldsymbol{y}^*_{\mathcal{S}^{(l)}}; \xi_{i,l_i}) - f_i(\mathcal{A}_{\boldsymbol{x}}(\mathcal{S}), \boldsymbol{y}^*_{\mathcal{S}}; \xi_{i,l_i}) \leq G \left\| \left( \begin{array}{c} \mathcal{A}_{\boldsymbol{x}}(\mathcal{S}^{(l)}) - \mathcal{A}_{\boldsymbol{x}}(\mathcal{S}) \\ \boldsymbol{y}^*_{\mathcal{S}^{(l)}} - \boldsymbol{y}^*_{\mathcal{S}} \end{array} \right) \right\|$$

$$\overset{(a)}{\leq} G\sqrt{1 + \frac{L^2}{\mu_{\boldsymbol{y}}^2}} \|\mathcal{A}_{\boldsymbol{x}}(\mathcal{S}^l) - \mathcal{A}_{\boldsymbol{x}}(\mathcal{S})\|$$

where inequality $(a)$ is a conclusion of Lemma 4.3 in [22] that $\|\boldsymbol{y}^*_{\mathcal{S}^{(l)}} - \boldsymbol{y}^*_{\mathcal{S}}\| \leq \frac{L}{\mu_{\boldsymbol{y}}} \|\mathcal{A}_{\boldsymbol{x}}(\mathcal{S}^{(l)}) - \mathcal{A}_{\boldsymbol{x}}(\mathcal{S})\|$.

Combining above two inequalities, we can get:

$$\mathbb{E}_{\mathcal{A},\mathcal{S}}[\sup_{\boldsymbol{y}' \in \mathcal{Y}} F(\mathcal{A}_{\boldsymbol{x}}(\mathcal{S}), \boldsymbol{y}')]$$

$$\leq \frac{1}{n^m} \sum_{l_1=1}^{n} \sum_{l_2=1}^{n} ... \sum_{l_m=1}^{n} \mathbb{E}_{\mathcal{A},\mathcal{S},\mathcal{S}'} \left[ \frac{1}{m} \sum_{i=1}^{m} \left( f_i(\mathcal{A}_{\boldsymbol{x}}(\mathcal{S}), \boldsymbol{y}^*_{\mathcal{S}}; \xi_{i,l_i}) + G\sqrt{1 + \frac{L^2}{\mu_{\boldsymbol{y}}^2}} \|\mathcal{A}_{\boldsymbol{x}}(\mathcal{S}^l) - \mathcal{A}_{\boldsymbol{x}}(\mathcal{S})\| \right) \right]$$

$$= \mathbb{E}_{\mathcal{A},\mathcal{S}} \left[ F_{\mathcal{S}}(\mathcal{A}_{\boldsymbol{x}}(\mathcal{S}), \boldsymbol{y}^*_{\mathcal{S}}) \right] + G\sqrt{1 + \frac{L^2}{\mu_{\boldsymbol{y}}^2}} \mathbb{E}_{\mathcal{A}}[\sup_{\mathcal{S},\mathcal{S}'} \|\mathcal{A}_{\boldsymbol{x}}(\mathcal{S}^l) - \mathcal{A}_{\boldsymbol{x}}(\mathcal{S})\|]$$

$$\leq \mathbb{E}_{\mathcal{A},\mathcal{S}} \left[ \sup_{\boldsymbol{y}' \in \mathcal{Y}} F_{\mathcal{S}}(\mathcal{A}_{\boldsymbol{x}}(\mathcal{S}), \boldsymbol{y}') \right] + G\sqrt{1 + \frac{L^2}{\mu_{\boldsymbol{y}}^2}} \mathbb{E}_{\mathcal{A}}[\sup_{\mathcal{S},\mathcal{S}'} \|\mathcal{A}_{\boldsymbol{x}}(\mathcal{S}^l) - \mathcal{A}_{\boldsymbol{x}}(\mathcal{S})\|]$$

where we get the last but one inequality from Lemma 4.

We can do a similar operation on the second counterpart since $f_i$ is also $\mu_{\boldsymbol{x}}$-strongly convex on $\boldsymbol{x}$.

$$\mathbb{E}_{\mathcal{A},\mathcal{S}}[\inf_{\boldsymbol{x}' \in \mathcal{X}} F_{\mathcal{S}}(\boldsymbol{x}', \mathcal{A}_{\boldsymbol{y}}(\mathcal{S})) - \inf_{\boldsymbol{x}' \in \mathcal{X}} F(\boldsymbol{x}', \mathcal{A}_{\boldsymbol{y}}(\mathcal{S}))] \leq G\sqrt{1 + \frac{L^2}{\mu_{\boldsymbol{x}}^2}} \mathbb{E}_{\mathcal{A}}[\sup_{\mathcal{S},\mathcal{S}'} \|\mathcal{A}_{\boldsymbol{y}}(\mathcal{S}^l) - \mathcal{A}_{\boldsymbol{y}}(\mathcal{S})\|]$$

So the overall strong primal-dual generalization gap satisfies:

$$\epsilon^s_{gen}(\mathcal{A}_{\boldsymbol{x}}(\mathcal{S}), \mathcal{A}_{\boldsymbol{y}}(\mathcal{S}))$$

$$\leq G\sqrt{1 + \frac{L^2}{\mu^2}} \sqrt{2} \mathbb{E}_{\mathcal{A}} \left[ \sup_{\mathcal{S},\mathcal{S}'} \left\| \begin{array}{c} \mathcal{A}_{\boldsymbol{x}}(\mathcal{S}) - \mathcal{A}_{\boldsymbol{x}}(\mathcal{S}^l) \\ \mathcal{A}_{\boldsymbol{y}}(\mathcal{S}) - \mathcal{A}_{\boldsymbol{y}}(\mathcal{S}^l) \end{array} \right\| \right]$$

$$\leq G\sqrt{2 + \frac{2L^2}{\mu^2}} \epsilon$$

where $\mu = \min\{\mu_{\boldsymbol{x}}, \mu_{\boldsymbol{y}}\}$. $\qquad\qquad\square$

# E Proof in the Strongly-Convex-Strongly-Concave Case

In this section, we will prove the stability and generalization gap of our D-SGDA in the case of $\mu_{\boldsymbol{x}}$-strongly convex and $\mu_{\boldsymbol{y}}$-strongly concave.

## E.1 Proof of Stability

*Proof of Theorem 2.* We use $(\boldsymbol{x}^t, \boldsymbol{y}^t)$ and $(\dot{\boldsymbol{x}}^t, \dot{\boldsymbol{y}}^t)$ to represent the output in the $t$-th iteration when applying D-SGDA on any arbitrary neighbouring dataset $\mathcal{S}$ and $\mathcal{S}'$ respectively. Since the D-SGDA algorithm is symmetric [4] with respect to the dataset $\mathcal{S} = \{\mathcal{S}_1, ..., \mathcal{S}_m\}$, we can assume the different samples appear in the last location in each $\mathcal{S}_i$ without loss of generality, i.e., $\mathcal{S}' = \{\mathcal{S}'_1, ..., \mathcal{S}'_{m-1}, \mathcal{S}'_m\}$ where $\mathcal{S}'_i = \{\xi_{i,1}, \xi_{i,2}, ..., \xi_{i,n-1}, \xi'_{i,n}\}$ differs from $\mathcal{S}_i = \{\xi_{i,1}, ..., \xi_{i,n}\}$ in the $n$-th data.

First concentrating on the iteration, we have:

$$
\begin{aligned}
\begin{pmatrix} \boldsymbol{x}^{t+1} \\ \boldsymbol{y}^{t+1} \end{pmatrix} &= \left( [\boldsymbol{X}, \boldsymbol{Y}]^{t+1} \right)^T \frac{\mathbb{1}_m}{m} \\
&= P_{(\mathcal{X}, \mathcal{Y})^m} \left[ \left( \boldsymbol{W}[\boldsymbol{X}, \boldsymbol{Y}]^t - \begin{pmatrix} \eta_{\boldsymbol{x},t} \nabla_{\boldsymbol{x}} \boldsymbol{f}(\boldsymbol{X}^t, \boldsymbol{Y}^t; \boldsymbol{\xi}^t) \\ -\eta_{\boldsymbol{y},t} \nabla_{\boldsymbol{y}} \boldsymbol{f}(\boldsymbol{X}^t, \boldsymbol{Y}^t; \boldsymbol{\xi}^t) \end{pmatrix} \right)^T \right] \frac{\mathbb{1}_m}{m} \\
&= P_{(\mathcal{X}, \mathcal{Y})} \left[ \left( \boldsymbol{W}[\boldsymbol{X}, \boldsymbol{Y}]^t - \begin{pmatrix} \eta_{\boldsymbol{x},t} \nabla_{\boldsymbol{x}} \boldsymbol{f}(\boldsymbol{X}^t, \boldsymbol{Y}^t; \boldsymbol{\xi}^t) \\ -\eta_{\boldsymbol{y},t} \nabla_{\boldsymbol{y}} \boldsymbol{f}(\boldsymbol{X}^t, \boldsymbol{Y}^t; \boldsymbol{\xi}^t) \end{pmatrix} \right)^T \frac{\mathbb{1}_m}{m} \right] \\
&= P_{(\mathcal{X}, \mathcal{Y})} \left[ ([\boldsymbol{X}, \boldsymbol{Y}]^t)^T \boldsymbol{W}^T \frac{\mathbb{1}_m}{m} - \begin{pmatrix} \eta_{\boldsymbol{x},t} \nabla_{\boldsymbol{x}} \boldsymbol{f}(\boldsymbol{X}^t, \boldsymbol{Y}^t; \boldsymbol{\xi}^t) \\ -\eta_{\boldsymbol{y},t} \nabla_{\boldsymbol{y}} \boldsymbol{f}(\boldsymbol{X}^t, \boldsymbol{Y}^t; \boldsymbol{\xi}^t) \end{pmatrix}^T \frac{\mathbb{1}_m}{m} \right] \\
&= P_{(\mathcal{X}, \mathcal{Y})} \left[ \begin{pmatrix} \boldsymbol{x}^t \\ \boldsymbol{y}^t \end{pmatrix} - \begin{pmatrix} \eta_{\boldsymbol{x},t} \nabla_{\boldsymbol{x}} \boldsymbol{f}(\boldsymbol{X}^t, \boldsymbol{Y}^t; \boldsymbol{\xi}^t) \\ -\eta_{\boldsymbol{y},t} \nabla_{\boldsymbol{y}} \boldsymbol{f}(\boldsymbol{X}^t, \boldsymbol{Y}^t; \boldsymbol{\xi}^t) \end{pmatrix}^T \frac{\mathbb{1}_m}{m} \right] \\
&= P_{(\mathcal{X}, \mathcal{Y})} \left[ \begin{pmatrix} \boldsymbol{x}^t \\ \boldsymbol{y}^t \end{pmatrix} - \frac{1}{m} \sum_{i=1}^m \begin{pmatrix} \eta_{\boldsymbol{x},t} \nabla_{\boldsymbol{x}} f_i(\boldsymbol{x}_i^t, \boldsymbol{y}_i^t; \xi_{i, j_t(i)}) \\ -\eta_{\boldsymbol{y},t} \nabla_{\boldsymbol{y}} f_i(\boldsymbol{x}_i^t, \boldsymbol{y}_i^t; \xi_{i, j_t(i)}) \end{pmatrix} \right]
\end{aligned}
$$

where in the first equality we use the definition that $\boldsymbol{x}^t = \frac{1}{m} \sum_{i=1}^m \boldsymbol{x}_i^t, \boldsymbol{y}^t = \frac{1}{m} \sum_{i=1}^m \boldsymbol{y}_i^t$, and the last equality is due to the fact that $\boldsymbol{W} \frac{\mathbb{1}_m}{m} = \frac{\mathbb{1}_m}{m}$. We use the notation $P_{(\mathcal{X}, \mathcal{Y})} \left[ \begin{pmatrix} \boldsymbol{x} \\ \boldsymbol{y} \end{pmatrix} \right] \in \mathbb{R}^{d_{\boldsymbol{x}} + d_{\boldsymbol{y}}}$ as an abbreviation of $\begin{pmatrix} P_{\mathcal{X}}[\boldsymbol{x}] \\ P_{\mathcal{Y}}[\boldsymbol{y}] \end{pmatrix} \in \mathbb{R}^{d_{\boldsymbol{x}} + d_{\boldsymbol{y}}}$, and $P_{(\mathcal{X}, \mathcal{Y})^m}[[\boldsymbol{X}, \boldsymbol{Y}]^T]$ means every column should obey $P_{(\mathcal{X}, \mathcal{Y})} \left[ \begin{pmatrix} \boldsymbol{x} \\ \boldsymbol{y} \end{pmatrix} \right]$.

In the $t$-th iteration, there is a probability of $C_m^{m_0} (1 - \frac{1}{n})^{m - m_0} (\frac{1}{n})^{m_0}$ that there are $m - m_0$ agents not selecting the last different samples while the rest $m_0$ agents selecting exactly the last different samples. Without loss of generalization, we assume the first $m - m_0$ not selecting and the rest $m_0$ agents selecting the different samples respectively. So we can get:

$$
\begin{aligned}
&\begin{pmatrix} \boldsymbol{x}^{t+1} - \dot{\boldsymbol{x}}^{t+1} \\ \boldsymbol{y}^{t+1} - \dot{\boldsymbol{y}}^{t+1} \end{pmatrix} \\
&= P_{\mathcal{X}, \mathcal{Y}} \left[ \begin{pmatrix} \boldsymbol{x}^t - \dot{\boldsymbol{x}}^t \\ \boldsymbol{y}^t - \dot{\boldsymbol{y}}^t \end{pmatrix} - \frac{1}{m} \sum_{i=1}^m \begin{pmatrix} \eta_{\boldsymbol{x},t} (\nabla_{\boldsymbol{x}} f_i(\boldsymbol{x}_i^t, \boldsymbol{y}_i^t; \xi_{i, j_t(i)}) - \nabla_{\boldsymbol{x}} f_i(\dot{\boldsymbol{x}}_i^t, \dot{\boldsymbol{y}}_i^t; \xi_{i, j_t(i)})) \\ -\eta_{\boldsymbol{y},t} (\nabla_{\boldsymbol{y}} f_i(\boldsymbol{x}_i^t, \boldsymbol{y}_i^t; \xi_{i, j_t(i)}) - \nabla_{\boldsymbol{y}} f_i(\dot{\boldsymbol{x}}_i^t, \dot{\boldsymbol{y}}_i^t; \xi_{i, j_t(i)})) \end{pmatrix} \right] \\
&\overset{(i)}{=} P_{\mathcal{X}, \mathcal{Y}} \left[ \begin{pmatrix} \boldsymbol{x}^t - \dot{\boldsymbol{x}}^t \\ \boldsymbol{y}^t - \dot{\boldsymbol{y}}^t \end{pmatrix} - \frac{1}{m} \left[ \sum_{i=1}^{m - m_0} \begin{pmatrix} \eta_{\boldsymbol{x},t} \nabla_{\boldsymbol{x}} f_i(\boldsymbol{x}_i^t, \boldsymbol{y}_i^t; \xi_{i, j_t(i)}) - \eta_{\boldsymbol{x},t} \nabla_{\boldsymbol{x}} f_i(\dot{\boldsymbol{x}}_i^t, \dot{\boldsymbol{y}}_i^t; \xi_{i, j_t(i)}) \\ -\eta_{\boldsymbol{y},t} \nabla_{\boldsymbol{y}} f_i(\boldsymbol{x}_i^t, \boldsymbol{y}_i^t; \xi_{i, j_t(i)}) + \eta_{\boldsymbol{y},t} \nabla_{\boldsymbol{y}} f_i(\dot{\boldsymbol{x}}_i^t, \dot{\boldsymbol{y}}_i^t; \xi_{i, j_t(i)}) \end{pmatrix} \right. \right. \\
&\left. \left. + \sum_{i=m-m_0+1}^m \begin{pmatrix} \eta_{\boldsymbol{x},t} \nabla_{\boldsymbol{x}} f_i(\boldsymbol{x}_i^t, \boldsymbol{y}_i^t; \xi_{i,n}) - \eta_{\boldsymbol{x},t} \nabla_{\boldsymbol{x}} f_i(\dot{\boldsymbol{x}}_i^t, \dot{\boldsymbol{y}}_i^t; \xi_{i,n}') \\ -\eta_{\boldsymbol{y},t} \nabla_{\boldsymbol{y}} f_i(\boldsymbol{x}_i^t, \boldsymbol{y}_i^t; \xi_{i,n}) + \eta_{\boldsymbol{y},t} \nabla_{\boldsymbol{y}} f_i(\dot{\boldsymbol{x}}_i^t, \dot{\boldsymbol{y}}_i^t; \xi_{i,n}') \end{pmatrix} \right] \right]
\end{aligned}
$$

where equality $(i)$ is due to the setting that agents from 1 to $m - m_0$ select $\xi_{i, j_t(i)}$ which is not the last different one while the rest agents from $m - m_0 + 1$ to $m$ select the last different samples, i.e., $\xi_{i,n}$ and $\xi_{i,n}'$ respectively.

Then we can decompose the term inside the projection into several parts as following:

$$
\underbrace{\frac{1}{m}\sum_{i=1}^{m-m_0}\left(\begin{array}{c} \boldsymbol{x}^t - \eta_{\boldsymbol{x},t}\nabla_{\boldsymbol{x}} f_i(\boldsymbol{x}^t,\boldsymbol{y}^t;\xi_{i,j_t(i)}) - (\dot{\boldsymbol{x}}^t - \eta_{\boldsymbol{x},t}\nabla_{\boldsymbol{x}} f_i(\dot{\boldsymbol{x}}^t,\dot{\boldsymbol{y}}^t;\xi_{i,j_t(i)})) \\ \boldsymbol{y}^t + \eta_{\boldsymbol{y},t}\nabla_{\boldsymbol{y}} f_i(\boldsymbol{x}^t,\boldsymbol{y}^t;\xi_{i,j_t(i)}) - (\dot{\boldsymbol{y}}^t + \eta_{\boldsymbol{y},t}\nabla_{\boldsymbol{y}} f_i(\dot{\boldsymbol{x}}^t,\dot{\boldsymbol{y}}^t;\xi_{i,j_t(i)})) \end{array}\right)}_{I_1}
$$

$$
+\underbrace{\frac{1}{m}\sum_{i=m-m_0+1}^{m}\left(\begin{array}{c} \boldsymbol{x}^t - \eta_{\boldsymbol{x},t}\nabla_{\boldsymbol{x}} f_i(\boldsymbol{x}^t,\boldsymbol{y}^t;\xi_{i,n}) - (\dot{\boldsymbol{x}}^t - \eta_{\boldsymbol{x},t}\nabla_{\boldsymbol{x}} f_i(\dot{\boldsymbol{x}}^t,\dot{\boldsymbol{y}}^t;\xi'_{i,n})) \\ \boldsymbol{y}^t + \eta_{\boldsymbol{y},t}\nabla_{\boldsymbol{y}} f_i(\boldsymbol{x}^t,\boldsymbol{y}^t;\xi_{i,n}) - (\dot{\boldsymbol{y}}^t + \eta_{\boldsymbol{y},t}\nabla_{\boldsymbol{y}} f_i(\dot{\boldsymbol{x}}^t,\dot{\boldsymbol{y}}^t;\xi'_{i,n})) \end{array}\right)}_{I_2}
$$

$$
+\underbrace{\frac{1}{m}\sum_{i=1}^{m-m_0}\left(\begin{array}{c} -\eta_{\boldsymbol{x},t}\left(\nabla_{\boldsymbol{x}} f_i(\boldsymbol{x}_i^t,\boldsymbol{y}_i^t;\xi_{i,j_t(i)}) - \nabla_{\boldsymbol{x}} f_i(\boldsymbol{x}^t,\boldsymbol{y}^t;\xi_{i,j_t(i)})\right) \\ \eta_{\boldsymbol{y},t}\left(\nabla_{\boldsymbol{y}} f_i(\boldsymbol{x}_i^t,\boldsymbol{y}_i^t;\xi_{i,j_t(i)}) - \nabla_{\boldsymbol{y}} f_i(\boldsymbol{x}^t,\boldsymbol{y}^t;\xi_{i,j_t(i)})\right) \end{array}\right)}_{II_1}
$$

$$
-\underbrace{\frac{1}{m}\sum_{i=1}^{m-m_0}\left(\begin{array}{c} -\eta_{\boldsymbol{x},t}\left(\nabla_{\boldsymbol{x}} f_i(\dot{\boldsymbol{x}}_i^t,\dot{\boldsymbol{y}}_i^t;\xi_{i,j_t(i)}) - \nabla_{\boldsymbol{x}} f_i(\dot{\boldsymbol{x}}^t,\dot{\boldsymbol{y}}^t;\xi_{i,j_t(i)})\right) \\ \eta_{\boldsymbol{y},t}\left(\nabla_{\boldsymbol{y}} f_i(\dot{\boldsymbol{x}}_i^t,\dot{\boldsymbol{y}}_i^t;\xi_{i,j_t(i)}) - \nabla_{\boldsymbol{y}} f_i(\dot{\boldsymbol{x}}^t,\dot{\boldsymbol{y}}^t;\xi_{i,j_t(i)})\right) \end{array}\right)}_{II_2}
$$

$$
+\underbrace{\frac{1}{m}\sum_{i=m-m_0+1}^{m}\left(\begin{array}{c} -\eta_{\boldsymbol{x},t}\left(\nabla_{\boldsymbol{x}} f_i(\boldsymbol{x}_i^t,\boldsymbol{y}_i^t;\xi_{i,n}) - \nabla_{\boldsymbol{x}} f_i(\boldsymbol{x}^t,\boldsymbol{y}^t;\xi_{i,n})\right) \\ \eta_{\boldsymbol{y},t}\left(\nabla_{\boldsymbol{y}} f_i(\boldsymbol{x}_i^t,\boldsymbol{y}_i^t;\xi_{i,n}) - \nabla_{\boldsymbol{y}} f_i(\boldsymbol{x}^t,\boldsymbol{y}^t;\xi_{i,n})\right) \end{array}\right)}_{II'_1}
$$

$$
-\underbrace{\frac{1}{m}\sum_{i=m-m_0+1}^{m}\left(\begin{array}{c} -\eta_{\boldsymbol{x},t}\left(\nabla_{\boldsymbol{x}} f_i(\dot{\boldsymbol{x}}_i^t,\dot{\boldsymbol{y}}_i^t;\xi'_{i,n}) - \nabla_{\boldsymbol{x}} f_i(\dot{\boldsymbol{x}}^t,\dot{\boldsymbol{y}}^t;\xi'_{i,n})\right) \\ \eta_{\boldsymbol{y},t}\left(\nabla_{\boldsymbol{y}} f_i(\dot{\boldsymbol{x}}_i^t,\dot{\boldsymbol{y}}_i^t;\xi'_{i,n}) - \nabla_{\boldsymbol{y}} f_i(\dot{\boldsymbol{x}}^t,\dot{\boldsymbol{y}}^t;\xi'_{i,n})\right) \end{array}\right)}_{II'_2}
\tag{7}
$$

According to Lemma 1 and the preoperty of Lipschitz continuity (see Assumption 1), we can bound the term $I_1$ as:

$$
\|I_1\| \leq \frac{m-m_0}{m}(1-\eta_t^{min}\frac{L\mu}{L+\mu})\left\|\left(\begin{array}{c} \boldsymbol{x}^t - \dot{\boldsymbol{x}}^t \\ \boldsymbol{y}^t - \dot{\boldsymbol{y}}^t \end{array}\right)\right\|
$$

where $\eta_t^{min} \triangleq \min\{\eta_{\boldsymbol{x},t},\eta_{\boldsymbol{y},t}\}$, $\eta_t^{max} \triangleq \max\{\eta_{\boldsymbol{x},t},\eta_{\boldsymbol{y},t}\}$, and $\mu \triangleq \min\{\mu_{\boldsymbol{x}},\mu_{\boldsymbol{y}}\}$.
For the term $I_2$, it is bounded by:

$$
\|I_2\| \leq \frac{1}{m}\sum_{i=m-m_0+1}^{m}\left\|\left(\begin{array}{c} \boldsymbol{x}^t - \eta_{\boldsymbol{x},t}\nabla_{\boldsymbol{x}} f_i(\boldsymbol{x}^t,\boldsymbol{y}^t;\xi_{i,n}) - (\dot{\boldsymbol{x}}^t - \eta_{\boldsymbol{x},t}\nabla_{\boldsymbol{x}} f_i(\dot{\boldsymbol{x}}^t,\dot{\boldsymbol{y}}^t;\xi'_{i,n})) \\ \boldsymbol{y}^t + \eta_{\boldsymbol{y},t}\nabla_{\boldsymbol{y}} f_i(\boldsymbol{x}^t,\boldsymbol{y}^t;\xi_{i,n}) - (\dot{\boldsymbol{y}}^t + \eta_{\boldsymbol{y},t}\nabla_{\boldsymbol{y}} f_i(\dot{\boldsymbol{x}}^t,\dot{\boldsymbol{y}}^t;\xi'_{i,n})) \end{array}\right)\right\|
$$

$$
\leq \frac{1}{m}\sum_{i=m-m_0+1}^{m}\left[\left\|\left(\begin{array}{c} \boldsymbol{x}^t - \eta_{\boldsymbol{x},t}\nabla_{\boldsymbol{x}} f_i(\boldsymbol{x}^t,\boldsymbol{y}^t;\xi_{i,n}) - (\dot{\boldsymbol{x}}^t - \eta_{\boldsymbol{x},t}\nabla_{\boldsymbol{x}} f_i(\dot{\boldsymbol{x}}^t,\dot{\boldsymbol{y}}^t;\xi_{i,n})) \\ \boldsymbol{y}^t + \eta_{\boldsymbol{y},t}\nabla_{\boldsymbol{y}} f_i(\boldsymbol{x}^t,\boldsymbol{y}^t;\xi_{i,n}) - (\dot{\boldsymbol{y}}^t + \eta_{\boldsymbol{y},t}\nabla_{\boldsymbol{y}} f_i(\dot{\boldsymbol{x}}^t,\dot{\boldsymbol{y}}^t;\xi_{i,n})) \end{array}\right)\right\|\right.
$$

$$
\left.+\left\|\left(\begin{array}{c} \eta_{\boldsymbol{x},t}\nabla_{\boldsymbol{x}} f_i(\dot{\boldsymbol{x}}^t,\dot{\boldsymbol{y}}^t;\xi'_{i,n}) - \eta_{\boldsymbol{x},t}\nabla_{\boldsymbol{x}} f_i(\dot{\boldsymbol{x}}^t,\dot{\boldsymbol{y}}^t;\xi_{i,n}) \\ \eta_{\boldsymbol{y},t}\nabla_{\boldsymbol{y}} f_i(\dot{\boldsymbol{x}}^t,\dot{\boldsymbol{y}}^t;\xi_{i,n}) - \eta_{\boldsymbol{y},t}\nabla_{\boldsymbol{y}} f_i(\dot{\boldsymbol{x}}^t,\dot{\boldsymbol{y}}^t;\xi'_{i,n}) \end{array}\right)\right\|\right]
$$

$$
\leq \frac{m_0}{m}\left[(1-\eta_t^{min}\frac{L\mu}{L+\mu})\left\|\left(\begin{array}{c} \boldsymbol{x}^t - \dot{\boldsymbol{x}}^t \\ \boldsymbol{y}^t - \dot{\boldsymbol{y}}^t \end{array}\right)\right\| + 2\eta_t^{max}G\right]
$$

For the term $II_1$ and $II_1'$,

$$
\|II_1 + II_1'\| \le \frac{1}{m} \sum_{i=1}^{m-m_0} \left[ \eta_t^{max} \left\| \left( \begin{array}{c} \nabla_{\boldsymbol{x}} f_i(\boldsymbol{x}_i^t, \boldsymbol{y}_i^t; \xi_{i,j_t(i)}) - \nabla_{\boldsymbol{x}} f_i(\boldsymbol{x}^t, \boldsymbol{y}^t; \xi_{i,j_t(i)}) \\ \nabla_{\boldsymbol{y}} f_i(\boldsymbol{x}^t, \boldsymbol{y}^t; \xi_{i,j_t(i)}) - \nabla_{\boldsymbol{y}} f_i(\boldsymbol{x}_i^t, \boldsymbol{y}_i^t; \xi_{i,j_t(i)}) \end{array} \right) \right\| \right]
$$

$$
+ \frac{1}{m} \sum_{i=m-m_0+1}^{m} \left[ \eta_t^{max} \left\| \left( \begin{array}{c} \nabla_{\boldsymbol{x}} f_i(\boldsymbol{x}_i^t, \boldsymbol{y}_i^t; \xi_{i,n}) - \nabla_{\boldsymbol{x}} f_i(\boldsymbol{x}^t, \boldsymbol{y}^t; \xi_{i,n}) \\ \nabla_{\boldsymbol{y}} f_i(\boldsymbol{x}^t, \boldsymbol{y}^t; \xi_{i,n}) - \nabla_{\boldsymbol{y}} f_i(\boldsymbol{x}_i^t, \boldsymbol{y}_i^t; \xi_{i,n}) \end{array} \right) \right\| \right]
$$

$$
\le \frac{1}{m} \left[ \sum_{i=1}^{m} \eta_t^{max} L \left\| \left( \begin{array}{c} \boldsymbol{x}_i^t - \boldsymbol{x}^t \\ \boldsymbol{y}_i^t - \boldsymbol{y}^t \end{array} \right) \right\| \right]
$$

$$
\le \frac{\eta_t^{max} L}{m} \sum_{i=1}^{m} \left\| \left( \begin{array}{c} \boldsymbol{x}_i^t - \boldsymbol{x}^t \\ \boldsymbol{y}_i^t - \boldsymbol{y}^t \end{array} \right) \right\|
$$

$$
\le \frac{\eta_t^{max} L}{\sqrt{m}} \left[ \sum_{i=1}^{m} \|\boldsymbol{x}_i^t - \boldsymbol{x}^t\|^2 + \|\boldsymbol{y}_i^t - \boldsymbol{y}^t\|^2 \right]^{1/2}
$$

$$
\le 2\eta_t^{max} L G \sum_{s=0}^{t-1} \eta_s^{max} \lambda^{t-1-s}
$$

And it is the same as the term $II_2 + II_2'$ that $\|II_2 + II_2'\| \le 2\eta_t^{max} L G \sum_{s=0}^{t-1} \eta_s^{max} \lambda^{t-1-s}$.
Taking expectation on the choice of each local sample, we can get:

$$
\mathbb{E}_{\mathcal{A}} \left\| \left( \begin{array}{c} \boldsymbol{x}^{t+1} - \dot{\boldsymbol{x}}^{t+1} \\ \boldsymbol{y}^{t+1} - \dot{\boldsymbol{y}}^{t+1} \end{array} \right) \right\|
$$

$$
\le \sum_{m_0=0}^{m} C_m^{m_0} (1 - \frac{1}{n})^{m-m_0} (\frac{1}{n})^{m_0} \left( \left( 1 - \eta_t^{min} \frac{L\mu}{L+\mu} \right) \mathbb{E}_{\mathcal{A}} \left\| \left( \begin{array}{c} \boldsymbol{x}^t - \dot{\boldsymbol{x}}^t \\ \boldsymbol{y}^t - \dot{\boldsymbol{y}}^t \end{array} \right) \right\| + \frac{m_0}{m} 2\eta_t^{max} G \right.
$$

$$
\left. + 4\eta_t^{max} L G \sum_{s=0}^{t-1} \eta_s^{max} \lambda^{t-1-s} \right)
$$

$$
\le \left( 1 - \eta_t^{min} \frac{L\mu}{L+\mu} \right) \mathbb{E}_{\mathcal{A}} \left\| \left( \begin{array}{c} \boldsymbol{x}^t - \dot{\boldsymbol{x}}^t \\ \boldsymbol{y}^t - \dot{\boldsymbol{y}}^t \end{array} \right) \right\| + 4\eta_t^{max} L G \sum_{s=0}^{t-1} \eta_s^{max} \lambda^{t-1-s}
$$

$$
+ \sum_{m_0=0}^{m} C_m^{m_0} (1 - \frac{1}{n})^{m-m_0} (\frac{1}{n})^{m_0} \frac{m_0}{m} 2\eta_t^{max} G
$$

$$
= \left( 1 - \eta_t^{min} \frac{L\mu}{L+\mu} \right) \mathbb{E}_{\mathcal{A}} \left\| \left( \begin{array}{c} \boldsymbol{x}^t - \dot{\boldsymbol{x}}^t \\ \boldsymbol{y}^t - \dot{\boldsymbol{y}}^t \end{array} \right) \right\| + 4\eta_t^{max} L G \sum_{s=0}^{t-1} \eta_s^{max} \lambda^{t-1-s} + \frac{2\eta_t^{max} G}{n}
$$

$$
\tag{8}
$$

where the last inequality is due to the property of binomial coefficient that $C_m^{m_0} \cdot \frac{m_0}{m} = C_{m-1}^{m_0-1}$.
Recursively applying the above inequality, we can get:

$$
\mathbb{E}_{\mathcal{A}} \left\| \left( \begin{array}{c} \mathcal{A}_{\boldsymbol{x}}(\mathcal{S}) - \mathcal{A}_{\boldsymbol{x}}(\mathcal{S}') \\ \mathcal{A}_{\boldsymbol{y}}(\mathcal{S}) - \mathcal{A}_{\boldsymbol{y}}(\mathcal{S}') \end{array} \right) \right\|
$$

$$
\le \frac{2G}{n} \sum_{k=0}^{T-1} \eta_k^{max} \prod_{s=k+1}^{T-1} (1 - \eta_s^{min} \frac{L\mu}{L+\mu}) + 4GL \sum_{k=1}^{T-1} \left( \eta_k^{max} \sum_{s=0}^{k-1} \eta_s^{max} \lambda^{k-1-s} \right) \prod_{j=k+1}^{T-1} (1 - \eta_j^{min} \frac{L\mu}{L+\mu})
$$

$$
\tag{9}
$$

where the initial difference is zero that $\boldsymbol{X}^0 = 0$ and $\boldsymbol{Y}^0 = 0$. And we use the last iterate as the output of Algorithm $\mathcal{A}$, i.e., $(\mathcal{A}_{\boldsymbol{x}}(\mathcal{S}), \mathcal{A}_{\boldsymbol{y}}(\mathcal{S})) = (\boldsymbol{x}^T, \boldsymbol{y}^T)$.

For case a. when $\eta_{\boldsymbol{x},t}$ and $\eta_{\boldsymbol{y},t}$ are fixed, we further get:

$$\epsilon_{sta}^{arg}(\mathcal{A}) \leq \frac{2G}{n}\sum_{k=0}^{T-1}\eta^{max}(1-\eta^{min}\frac{L\mu}{L+\mu})^{T-k-1} + 4GL\sum_{k=1}^{T-1}\left(\eta^{max}\sum_{s=0}^{k-1}\eta^{max}\lambda^{k-1-s}\right)(1-\eta^{min}\frac{L\mu}{L+\mu})^{T-k-1}$$

$$\leq \left(4\eta^{max}GL\eta^{max}\frac{1}{1-\lambda} + \frac{2\eta^{max}G}{n}\right)\sum_{k=0}^{T-1}(1-\eta^{min}\frac{L\mu}{L+\mu})^k$$

$$\leq 2G\frac{L+\mu}{\eta^{min}L\mu}\left(\frac{2(\eta^{max})^2L}{1-\lambda} + \frac{\eta^{max}}{n}\right)$$

Further when $\eta^{max} = \eta^{min} \triangleq \eta$, the argument stability turns out to be

$$\epsilon_{sta}^{arg}(\mathcal{A}) \leq 2G\frac{L+\mu}{L\mu}\left(\frac{2\eta L}{1-\lambda} + \frac{1}{n}\right)$$

For case b. when $\eta_t^{max} = \frac{1}{\mu(t+1)^c}$ and $\eta_t^{min} = \frac{1}{\mu(t+1)}, c \leq 1$. Since $1 \pm a \leq \exp\{a\}$, we have:

$$\prod_{j=k+1}^{T-1}(1-\eta_j^{min}\frac{L\mu}{L+\mu}) = \prod_{j=k+1}^{T-1}(1-\frac{1}{j+1}\frac{L}{L+\mu})$$

$$\leq \prod_{j=k+1}^{T-1}\exp\left\{-\frac{1}{j+1}\frac{L}{L+\mu}\right\}$$

$$= \exp\left\{\sum_{j=k+1}^{T-1} -\frac{1}{j+1}\frac{L}{L+\mu}\right\}$$

$$\leq \exp\left\{-\frac{L}{L+\mu}(\ln T - \ln(k+1))\right\}$$

$$= (\frac{k+1}{T})^{\frac{L}{L+\mu}}$$

Then back to inequality (9), we can simplify it as:

$$\mathbb{E}_{\mathcal{A}}\left\|\begin{pmatrix}\mathcal{A}_{\boldsymbol{x}}(\mathcal{S}) - \mathcal{A}_{\boldsymbol{x}}(\mathcal{S}')\\ \mathcal{A}_{\boldsymbol{y}}(\mathcal{S}) - \mathcal{A}_{\boldsymbol{y}}(\mathcal{S}')\end{pmatrix}\right\|$$

$$\leq \frac{2G}{n}\sum_{k=0}^{T-1}\frac{1}{\mu(k+1)^c}(\frac{k+1}{T})^{\frac{L}{L+\mu}} + 4GL\sum_{k=1}^{T-1}\left(\frac{1}{\mu(k+1)^c}\sum_{s=0}^{k-1}\frac{1}{\mu(s+1)^c}\lambda^{k-1-s}\right)(\frac{k+1}{T})^{\frac{L}{L+\mu}}$$

$$\leq \frac{2G}{\mu n T^{\frac{L}{L+\mu}}}\sum_{k=0}^{T-1}\frac{1}{(k+1)^{c-\frac{L}{L+\mu}}} + \frac{4GL}{\mu^2 T^{\frac{L}{L+\mu}}}\sum_{k=1}^{T-1}\frac{1}{(k+1)^{c-\frac{L}{L+\mu}}}\frac{C_\lambda}{k^c}$$

$\square$

## E.2   Proof of the Empirical Risk

Then we are going to present the optimization error of D-SGDA in the SC-SC condition.

**Theorem 7 (Optimization error).** *Under assumption 1,2,3, each local function is $f_i$ is $\mu_{\boldsymbol{x}}SC$-$\mu_{\boldsymbol{y}}SC$. We further bound the restriction set by $\sup_{\boldsymbol{x}\in\mathcal{X}}\|\boldsymbol{x}\| \leq C_{\boldsymbol{x}}$ and $\sup_{\boldsymbol{y}\in\mathcal{Y}}\|\boldsymbol{y}\| \leq C_{\boldsymbol{y}}$, then we have the strong primal-dual empirical risk over the dataset $\mathcal{S}$ on the average output in $T$ iterations as following:*

*a. When learning rates $\eta_{\boldsymbol{x},t}$ and $\eta_{\boldsymbol{y},t}$ are fixed,*

$$\Delta_{\mathcal{S}}^s(\boldsymbol{x}_{ave}^T, \boldsymbol{y}_{ave}^T) \leq \frac{C_{\boldsymbol{x}}^2 + C_{\boldsymbol{y}}^2}{2\eta^{min}T} + \eta^{max}G^2 + \frac{4(C_{\boldsymbol{x}} + C_{\boldsymbol{y}})GL\eta^{max}}{1-\lambda} + \frac{2(C_{\boldsymbol{x}} + C_{\boldsymbol{y}})G}{\sqrt{T}}.$$

*b. When learning rates are varying that $\eta_{\boldsymbol{x},t} = \frac{1}{\mu_{\boldsymbol{x}}(t+1)^{c_{\boldsymbol{x}}}}$ and $\eta_{\boldsymbol{y},t} = \frac{1}{\mu_{\boldsymbol{y}}(t+1)^{c_{\boldsymbol{y}}}}$,*

$$\Delta_{\mathcal{S}}^{s}(\boldsymbol{x}_{ave}^{T}, \boldsymbol{y}_{ave}^{T}) \leq \frac{2G(C_{\boldsymbol{x}} + C_{\boldsymbol{y}})}{\sqrt{T}} + T_{\boldsymbol{x}} + T_{\boldsymbol{y}} + T_{max}.$$

*where $\alpha \in \{\boldsymbol{x}, \boldsymbol{y}\}$, $k_{min} = \min\{c_{\boldsymbol{x}}, c_{\boldsymbol{y}}\}$ and $\mu = \min\{\mu_{\boldsymbol{x}}, \mu_{\boldsymbol{y}}\}$:*

$$T_{\alpha} = \begin{cases} \frac{G^2}{2\mu_{\alpha}} \frac{1+\ln T}{T} & c_{\alpha} = 1 \\ \frac{G^2}{2\mu_{\alpha}(1-c_{\alpha})T^{c_{\alpha}}} & 0 < c_{\alpha} < 1 \end{cases} \quad ; \quad T_{max} = \begin{cases} \frac{4GLC_{\lambda}(C_{\boldsymbol{x}}+C_{\boldsymbol{y}})\ln T}{\mu T} & k_{min} = 1 \\ \frac{4GLC_{\lambda}(C_{\boldsymbol{x}}+C_{\boldsymbol{y}})}{\mu(1-k_{min})T^{k_{min}}} & 0 < k_{min} < 1 \end{cases}.$$

*Proof of Theorem 7.* First, we should notice that when each $f_i$ owns the property of $\mu_{\boldsymbol{x}}$-strong convexity and $\mu_{\boldsymbol{y}}$-strong concavity, then the linear summation should inherit the properties, i.e., $F_{\mathcal{S}}(\boldsymbol{x}, \boldsymbol{y}) = \frac{1}{m}\sum_{i=1}^{m}\frac{1}{n}\sum_{l=1}^{n}f_i(\boldsymbol{x}, \boldsymbol{y}; \xi_{i,l})$ is $\mu_{\boldsymbol{x}}$-strongly convex on $\boldsymbol{x}$ and $\mu_{\boldsymbol{y}}$-strongly concave on $\boldsymbol{y}$.

First observe the update rule of $\boldsymbol{x}$ (see Algorithm 1), for any given $\boldsymbol{x} \in \mathcal{X}$:

$$\|\boldsymbol{x}^{t+1} - \boldsymbol{x}\|^2$$

$$\leq \left\|\boldsymbol{x}^t - \frac{1}{m}\sum_{i=1}^{m}\eta_{\boldsymbol{x},t}\nabla_{\boldsymbol{x}}f_i(\boldsymbol{x}_i^t, \boldsymbol{y}_i^t; \xi_{i,j_t(i)}) - \boldsymbol{x}\right\|^2$$

$$= \|\boldsymbol{x}^t - \boldsymbol{x}\|^2 + \left\|\frac{1}{m}\sum_{i=1}^{m}\eta_{\boldsymbol{x},t}\nabla_{\boldsymbol{x}}f_i(\boldsymbol{x}_i^t, \boldsymbol{y}_i^t; \xi_{i,j_t(i)})\right\|^2 - 2\left\langle\boldsymbol{x}^t - \boldsymbol{x}, \frac{1}{m}\sum_{i=1}^{m}\eta_{\boldsymbol{x},t}\nabla_{\boldsymbol{x}}f_i(\boldsymbol{x}_i^t, \boldsymbol{y}_i^t; \xi_{i,j_t(i)})\right\rangle$$

$$(10)$$

where for the first inequality, we can recall how we transform in the proof for stability (see Appendix E.1) that $x^{t+1} = (\boldsymbol{X}^{t+1})^T\frac{\mathbb{1}_m}{m} = (\boldsymbol{W}\boldsymbol{X}^t - \eta_{\boldsymbol{x},t}\nabla_{\boldsymbol{x}}\boldsymbol{f}(\boldsymbol{X}^t, \boldsymbol{Y}^t; \boldsymbol{\xi}^t))^T\frac{\mathbb{1}_m}{m} = \boldsymbol{x}^t - \eta_{\boldsymbol{x},t}\nabla_{\boldsymbol{x}}\boldsymbol{f}(\boldsymbol{X}^t, \boldsymbol{Y}^t; \boldsymbol{\xi}^t)^T\frac{\mathbb{1}_m}{m}$ regardless of the projection.

For the second term, we have:

$$\left\|\frac{1}{m}\sum_{i=1}^{m}\eta_{\boldsymbol{x},t}\nabla_{\boldsymbol{x}}f_i(\boldsymbol{x}_i^t, \boldsymbol{y}_i^t; \xi_{i,j_t(i)})\right\|^2$$

$$= \frac{\eta_{\boldsymbol{x},t}^2}{m^2}\left[\sum_{i=1}^{m}\|\nabla_{\boldsymbol{x}}f_i(\boldsymbol{x}_i^t, \boldsymbol{y}_i^t; \xi_{i,j_t(i)})\|^2 + \sum_{i \neq k}\langle\nabla_{\boldsymbol{x}}f_i(\boldsymbol{x}_i^t, \boldsymbol{y}_i^t; \xi_{i,j_t(i)}), \nabla_{\boldsymbol{x}}f_k(\boldsymbol{x}_k^t, \boldsymbol{y}_k^t; \xi_{k,j_t(k)})\rangle\right]$$

$$\leq \frac{\eta_{\boldsymbol{x},t}^2}{m^2}\left(mG^2 + (m^2 - m)G^2\right)$$

$$= \eta_{\boldsymbol{x},t}^2 G^2$$

$$(11)$$

We decompose the third term in association with the empirical function that:

$$2\eta_{\boldsymbol{x},t}\left\langle\boldsymbol{x} - \boldsymbol{x}^t, \frac{1}{m}\sum_{i=1}^{m}\nabla_{\boldsymbol{x}}f_i(\boldsymbol{x}_i^t, \boldsymbol{y}_i^t; \xi_{i,j_t(i)})\right\rangle$$

$$= 2\eta_{\boldsymbol{x},t}\langle\boldsymbol{x} - \boldsymbol{x}^t, \nabla_{\boldsymbol{x}}F_{\mathcal{S}}(\boldsymbol{x}^t, \boldsymbol{y}^t)\rangle + 2\eta_{\boldsymbol{x},t}\left\langle\boldsymbol{x} - \boldsymbol{x}^t, \frac{1}{m}\sum_{i=1}^{m}\nabla_{\boldsymbol{x}}f_i(\boldsymbol{x}_i^t, \boldsymbol{y}_i^t; \xi_{i,j_t(i)}) - \nabla_{\boldsymbol{x}}F_{\mathcal{S}}(\boldsymbol{x}^t, \boldsymbol{y}^t)\right\rangle$$

$$\leq 2\eta_{\boldsymbol{x},t}(F_{\mathcal{S}}(\boldsymbol{x}, \boldsymbol{y}^t) - F_{\mathcal{S}}(\boldsymbol{x}^t, \boldsymbol{y}^t)) - \eta_{\boldsymbol{x},t}\mu_{\boldsymbol{x}}\|\boldsymbol{x} - \boldsymbol{x}^t\|^2$$

$$+ 2\eta_{\boldsymbol{x},t}\left\langle\boldsymbol{x} - \boldsymbol{x}^t, \frac{1}{m}\sum_{i=1}^{m}\nabla_{\boldsymbol{x}}f_i(\boldsymbol{x}_i^t, \boldsymbol{y}_i^t; \xi_{i,j_t(i)}) - \nabla_{\boldsymbol{x}}F_{\mathcal{S}}(\boldsymbol{x}^t, \boldsymbol{y}^t)\right\rangle$$

$$(12)$$

where the last inequality is due to the strong convexity of the empirical function $F_{\mathcal{S}}$ on parameter $\boldsymbol{x}$.

Then we combine the inequalities (10) (11) (12) and we can get:

$$2\eta_{\boldsymbol{x},t}(F_{\mathcal{S}}(\boldsymbol{x}^t, \boldsymbol{y}^t) - F_{\mathcal{S}}(\boldsymbol{x}, \boldsymbol{y}^t)) \leq (1 - \eta_{\boldsymbol{x},t}\mu_{\boldsymbol{x}})\|\boldsymbol{x} - \boldsymbol{x}^t\|^2 - \|\boldsymbol{x}^{t+1} - \boldsymbol{x}\|^2 + \eta_{\boldsymbol{x},t}^2 G^2$$

$$+ 2\eta_{\boldsymbol{x},t}\left\langle\boldsymbol{x} - \boldsymbol{x}^t, \frac{1}{m}\sum_{i=1}^{m}\nabla_{\boldsymbol{x}}f_i(\boldsymbol{x}_i^t, \boldsymbol{y}_i^t; \xi_{i,j_t(i)}) - \nabla_{\boldsymbol{x}}F_{\mathcal{S}}(\boldsymbol{x}^t, \boldsymbol{y}^t)\right\rangle$$

$$(13)$$

For case a. when learning rates $\eta_{\boldsymbol{x},t}$ and $\eta_{\boldsymbol{y},t}$ are fixed and we write them as $\eta_{\boldsymbol{x}}$ and $\eta_{\boldsymbol{y}}$ respectively.

Taking summatation over the above inequality from $t = 0$ to $t = T$ and using the concavity of the empirical function on $\boldsymbol{y}$:

$$\sum_{t=0}^{T-1} 2\eta_{\boldsymbol{x}} F_{\mathcal{S}}(\boldsymbol{x}^t, \boldsymbol{y}^t) - F_{\mathcal{S}}(\boldsymbol{x}, \boldsymbol{y}_{ave}^T)$$

$$\leq \sum_{t=0}^{T-1} 2\eta_{\boldsymbol{x}} (F_{\mathcal{S}}(\boldsymbol{x}^t, \boldsymbol{y}^t) - F_{\mathcal{S}}(\boldsymbol{x}, \boldsymbol{y}^t))$$

$$\leq (1 - \eta_{\boldsymbol{x}}\mu_{\boldsymbol{x}})\|\boldsymbol{x}\|^2 + \eta_{\boldsymbol{x}}^2 G^2 T + 2\eta_{\boldsymbol{x}} \sum_{t=0}^{T-1} \langle \boldsymbol{x} - \boldsymbol{x}^t, \frac{1}{m}\sum_{i=1}^{m} \nabla_{\boldsymbol{x}} f_i(\boldsymbol{x}_i^t, \boldsymbol{y}_i^t; \xi_{i,j_t(i)}) - \nabla_{\boldsymbol{x}} F_{\mathcal{S}}(\boldsymbol{x}^t, \boldsymbol{y}^t) \rangle$$

$$\leq (1 - \eta_{\boldsymbol{x}}\mu_{\boldsymbol{x}})\|\boldsymbol{x}\|^2 + \eta_{\boldsymbol{x}}^2 G^2 T + 2\eta_{\boldsymbol{x}} \sum_{t=0}^{T-1} \langle \boldsymbol{x}^t, \nabla_{\boldsymbol{x}} F_{\mathcal{S}}(\boldsymbol{x}^t, \boldsymbol{y}^t) - \frac{1}{m}\sum_{i=1}^{m} \nabla_{\boldsymbol{x}} f_i(\boldsymbol{x}_i^t, \boldsymbol{y}_i^t; \xi_{i,j_t(i)}) \rangle$$

$$+ 2\eta_{\boldsymbol{x}}\|\boldsymbol{x}\| \left\| \sum_{t=0}^{T-1} \left( \frac{1}{m}\sum_{i=1}^{m} \nabla_{\boldsymbol{x}} f_i(\boldsymbol{x}_i^t, \boldsymbol{y}_i^t; \xi_{i,j_t(i)}) - \nabla_{\boldsymbol{x}} F_{\mathcal{S}}(\boldsymbol{x}^t, \boldsymbol{y}^t) \right) \right\|$$

Then we take expectation on the randomness of the algorithm (notice that we do not take expectation on the randomness of dataset) on both sides of above inequality and choose the infinity of $\boldsymbol{x}$ on the left side since the above inequality holds for any $\boldsymbol{x} \in \mathcal{X}$:

$$\sum_{t=0}^{T-1} 2\eta_{\boldsymbol{x}} \mathbb{E}_{\mathcal{A}}[F_{\mathcal{S}}(\boldsymbol{x}^t, \boldsymbol{y}^t) - \inf_{\boldsymbol{x} \in \mathcal{X}} F_{\mathcal{S}}(\boldsymbol{x}, \boldsymbol{y}_{ave}^T)]$$

$$\leq (1 - \eta_{\boldsymbol{x}}\mu_{\boldsymbol{x}})C_{\boldsymbol{x}}^2 + \eta_{\boldsymbol{x}}^2 G^2 T + 2\eta_{\boldsymbol{x}} \underbrace{\sum_{t=0}^{T-1} \mathbb{E}_{\mathcal{A}} \langle \boldsymbol{x}^t, \nabla_{\boldsymbol{x}} F_{\mathcal{S}}(\boldsymbol{x}^t, \boldsymbol{y}^t) - \frac{1}{m}\sum_{i=1}^{m} \nabla_{\boldsymbol{x}} f_i(\boldsymbol{x}_i^t, \boldsymbol{y}_i^t; \xi_{i,j_t(i)}) \rangle}_{I}$$

$$+ 2C_{\boldsymbol{x}}\eta_{\boldsymbol{x}} \underbrace{\mathbb{E}_{\mathcal{A}} \left\| \sum_{t=0}^{T-1} \left( \frac{1}{m}\sum_{i=1}^{m} \nabla_{\boldsymbol{x}} f_i(\boldsymbol{x}_i^t, \boldsymbol{y}_i^t; \xi_{i,j_t(i)}) - \nabla_{\boldsymbol{x}} F_{\mathcal{S}}(\boldsymbol{x}^t, \boldsymbol{y}^t) \right) \right\|}_{II} \tag{14}$$

For term $I$, it is decomposed as:

$$\sum_{t=0}^{T-1} \mathbb{E}_{j_t} \langle \boldsymbol{x}^t, \nabla_{\boldsymbol{x}} F_{\mathcal{S}}(\boldsymbol{x}^t, \boldsymbol{y}^t) - \frac{1}{m}\sum_{i=1}^{m} \nabla_{\boldsymbol{x}} f_i(\boldsymbol{x}_i^t, \boldsymbol{y}_i^t; \xi_{i,j_t(i)}) \rangle$$

$$= \sum_{t=0}^{T-1} \mathbb{E}_{j_t} \langle \boldsymbol{x}^t, \frac{1}{m}\sum_{i=1}^{m} \frac{1}{n}\sum_{l=1}^{n} \nabla_{\boldsymbol{x}} f_i(\boldsymbol{x}^t, \boldsymbol{y}^t; \xi_{i,l}) - \frac{1}{m}\sum_{i=1}^{m} \nabla_{\boldsymbol{x}} f_i(\boldsymbol{x}^t, \boldsymbol{y}^t; \xi_{i,j_t(i)}) \rangle$$

$$+ \sum_{t=0}^{T-1} \mathbb{E}_{j_t} \langle \boldsymbol{x}^t, \frac{1}{m}\sum_{i=1}^{m} \nabla_{\boldsymbol{x}} f_i(\boldsymbol{x}^t, \boldsymbol{y}^t; \xi_{i,j_t(i)}) - \frac{1}{m}\sum_{i=1}^{m} \nabla_{\boldsymbol{x}} f_i(\boldsymbol{x}_i^t, \boldsymbol{y}_i^t; \xi_{i,j_t(i)}) \rangle$$

$$\overset{(i)}{\leq} 2C_{\boldsymbol{x}} \sum_{t=0}^{T-1} \mathbb{E}_{\mathcal{A}} \left\| \frac{1}{m}\sum_{i=1}^{m} \left( \nabla_{\boldsymbol{x}} f_i(\boldsymbol{x}^t, \boldsymbol{y}^t; \xi_{i,j_t(i)}) - \nabla_{\boldsymbol{x}} f_i(\boldsymbol{x}_i^t, \boldsymbol{y}_i^t; \xi_{i,j_t(i)}) \right) \right\|$$

$$\leq 2C_{\boldsymbol{x}} \sum_{t=0}^{T-1} \frac{L}{m} \mathbb{E}_{\mathcal{A}} \left[ \sum_{i=1}^{m} \left( \|\boldsymbol{x}^t - \boldsymbol{x}_i^t\|^2 + \|\boldsymbol{y}^t - \boldsymbol{y}_i^t\|^2 \right)^{1/2} \right]$$

$$\overset{(ii)}{\leq} 4C_{\boldsymbol{x}} GL \sum_{t=0}^{T-1}\sum_{s=0}^{t-1} \eta_s^{max} \lambda^{t-1-s}$$

where inequality $(i)$ is because of the fact that taking expectation on the randomness of $j_t$ is equal to the empirical result on the dataset $\mathcal{S}$ and inequality $(ii)$ is due to Cauchy-Schwarz inequality and Lemma 2.

Then for the term $II$, we decompose its corresponding quadratic one:

$$\left( \mathbb{E}_{\mathcal{A}} \left\| \sum_{t=0}^{T-1} \left( \frac{1}{m} \sum_{i=1}^{m} \nabla_{\boldsymbol{x}} f_i(\boldsymbol{x}_i^t, \boldsymbol{y}_i^t; \xi_{i,j_t(i)}) - \nabla_{\boldsymbol{x}} F_{\mathcal{S}}(\boldsymbol{x}^t, \boldsymbol{y}^t) \right) \right\| \right)^2$$

$$\leq \mathbb{E}_{\mathcal{A}} \left\| \sum_{t=0}^{T-1} \left( \frac{1}{m} \sum_{i=1}^{m} \nabla_{\boldsymbol{x}} f_i(\boldsymbol{x}_i^t, \boldsymbol{y}_i^t; \xi_{i,j_t(i)}) - \nabla_{\boldsymbol{x}} F_{\mathcal{S}}(\boldsymbol{x}^t, \boldsymbol{y}^t) \right) \right\|^2$$

$$= \sum_{t=0}^{T-1} \mathbb{E}_{\mathcal{A}} \left\| \frac{1}{m} \sum_{i=1}^{m} \nabla_{\boldsymbol{x}} f_i(\boldsymbol{x}_i^t, \boldsymbol{y}_i^t; \xi_{i,j_t(i)}) - \frac{1}{m} \sum_{i=1}^{m} \frac{1}{n} \sum_{l=1}^{n} \nabla_{\boldsymbol{x}} f_i(\boldsymbol{x}^t, \boldsymbol{y}^t; \xi_{i,l}) \right\|^2$$

$$+ \sum_{t \neq t'} \mathbb{E}_{\mathcal{A}} \langle \frac{1}{m} \sum_{i=1}^{m} \nabla_{\boldsymbol{x}} f_i(\boldsymbol{x}_i^t, \boldsymbol{y}_i^t; \xi_{i,j_t(i)}) - \frac{1}{m} \sum_{i=1}^{m} \frac{1}{n} \sum_{l=1}^{n} \nabla_{\boldsymbol{x}} f_i(\boldsymbol{x}^t, \boldsymbol{y}^t; \xi_{i,l}),$$

$$\frac{1}{m} \sum_{i=1}^{m} \nabla_{\boldsymbol{x}} f_i(\boldsymbol{x}_i^{t'}, \boldsymbol{y}_i^{t'}; \xi_{i,j_{t'}(i)}) - \frac{1}{m} \sum_{i=1}^{m} \frac{1}{n} \sum_{l=1}^{n} \nabla_{\boldsymbol{x}} f_i(\boldsymbol{x}^{t'}, \boldsymbol{y}^{t'}; \xi_{i,l}) \rangle$$

$$\leq 4G^2 T + \sum_{t \neq t'} \mathbb{E}_{\mathcal{A}} \langle \frac{1}{m} \sum_{i=1}^{m} \nabla_{\boldsymbol{x}} f_i(\boldsymbol{x}_i^t, \boldsymbol{y}_i^t; \xi_{i,j_t(i)}) - \frac{1}{m} \sum_{i=1}^{m} \nabla_{\boldsymbol{x}} f_i(\boldsymbol{x}^t, \boldsymbol{y}^t; \xi_{i,j_t(i)})$$

$$+ \frac{1}{m} \sum_{i=1}^{m} \nabla_{\boldsymbol{x}} f_i(\boldsymbol{x}^t, \boldsymbol{y}^t; \xi_{i,j_t(i)}) - \frac{1}{m} \sum_{i=1}^{m} \frac{1}{n} \sum_{l=1}^{n} \nabla_{\boldsymbol{x}} f_i(\boldsymbol{x}^t, \boldsymbol{y}^t; \xi_{i,l}),$$

$$\frac{1}{m} \sum_{i=1}^{m} \nabla_{\boldsymbol{x}} f_i(\boldsymbol{x}_i^{t'}, \boldsymbol{y}_i^{t'}; \xi_{i,j_{t'}(i)}) - \frac{1}{m} \sum_{i=1}^{m} \nabla_{\boldsymbol{x}} f_i(\boldsymbol{x}^{t'}, \boldsymbol{y}^{t'}; \xi_{i,j_{t'}(i)})$$

$$+ \frac{1}{m} \sum_{i=1}^{m} \nabla_{\boldsymbol{x}} f_i(\boldsymbol{x}^{t'}, \boldsymbol{y}^{t'}; \xi_{i,j_{t'}(i)}) - \frac{1}{m} \sum_{i=1}^{m} \frac{1}{n} \sum_{l=1}^{n} \nabla_{\boldsymbol{x}} f_i(\boldsymbol{x}^{t'}, \boldsymbol{y}^{t'}; \xi_{i,l}) \rangle$$

$$\overset{(a)}{=} 4G^2 T + \sum_{t \neq t'} \mathbb{E}_{\mathcal{A}} \langle \frac{1}{m} \sum_{i=1}^{m} \nabla_{\boldsymbol{x}} f_i(\boldsymbol{x}_i^t, \boldsymbol{y}_i^t; \xi_{i,j_t(i)}) - \frac{1}{m} \sum_{i=1}^{m} \nabla_{\boldsymbol{x}} f_i(\boldsymbol{x}^t, \boldsymbol{y}^t; \xi_{i,j_t(i)}),$$

$$\frac{1}{m} \sum_{i=1}^{m} \nabla_{\boldsymbol{x}} f_i(\boldsymbol{x}_i^{t'}, \boldsymbol{y}_i^{t'}; \xi_{i,j_{t'}(i)}) - \frac{1}{m} \sum_{i=1}^{m} \nabla_{\boldsymbol{x}} f_i(\boldsymbol{x}^{t'}, \boldsymbol{y}^{t'}; \xi_{i,j_{t'}(i)}) \rangle$$

$$\leq 4G^2 T + \sum_{t \neq t'} \mathbb{E}_{\mathcal{A}} \left( \left\| \frac{1}{m} \sum_{i=1}^{m} \left( \nabla_{\boldsymbol{x}} f_i(\boldsymbol{x}_i^t, \boldsymbol{y}_i^t; \xi_{i,j_t(i)}) - \nabla_{\boldsymbol{x}} f_i(\boldsymbol{x}^t, \boldsymbol{y}^t; \xi_{i,j_t(i)}) \right) \right\| \cdot \right.$$

$$\left. \left\| \frac{1}{m} \sum_{i=1}^{m} \left( \nabla_{\boldsymbol{x}} f_i(\boldsymbol{x}_i^{t'}, \boldsymbol{y}_i^{t'}; \xi_{i,j_{t'}(i)}) - \nabla_{\boldsymbol{x}} f_i(\boldsymbol{x}^{t'}, \boldsymbol{y}^{t'}; \xi_{i,j_{t'}(i)}) \right) \right\| \right)$$

$$\overset{(b)}{\leq} 4G^2 T + \sum_{t \neq t'} \frac{L^2}{m} \mathbb{E}_{\mathcal{A}} \left[ \sum_{i=1}^{m} (\|\boldsymbol{x}^t - \boldsymbol{x}_i^t\|^2 + \|\boldsymbol{y}^t - \boldsymbol{y}_i^t\|^2) \cdot \sum_{i=1}^{m} (\|\boldsymbol{x}^{t'} - \boldsymbol{x}_i^{t'}\|^2 + \|\boldsymbol{y}^{t'} - \boldsymbol{y}_i^{t'}\|^2) \right]^{1/2}$$

$$\leq 4G^2 T + \sum_{t \neq t'} 4G^2 L^2 \left( \sum_{s=0}^{t-1} \eta_s^{max} \lambda^{t-1-s} \right) \left( \sum_{s=0}^{t'-1} \eta_s^{max} \lambda^{t-1-s} \right)$$

where equality $(a)$ owes to that taking expectation on the randomness of $j_t$ or $j_{t'}$ equals to the empirical risk and inequality $(b)$ is due to the Lipschitz smoothness of each $f_i$ and Cauchy-Schwarz inequality.

Combining above inequalities of term $I$ and $II$ into the inequality (14) and we can summarize as:

$$\sum_{t=0}^{T-1} 2\eta_{\boldsymbol{x}} \mathbb{E}_{\mathcal{A}}[F_{\mathcal{S}}(\boldsymbol{x}^t, \boldsymbol{y}^t) - \inf_{\boldsymbol{x} \in \mathcal{X}} F_{\mathcal{S}}(\boldsymbol{x}, \boldsymbol{y}_{ave}^T)]$$

$$\leq (1 - \eta_{\boldsymbol{x}} \mu_{\boldsymbol{x}}) C_{\boldsymbol{x}}^2 + \eta_{\boldsymbol{x}}^2 G^2 T + 4\eta_{\boldsymbol{x}} C_{\boldsymbol{x}} G L \sum_{t=0}^{T-1} \sum_{s=0}^{t-1} \eta_s^{max} \lambda^{t-1-s}$$

$$+ 2C_{\boldsymbol{x}} \eta_{\boldsymbol{x}} \sqrt{4G^2 T + 4G^2 L^2 \sum_{t \neq t'} \left( \sum_{s=0}^{t-1} \eta_s^{max} \lambda^{t-1-s} \right) \left( \sum_{s=0}^{t'-1} \eta_s^{max} \lambda^{t'-1-s} \right)}$$

Dividing both sides by $T$ and we can get:

$$\frac{1}{T} \sum_{t=0}^{T-1} \mathbb{E}_{\mathcal{A}}[F_{\mathcal{S}}(\boldsymbol{x}^t, \boldsymbol{y}^t)] - \mathbb{E}_{\mathcal{A}}[\inf_{\boldsymbol{x} \in \mathcal{X}} F_{\mathcal{S}}(\boldsymbol{x}, \boldsymbol{y}_{ave}^T)]$$

$$\leq \frac{(1 - \eta_{\boldsymbol{x}} \mu_{\boldsymbol{x}}) C_{\boldsymbol{x}}^2}{2\eta_{\boldsymbol{x}} T} + \frac{\eta_{\boldsymbol{x}} G^2}{2} + \frac{4C_{\boldsymbol{x}} G L \eta^{\max}}{1 - \lambda} + \frac{2C_{\boldsymbol{x}} G}{\sqrt{T}}$$

And we can get the other-hand result in the same symmetric way:

$$\mathbb{E}_{\mathcal{A}}[\sup_{\boldsymbol{y} \in \mathcal{Y}} F_{\mathcal{S}}(\boldsymbol{x}_{ave}^T, \boldsymbol{y})] - \frac{1}{T} \sum_{t=0}^{T-1} \mathbb{E}_{\mathcal{A}}[F_{\mathcal{S}}(\boldsymbol{x}^t, \boldsymbol{y}^t)]$$

$$\leq \frac{(1 - \eta_{\boldsymbol{y}} \mu_{\boldsymbol{y}}) C_{\boldsymbol{y}}^2}{2\eta_{\boldsymbol{y}} T} + \frac{\eta_{\boldsymbol{y}} G^2}{2} + \frac{4C_{\boldsymbol{y}} G L \eta^{\max}}{1 - \lambda} + \frac{2C_{\boldsymbol{y}} G}{\sqrt{T}}$$

Combining above two inequalities we can get the result:

$$\Delta_{\mathcal{S}}^s(\boldsymbol{x}_{ave}^T, \boldsymbol{y}_{ave}^T) \leq \frac{C_{\boldsymbol{x}}^2 + C_{\boldsymbol{y}}^2}{2\eta^{min} T} + \eta^{max} G^2 + \frac{4(C_{\boldsymbol{x}} + C_{\boldsymbol{y}}) G L \eta^{\max}}{1 - \lambda} + \frac{2(C_{\boldsymbol{x}} + C_{\boldsymbol{y}}) G}{\sqrt{T}}$$

Then we come to the case b. when $\eta_{\boldsymbol{x},t} = \frac{1}{\mu_{\boldsymbol{x}} \cdot (t+1)^{c_{\boldsymbol{x}}}}$ and $\eta_{\boldsymbol{y},t} = \frac{1}{\mu_{\boldsymbol{y}} \cdot (t+1)^{c_{\boldsymbol{y}}}}$. Back to the inequality (13), we simplify it into:

$$\frac{2}{\mu_{\boldsymbol{x}}(t+1)^{c_{\boldsymbol{x}}}}(F_{\mathcal{S}}(\boldsymbol{x}^t, \boldsymbol{y}^t) - F_{\mathcal{S}}(\boldsymbol{x}, \boldsymbol{y}^t)) \leq (1 - \frac{1}{(t+1)^{c_{\boldsymbol{x}}}}) \|\boldsymbol{x} - \boldsymbol{x}^t\|^2 - \|\boldsymbol{x}^{t+1} - \boldsymbol{x}\|^2 + \frac{G^2}{\mu_{\boldsymbol{x}}^2 (t+1)^{2c_{\boldsymbol{x}}}}$$

$$+ \frac{2}{\mu_{\boldsymbol{x}}(t+1)^{c_{\boldsymbol{x}}}} \langle \boldsymbol{x} - \boldsymbol{x}^t, \frac{1}{m} \sum_{i=1}^m \nabla_{\boldsymbol{x}} f_i(\boldsymbol{x}_i^t, \boldsymbol{y}_i^t; \xi_{i,j_t(i)}) - \nabla_{\boldsymbol{x}} F_{\mathcal{S}}(\boldsymbol{x}^t, \boldsymbol{y}^t) \rangle$$

$$F_{\mathcal{S}}(\boldsymbol{x}^t, \boldsymbol{y}^t) - F_{\mathcal{S}}(\boldsymbol{x}, \boldsymbol{y}^t) \leq \frac{\mu_{\boldsymbol{x}}}{2}((t+1)^{c_{\boldsymbol{x}}} - 1)\|\boldsymbol{x} - \boldsymbol{x}^t\|^2 - \frac{\mu_{\boldsymbol{x}}}{2}(t+1)^{c_{\boldsymbol{x}}} \|\boldsymbol{x}^{t+1} - \boldsymbol{x}\|^2 + \frac{G^2}{2\mu_{\boldsymbol{x}}(t+1)^{c_{\boldsymbol{x}}}}$$

$$+ \langle \boldsymbol{x} - \boldsymbol{x}^t, \frac{1}{m} \sum_{i=1}^m \nabla_{\boldsymbol{x}} f_i(\boldsymbol{x}_i^t, \boldsymbol{y}_i^t; \xi_{i,j_t(i)}) - \nabla_{\boldsymbol{x}} F_{\mathcal{S}}(\boldsymbol{x}^t, \boldsymbol{y}^t) \rangle$$

Taking summation on the above inequality from $t = 0$ to $t = T - 1$ and making use of the convexity of $F_{\mathcal{S}}$ on the second parameter that:

$$\sum_{t=0}^{T-1} \left[ F_{\mathcal{S}}(\boldsymbol{x}^t, \boldsymbol{y}^t) - F_{\mathcal{S}}(\boldsymbol{x}, \boldsymbol{y}_{ave}^T) \right]$$

$$\leq \frac{G^2}{2\mu_{\boldsymbol{x}}} \sum_{t=0}^{T-1} \frac{1}{(t+1)^{c_{\boldsymbol{x}}}} + \sum_{t=0}^{T-1} \langle \boldsymbol{x} - \boldsymbol{x}^t, \frac{1}{m} \sum_{i=1}^m \nabla_{\boldsymbol{x}} f_i(\boldsymbol{x}_i^t, \boldsymbol{y}_i^t; \xi_{i,j_t(i)}) - \nabla_{\boldsymbol{x}} F_{\mathcal{S}}(\boldsymbol{x}^t, \boldsymbol{y}^t) \rangle$$

$$\leq \frac{G^2}{2\mu_{\boldsymbol{x}}} \sum_{t=0}^{T-1} \frac{1}{(t+1)^{c_{\boldsymbol{x}}}} + \sum_{t=0}^{T-1} \langle \boldsymbol{x}^t, \nabla_{\boldsymbol{x}} F_{\mathcal{S}}(\boldsymbol{x}^t, \boldsymbol{y}^t) - \frac{1}{m} \sum_{i=1}^m \nabla_{\boldsymbol{x}} f_i(\boldsymbol{x}_i^t, \boldsymbol{y}_i^t; \xi_{i,j_t(i)}) \rangle$$

$$+ \|\boldsymbol{x}\| \left\| \sum_{t=0}^{T-1} \left( \frac{1}{m} \sum_{i=1}^m \nabla_{\boldsymbol{x}} f_i(\boldsymbol{x}_i^t, \boldsymbol{y}_i^t; \xi_{i,j_t(i)}) - \nabla_{\boldsymbol{x}} F_{\mathcal{S}}(\boldsymbol{x}^t, \boldsymbol{y}^t) \right) \right\|$$

Next we will take expectation on the randomness of the algorithm on both sides and proceed in the same way as we do in case a. that:

$$\sum_{t=0}^{T-1} \mathbb{E}_{\mathcal{A}} \left[ F_{\mathcal{S}}(\boldsymbol{x}^t, \boldsymbol{y}^t) - \inf_{\boldsymbol{x} \in \mathcal{X}} F_{\mathcal{S}}(\boldsymbol{x}, \boldsymbol{y}_{ave}^T) \right]$$

$$\leq \frac{G^2}{2\mu_{\boldsymbol{x}}} \sum_{t=0}^{T-1} \frac{1}{(t+1)^{c_{\boldsymbol{x}}}} + 2C_{\boldsymbol{x}}GL \sum_{t=0}^{T-1} \sum_{s=0}^{t-1} \eta_s^{max} \lambda^{t-1-s}$$

$$+ C_{\boldsymbol{x}} \sqrt{4G^2 T + 4G^2 L^2 \sum_{t \neq t'} \left( \sum_{s=0}^{t-1} \eta_s^{max} \lambda^{t-1-s} \right) \left( \sum_{s=0}^{t'-1} \eta_s^{max} \lambda^{t-1-s} \right)}$$

Without loss of generalization, we assume $\eta_t^{max} = \eta_{\boldsymbol{x}, t} = \frac{1}{\mu_{\boldsymbol{x}}(t+1)^{c_{\boldsymbol{x}}}}$. Therefore the summation $\sum_{s=0}^{t-1} \eta_s^{max} \lambda^{t-1-s}$ comes out to be $\sum_{s=0}^{t-1} \frac{\lambda^{t-1-s}}{\mu_{\boldsymbol{x}}(t+1)^{c_{\boldsymbol{x}}}} \leq \frac{C_\lambda}{\mu_{\boldsymbol{x}} t^{c_{\boldsymbol{x}}}}$. Then we have to analyse in different categories for the value of $c_{\boldsymbol{x}}$.

First when $c_{\boldsymbol{x}} = 1$, the summation result turns out to be $\sum_{t=0}^{T-1} \frac{1}{t+1} \leq 1 + \ln T$,

$$\frac{1}{T} \sum_{t=0}^{T-1} \mathbb{E}_{\mathcal{A}} \left[ F_{\mathcal{S}}(\boldsymbol{x}^t, \boldsymbol{y}^t) \right] - \inf_{\boldsymbol{x} \in \mathcal{X}} F_{\mathcal{S}}(\boldsymbol{x}, \boldsymbol{y}_{ave}^T) \leq \frac{G^2}{2\mu_{\boldsymbol{x}}} \frac{1 + \ln T}{T} + \frac{4GLC_{\boldsymbol{x}}C_\lambda \ln T}{\mu_{\boldsymbol{x}} T} + \frac{2GC_{\boldsymbol{x}}}{\sqrt{T}}$$

When $0 < c_{\boldsymbol{x}} < 1$, $\sum_{t=0}^{T-1} \frac{1}{(t+1)^{c_{\boldsymbol{x}}}} \leq \frac{T^{1-c_{\boldsymbol{x}}}}{1-c_{\boldsymbol{x}}}$, then we have:

$$\frac{1}{T} \sum_{t=0}^{T-1} \mathbb{E}_{\mathcal{A}} \left[ F_{\mathcal{S}}(\boldsymbol{x}^t, \boldsymbol{y}^t) \right] - \inf_{\boldsymbol{x} \in \mathcal{X}} F_{\mathcal{S}}(\boldsymbol{x}, \boldsymbol{y}_{ave}^T) \leq \frac{G^2}{2\mu_{\boldsymbol{x}}(1-c_{\boldsymbol{x}})T^{c_{\boldsymbol{x}}}} + \frac{4GLC_{\boldsymbol{x}}C_\lambda}{\mu_{\boldsymbol{x}}(1-c_{\boldsymbol{x}})T^{c_{\boldsymbol{x}}}} + \frac{2GC_{\boldsymbol{x}}}{\sqrt{T}}$$

It is in a similar way to get the symmetric result on the other hand and combining both results we can obtain the strong primal-dual empirical risk:

$$\mathbb{E}_{\mathcal{A}}[\sup_{\boldsymbol{y} \in \mathcal{Y}} F_{\mathcal{S}}(\boldsymbol{x}_{ave}^T, \boldsymbol{y}) - \inf_{\boldsymbol{x} \in \mathcal{X}} F_{\mathcal{S}}(\boldsymbol{x}, \boldsymbol{y}_{ave}^T)] \leq \frac{2G(C_{\boldsymbol{x}} + C_{\boldsymbol{y}})}{\sqrt{T}} + T_{\boldsymbol{x}} + T_{\boldsymbol{y}} + T_{max}$$

where

$$T_{\boldsymbol{x}} = \begin{cases} \frac{G^2}{2\mu_{\boldsymbol{x}}} \frac{1 + \ln T}{T} & c_{\boldsymbol{x}} = 1 \\ \frac{G^2}{2\mu_{\boldsymbol{x}}(1-c_{\boldsymbol{x}})T^{c_{\boldsymbol{x}}}} & 0 < c_{\boldsymbol{x}} < 1 \end{cases} ; \quad T_{\boldsymbol{y}} = \begin{cases} \frac{G^2}{2\mu_{\boldsymbol{y}}} \frac{1 + \ln T}{T} & c_{\boldsymbol{y}} = 1 \\ \frac{G^2}{2\mu_{\boldsymbol{y}}(1-c_{\boldsymbol{y}})T^{c_{\boldsymbol{y}}}} & 0 < c_{\boldsymbol{y}} < 1 \end{cases}$$

and

$$c_{min} = \min\{c_{\boldsymbol{x}}, c_{\boldsymbol{y}}\}; \quad \mu = \min\{\mu_{\boldsymbol{x}}, \mu_{\boldsymbol{y}}\}; \quad T_{max} = \begin{cases} \frac{4GLC_\lambda(C_{\boldsymbol{x}} + C_{\boldsymbol{y}}) \ln T}{\mu T} & c_{min} = 1 \\ \frac{4GLC_\lambda(C_{\boldsymbol{x}} + C_{\boldsymbol{y}})}{\mu(1-c_{min})T^{c_{min}}} & 0 < c_{min} < 1 \end{cases}$$

□

### E.3 Proof of Strong/Weak Primal-Dual Population Risk

In this part, we are going to prove the strong and weak primal-dual population risk of the algorithm D-SGDA. Actually this is an obvious result as a summary of above lemmas and theorems.

*Proof of Theorem 3.* As we introduced the population risk (see Def. 2), we decompose the population risk into generalization gap and empirical risk that:

$$\Delta^s(\boldsymbol{x}_{ave}^T, \boldsymbol{y}_{ave}^T) = (\Delta^s(\boldsymbol{x}_{ave}^T, \boldsymbol{y}_{ave}^T) - \Delta_{\mathcal{S}}^s(\boldsymbol{x}_{ave}^T, \boldsymbol{y}_{ave}^T)) + \Delta_{\mathcal{S}}^s(\boldsymbol{x}_{ave}^T, \boldsymbol{y}_{ave}^T)$$

Notice that we use the average iterate $(\boldsymbol{x}_{ave}^T, \boldsymbol{y}_{ave}^T)$ instead of the last iterate to denote the output of D-SGDA. The first term is the averaged version of the strong primal-dual generalization gap and we will make some adjustments.

First we have the argument stability bound of D-SGDA that $\mathbb{E}_{\mathcal{A}} \left\| \begin{pmatrix} \boldsymbol{x}^T - \dot{\boldsymbol{x}}^T \\ \boldsymbol{y}^T - \dot{\boldsymbol{y}}^T \end{pmatrix} \right\| \leq \epsilon_{sta}^{arg}(\mathcal{A})$, where $(\boldsymbol{x}^T, \boldsymbol{y}^T)$ and $(\dot{\boldsymbol{x}}^T, \dot{\boldsymbol{y}}^T)$ denote the $T$-th output when D-SGDA is executed on the neighboring dataset

respectively. So we can make use of the convexity of the norm and obtain the argument stability of averaged-version:

$$\mathbb{E}_{\mathcal{A}}\left\|\begin{pmatrix} \boldsymbol{x}^T_{ave} - \dot{\boldsymbol{x}}^T_{ave} \\ \boldsymbol{y}^T_{ave} - \dot{\boldsymbol{y}}^T_{ave} \end{pmatrix}\right\| \le \mathbb{E}_{\mathcal{A}}\left\|\begin{pmatrix} \boldsymbol{x}^T - \dot{\boldsymbol{x}}^T \\ \boldsymbol{y}^T - \dot{\boldsymbol{y}}^T \end{pmatrix}\right\| \le \epsilon^{arg}_{sta}(\mathcal{A}) \tag{15}$$

Then the strong primal-dual generalization gap should be $G\sqrt{2 + \frac{2L^2}{\mu^2}}\epsilon^{arg}_{sta}(\mathcal{A})$ following Thm. 1 when each $f_i$ is strongly convex w.r.t. $\boldsymbol{x}$ and strongly concave w.r.t. $\boldsymbol{y}$.

At last, the strong primal-dual empirical risk $\mathbb{E}_{\mathcal{A}}[\Delta^s_{\mathcal{S}}(\boldsymbol{x}^T_{ave}, \boldsymbol{y}^T_{ave})]$ is studied in Thm. 7. So we can combine above two bounds to analyze the strong primal-dual population risk in different categories when learning rates are fixed:

$$\begin{aligned}
&\mathbb{E}_{\mathcal{A}}[\Delta^s(\boldsymbol{x}^T_{ave}, \boldsymbol{y}^T_{ave})] \\
&\le G\sqrt{2 + \frac{2L^2}{\mu^2}}\left(2G\frac{L+\mu}{\eta^{min}L\mu}(\frac{2(\eta^{max})^2 L}{1-\lambda} + \frac{\eta^{max}}{n})\right) + \frac{C^2_{\boldsymbol{x}} + C^2_{\boldsymbol{y}}}{2\eta^{min}T} + \eta^{max}G^2 \\
&\quad + \frac{4(C_{\boldsymbol{x}} + C_{\boldsymbol{y}})GL\eta^{\max}}{1-\lambda} + \frac{2(C_{\boldsymbol{x}} + C_{\boldsymbol{y}})G}{\sqrt{T}}
\end{aligned}$$

And when learning rates are varying that $\eta^{min}_t = \frac{1}{\mu(t+1)}$ and $\eta^{max}_t = \frac{1}{\mu(t+1)^c}, c = 1$, requiring $2c > \frac{L}{L+\mu} + 1$:

$$\begin{aligned}
&\mathbb{E}_{\mathcal{A}}[\Delta^s(\boldsymbol{x}^T_{ave}, \boldsymbol{y}^T_{ave})] \\
&\le G\sqrt{2 + \frac{2L^2}{\mu^2}}\left(\frac{2G}{\mu n T^{\frac{L}{L+\mu}}}\sum_{k=0}^{T-1}\frac{1}{(k+1)^{c-\frac{L}{L+\mu}}} + \frac{4GL}{\mu^2 T^{\frac{L}{L+\mu}}}\sum_{k=1}^{T-1}\frac{1}{(k+1)^{c-\frac{L}{L+\mu}}}\frac{C_\lambda}{k^c}\right) \\
&\quad + \frac{2G(C_{\boldsymbol{x}} + C_{\boldsymbol{y}})}{\sqrt{T}} + \frac{G^2}{\mu}\frac{1+\ln T}{T} + \frac{4GLC_\lambda(C_{\boldsymbol{x}} + C_{\boldsymbol{y}})\ln T}{\mu T} \\
&\le G\sqrt{2 + \frac{2L^2}{\mu^2}}\left(\frac{2G}{\mu^{\frac{L}{L+\mu}}}\frac{1}{n} + \frac{4GLC_\lambda}{\mu^2(1-\frac{L}{L+\mu})}\frac{1}{T^{\frac{L}{L+\mu}}}\right) + \frac{2G(C_{\boldsymbol{x}} + C_{\boldsymbol{y}})}{\sqrt{T}} + \frac{G^2}{\mu}\frac{1+\ln T}{T} \\
&\quad + \frac{4GLC_\lambda(C_{\boldsymbol{x}} + C_{\boldsymbol{y}})\ln T}{\mu T}
\end{aligned}$$

When $c < 1$ and requiring $2c > \frac{L}{L+\mu} + 1$:

$$\begin{aligned}
&\mathbb{E}_{\mathcal{A}}[\Delta^s(\boldsymbol{x}^T_{ave}, \boldsymbol{y}^T_{ave})] \\
&\le G\sqrt{2 + \frac{2L^2}{\mu^2}}\left(\frac{2G}{\mu n T^{\frac{L}{L+\mu}}}\sum_{k=0}^{T-1}\frac{1}{(k+1)^{c-\frac{L}{L+\mu}}} + \frac{4GL}{\mu^2 T^{\frac{L}{L+\mu}}}\sum_{k=1}^{T-1}\frac{1}{(k+1)^{c-\frac{L}{L+\mu}}}\frac{C_\lambda}{k^c}\right) \\
&\quad + \frac{2G(C_{\boldsymbol{x}} + C_{\boldsymbol{y}})}{\sqrt{T}} + \frac{G^2}{2\mu}(\frac{1+\ln T}{T} + \frac{1}{(1-c)T^c}) + \frac{4GLC_\lambda(C_{\boldsymbol{x}} + C_{\boldsymbol{y}})}{\mu(1-c)T^c} \\
&\le G\sqrt{2 + \frac{2L^2}{\mu^2}}\left(\frac{2G}{\mu(1-c+\frac{L}{L+\mu})}\frac{T^{1-c}}{n} + \frac{4GLC_\lambda}{\mu^2(2c-\frac{L}{L+\mu}-1)}\frac{1}{T^{\frac{L}{L+\mu}}}\right) + \frac{2G(C_{\boldsymbol{x}} + C_{\boldsymbol{y}})}{\sqrt{T}} \\
&\quad + \frac{G^2}{2\mu}(\frac{1+\ln T}{T} + \frac{1}{(1-c)T^c}) + \frac{4GLC_\lambda(C_{\boldsymbol{x}} + C_{\boldsymbol{y}})}{\mu(1-c)T^c}
\end{aligned}$$

When $c < 1$ and $2c = \frac{L}{L+\mu} + 1$:

$$\mathbb{E}_{\mathcal{A}}[\Delta^s(\boldsymbol{x}_{ave}^T, \boldsymbol{y}_{ave}^T)]$$

$$\leq G\sqrt{2 + \frac{2L^2}{\mu^2}\left(\frac{2G}{\mu n T^{\frac{L}{L+\mu}}}\sum_{k=0}^{T-1}\frac{1}{(k+1)^{c-\frac{L}{L+\mu}}} + \frac{4GL}{\mu^2 T^{\frac{L}{L+\mu}}}\sum_{k=1}^{T-1}\frac{1}{(k+1)^{c-\frac{L}{L+\mu}}}\frac{C_\lambda}{k^c}\right)}$$

$$+ \frac{2G(C_{\boldsymbol{x}} + C_{\boldsymbol{y}})}{\sqrt{T}} + \frac{G^2}{2\mu}(\frac{1+\ln T}{T} + \frac{1}{(1-c)T^c}) + \frac{4GLC_\lambda(C_{\boldsymbol{x}} + C_{\boldsymbol{y}})}{\mu(1-c)T^c}$$

$$\leq G\sqrt{2 + \frac{2L^2}{\mu^2}\left(\frac{2G}{c\mu}\frac{T^{1-c}}{n} + \frac{4GLC_\lambda}{\mu^2}\frac{\ln T}{T^{\frac{L}{L+\mu}}}\right)} + \frac{2G(C_{\boldsymbol{x}} + C_{\boldsymbol{y}})}{\sqrt{T}} + \frac{G^2}{2\mu}(\frac{1+\ln T}{T} + \frac{1}{(1-c)T^c})$$

$$+ \frac{4GLC_\lambda(C_{\boldsymbol{x}} + C_{\boldsymbol{y}})}{\mu(1-c)T^c}$$

$\square$

## F  Proof in the Convex-Concave Case

In this section, we will provide corresponding proof for argument stability, optimization error and weak primal-dual population risk in the C-C condition.

### F.1  Proof of Stability

*Proof of Theorem 4.* Analogous to Eq. (9) in the proof for SC-SC (see Appendix E.1), considering C-C a special case for $\mu_{\boldsymbol{x}}$SC-$\mu_{\boldsymbol{y}}$SC when $\mu_{\boldsymbol{x}} = 0, \mu_{\boldsymbol{y}} = 0$.

Thus we can get the result:

$$\mathbb{E}_{\mathcal{A}}\left\|\begin{pmatrix}\mathcal{A}_{\boldsymbol{x}}(\mathcal{S}) - \mathcal{A}_{\boldsymbol{x}}(\mathcal{S}') \\ \mathcal{A}_{\boldsymbol{y}}(\mathcal{S}) - \mathcal{A}_{\boldsymbol{y}}(\mathcal{S}')\end{pmatrix}\right\|$$

$$\leq \frac{2G}{n}\sum_{k=0}^{T-1}\eta_k^{max} + 4GL\sum_{k=1}^{T-1}\left(\eta_k^{max}\sum_{s=0}^{k-1}\eta_s^{max}\lambda^{k-1-s}\right)$$

$\square$

### F.2  Proof of the Empirical Risk

Similar to Thm. 7, considering C-C as a special case of SC-SC for $\mu_{\boldsymbol{x}} = \mu_{\boldsymbol{y}} = 0$, we can get the weak primal-dual empirical risk in the following corollary, where we use the Jensen's inequality that $\Delta_{\mathcal{S}}^w(\boldsymbol{x}, \boldsymbol{y}) \leq \Delta_{\mathcal{S}}^s(\boldsymbol{x}, \boldsymbol{y})$.

**Corollary 1.** *Under assumption 1,2,3 and the restriction that $\sup_{\boldsymbol{x}\in\mathcal{X}}\|\boldsymbol{x}\| \leq C_{\boldsymbol{x}}$ and $\sup_{\boldsymbol{y}\in\mathcal{Y}}\|\boldsymbol{y}\| \leq C_{\boldsymbol{y}}$, each local function is $f_i$ is C-C. We have the weak primal-dual empirical risk over the dataset $\mathcal{S}$ on the average output in $T$ iterations as following for fixed learning rates:*

$$\Delta_{\mathcal{S}}^w(\boldsymbol{x}_{ave}^T, \boldsymbol{y}_{ave}^T) \leq \frac{C_{\boldsymbol{x}}^2 + C_{\boldsymbol{y}}^2}{2\eta^{min}T} + \eta^{max}G^2 + \frac{4(C_{\boldsymbol{x}} + C_{\boldsymbol{y}})GL\eta^{max}}{1-\lambda} + \frac{2(C_{\boldsymbol{x}} + C_{\boldsymbol{y}})G}{\sqrt{T}}.$$

### F.3  Proof of Weak Primal-Dual Population Risk

*Proof of Theorem 5.* When each local function $f_i$ is not strongly convex or strongly concave, we can not get access to the strong primal-dual generalization gap but weak primal-dual generalization gap. Following the same step in above proof that:

$$\Delta^w(\boldsymbol{x}_{ave}^T, \boldsymbol{y}_{ave}^T) = (\Delta^w(\boldsymbol{x}_{ave}^T, \boldsymbol{y}_{ave}^T) - \Delta_{\mathcal{S}}^w(\boldsymbol{x}_{ave}^T, \boldsymbol{y}_{ave}^T)) + \Delta_{\mathcal{S}}^w(\boldsymbol{x}_{ave}^T, \boldsymbol{y}_{ave}^T)$$

Analogously we have the weak primal-dual generalization gap according to Thm. 1 that:

$$\Delta^w(\boldsymbol{x}_{ave}^T, \boldsymbol{y}_{ave}^T) - \Delta_{\mathcal{S}}^w(\boldsymbol{x}_{ave}^T, \boldsymbol{y}_{ave}^T) \leq \sqrt{2}G\epsilon_{sta}^{arg}(\mathcal{A})$$

where $\epsilon_{sta}^{arg}(\mathcal{A}) \le \frac{2G\eta^{max}T}{n} + \frac{4GL(\eta^{max})^2T}{1-\lambda}$ follows Thm. 4 when learning rates are fixed.

In the case without strong convexity or strong concavity, we select the fixed learning rates and following the Thm. 7, we can bound the weak primal-dual empirical risk as:

$$\Delta_{\mathcal{S}}^w(\boldsymbol{x}_{ave}^T, \boldsymbol{y}_{ave}^T) \le \frac{C_{\boldsymbol{x}}^2 + C_{\boldsymbol{y}}^2}{2\eta^{min}T} + \eta^{max}G^2 + \frac{4(C_{\boldsymbol{x}} + C_{\boldsymbol{y}})GL\eta^{max}}{1-\lambda} + \frac{2(C_{\boldsymbol{x}} + C_{\boldsymbol{y}})G}{\sqrt{T}}$$

Finally we combine above patterns and we can get the weak primal-dual population risk:

$$\Delta^w(\boldsymbol{x}_{ave}^T, \boldsymbol{y}_{ave}^T) \le \sqrt{2}G(\frac{2G\eta^{max}T}{n} + \frac{4GL(\eta^{max})^2T}{1-\lambda})$$
$$+ \frac{C_{\boldsymbol{x}}^2 + C_{\boldsymbol{y}}^2}{2\eta^{min}T} + \eta^{max}G^2 + \frac{4(C_{\boldsymbol{x}} + C_{\boldsymbol{y}})GL\eta^{max}}{1-\lambda} + \frac{2(C_{\boldsymbol{x}} + C_{\boldsymbol{y}})G}{\sqrt{T}}$$

$\square$

# G  Proof in Nonconvex-Nonconcave Case

In this section, we will provide proof for weak stability and therefore we can derive the weak primal-dual generalization gap following Thm. 1 in the NC-NC condition.

## G.1  Important Lemmas

Before we present the proof for the stability bound in nonconvex-nonconcave case, we should first introduce an important lemma which describes the fact that D-SGDA will run several iterations before encountering the different samples. We extend the Lemma 3.11 in [13] and make adjustments on Lemma F.1 in [19] to fit our decentralized setting.

**Lemma 5.** *Let $\mathcal{S} = \{\mathcal{S}_1, ..., \mathcal{S}_m\}$ and $\mathcal{S}' = \{\mathcal{S}_1', ..., \mathcal{S}_m'\}$ be any arbitrary neighboring datasets, $(\boldsymbol{x}^t, \boldsymbol{y}^t)$ and $(\dot{\boldsymbol{x}}^t, \dot{\boldsymbol{y}}^t)$ represent output in $t$-th iteration under dataset $\mathcal{S}$ and $\mathcal{S}'$ respectively. We further require each local function is bounded that $|f_i(\boldsymbol{x}, \boldsymbol{y}; \xi)| \le B, \forall \boldsymbol{x} \in \mathcal{X}, \boldsymbol{y} \in \mathcal{Y}$. Denoting $\delta_t = \left\| \begin{pmatrix} \boldsymbol{x}^t - \dot{\boldsymbol{x}}^t \\ \boldsymbol{y}^t - \dot{\boldsymbol{y}}^t \end{pmatrix} \right\|$, then we have:*

$$\mathbb{E}_{\mathcal{A}}[\boldsymbol{f}(\boldsymbol{x}^t, \boldsymbol{y}'; \boldsymbol{\xi}) - \boldsymbol{f}(\dot{\boldsymbol{x}}^t, \boldsymbol{y}'; \boldsymbol{\xi}) + \boldsymbol{f}(\boldsymbol{x}', \boldsymbol{y}^t; \boldsymbol{\xi}) - \boldsymbol{f}(\boldsymbol{x}', \dot{\boldsymbol{y}}^t; \boldsymbol{\xi})] \le \sqrt{2}G\mathbb{E}_{\mathcal{A}}\left[\delta_t \middle| \delta_{t_0} = 0\right] + \frac{Bmt_0}{n}.$$

*Proof.* First according to the property of Lipschitz continuity (see Assumption 1):

$$\boldsymbol{f}(\boldsymbol{x}^t, \boldsymbol{y}'; \boldsymbol{\xi}) - \boldsymbol{f}(\dot{\boldsymbol{x}}^t, \boldsymbol{y}'; \boldsymbol{\xi}) + \boldsymbol{f}(\boldsymbol{x}', \boldsymbol{y}^t; \boldsymbol{\xi}) - \boldsymbol{f}(\boldsymbol{x}', \dot{\boldsymbol{y}}^t; \boldsymbol{\xi})$$
$$\le G\|\boldsymbol{x}^t - \dot{\boldsymbol{x}}^t\| + G\|\boldsymbol{y}^t - \dot{\boldsymbol{y}}^t\|$$
$$\le G\sqrt{2}\delta_t$$

Then we decompose the expectation by the law of total expectation:

$$\mathbb{E}_{\mathcal{A}}[\boldsymbol{f}(\boldsymbol{x}^t, \boldsymbol{y}'; \boldsymbol{\xi}) - \boldsymbol{f}(\dot{\boldsymbol{x}}^t, \boldsymbol{y}'; \boldsymbol{\xi}) + \boldsymbol{f}(\boldsymbol{x}', \boldsymbol{y}^t; \boldsymbol{\xi}) - \boldsymbol{f}(\boldsymbol{x}', \dot{\boldsymbol{y}}^t; \boldsymbol{\xi})]$$

$$= \mathbb{P}(\delta_{t_0} = 0)\mathbb{E}_{\mathcal{A}}\left[\boldsymbol{f}(\boldsymbol{x}^t, \boldsymbol{y}'; \boldsymbol{\xi}) - \boldsymbol{f}(\dot{\boldsymbol{x}}^t, \boldsymbol{y}'; \boldsymbol{\xi}) + \boldsymbol{f}(\boldsymbol{x}', \boldsymbol{y}^t; \boldsymbol{\xi}) - \boldsymbol{f}(\boldsymbol{x}', \dot{\boldsymbol{y}}^t; \boldsymbol{\xi}) \middle| \delta_{t_0} = 0\right]$$

$$+ \mathbb{P}(\delta_{t_0} \ne 0)\mathbb{E}_{\mathcal{A}}\left[\boldsymbol{f}(\boldsymbol{x}^t, \boldsymbol{y}'; \boldsymbol{\xi}) - \boldsymbol{f}(\dot{\boldsymbol{x}}^t, \boldsymbol{y}'; \boldsymbol{\xi}) + \boldsymbol{f}(\boldsymbol{x}', \boldsymbol{y}^t; \boldsymbol{\xi}) - \boldsymbol{f}(\boldsymbol{x}', \dot{\boldsymbol{y}}^t; \boldsymbol{\xi}) \middle| \delta_{t_0} \ne 0\right]$$

$$\le \sqrt{2}G\mathbb{E}_{\mathcal{A}}\left[\delta_t \middle| \delta_{t_0} = 0\right] + B\mathbb{P}(\delta_{t_0} \ne 0)$$

While the event that $\delta_{t_0} \ne 0$ means the training process has already encountered the different samples before $t_0$:

$$\mathbb{P}(\delta_{t_0} \ne 0) \le \sum_{t=1}^{t_0}\sum_{k=1}^{m} C_m^k (\frac{1}{n})^k (1 - \frac{1}{n})^{m-k} = t_0(1 - (1 - \frac{1}{n})^m) \le \frac{mt_0}{n}$$

Combining above inequalities and we can prove the Lemma. $\square$

## G.2 Proof of Stability

*Proof of Theorem 6.* We are under the same setting as in the proof for SC-SC (see Appendix E.1) that $(\boldsymbol{x}^t, \boldsymbol{y}^t)$ and $(\dot{\boldsymbol{x}}^t, \dot{\boldsymbol{y}}^t)$ representing the $t$-th output over neighboring dataset $\mathcal{S}$ and $\mathcal{S}'$ respectively. And we assume each local dataset $\mathcal{S}'_i$ in $\mathcal{S}' = \{\mathcal{S}'_1, ..., \mathcal{S}'_m\}$ differs from $\mathcal{S}$ by the last sample without loss of generalization, i.e., $\mathcal{S}_i = \{\xi_{i,1}, ..., \xi_{i,n}\}$ while $\mathcal{S}'_i = \{\xi_{i,1}, ..., \xi_{i,n-1}, \xi'_{i,n}\}$.

Analogous to the decomposition equality (7), we will bound the term $I_1$ and $I_2$ without convexity or concavity. Referring to Lemma 1, we have:

$$\|I_1\| \le \frac{m - m_0}{m}(1 + \eta_t^{min}L)\left\|\begin{pmatrix} \boldsymbol{x}^t - \dot{\boldsymbol{x}}^t \\ \boldsymbol{y}^t - \dot{\boldsymbol{y}}^t \end{pmatrix}\right\|$$

$$\|I_2\| \le \frac{m_0}{m}\left[(1 + \eta_t^{min}L)\left\|\begin{pmatrix} \boldsymbol{x}^t - \dot{\boldsymbol{x}}^t \\ \boldsymbol{y}^t - \dot{\boldsymbol{y}}^t \end{pmatrix}\right\| + 2\eta_t^{max}G\right]$$

where we use the same denotation that $\eta_t^{min} = \min\{\eta_{\boldsymbol{x},t}, \eta_{\boldsymbol{y},t}\}$ and $\eta_t^{max} = \max\{\eta_{\boldsymbol{x},t}, \eta_{\boldsymbol{y},t}\}$. So we can get the similar result as the inequality (8):

$$\mathbb{E}_{\mathcal{A}}\left[\left\|\begin{pmatrix} \boldsymbol{x}^{t+1} - \dot{\boldsymbol{x}}^{t+1} \\ \boldsymbol{y}^{t+1} - \dot{\boldsymbol{y}}^{t+1} \end{pmatrix}\right\|\,\bigg|\,\delta_{t_0} = 0\right]$$

$$\le \sum_{m_0=0}^{m} C_m^{m_0}(1 - \frac{1}{n})^{m - m_0}(\frac{1}{n})^{m_0}\left((1 + \eta_t^{min}L)\,\mathbb{E}_{\mathcal{A}}\left[\left\|\begin{pmatrix} \boldsymbol{x}^t - \dot{\boldsymbol{x}}^t \\ \boldsymbol{y}^t - \dot{\boldsymbol{y}}^t \end{pmatrix}\right\|\,\bigg|\,\delta_{t_0} = 0\right]\right.$$

$$\left. + \frac{m_0}{m}2\eta_t^{max}G + 4\eta_t^{max}LG\sum_{s=0}^{t-1}\eta_s^{max}\lambda^{t-1-s}\right)$$

$$\le (1 + \eta_t^{min}L)\,\mathbb{E}_{\mathcal{A}}\left[\left\|\begin{pmatrix} \boldsymbol{x}^t - \dot{\boldsymbol{x}}^t \\ \boldsymbol{y}^t - \dot{\boldsymbol{y}}^t \end{pmatrix}\right\|\,\bigg|\,\delta_{t_0} = 0\right] + 4\eta_t^{max}LG\sum_{s=0}^{t-1}\eta_s^{max}\lambda^{t-1-s}$$

$$+ \sum_{m_0=0}^{m} C_m^{m_0}(1 - \frac{1}{n})^{m - m_0}(\frac{1}{n})^{m_0}\frac{m_0}{m}2\eta_t^{max}G$$

$$= (1 + \eta_t^{min}L)\,\mathbb{E}_{\mathcal{A}}\left[\left\|\begin{pmatrix} \boldsymbol{x}^t - \dot{\boldsymbol{x}}^t \\ \boldsymbol{y}^t - \dot{\boldsymbol{y}}^t \end{pmatrix}\right\|\,\bigg|\,\delta_{t_0} = 0\right] + 4\eta_t^{max}LG\sum_{s=0}^{t-1}\eta_s^{max}\lambda^{t-1-s} + \frac{2\eta_t^{max}G}{n}$$

Recursively applying above inequalities from $t = t_0$ to $t = T - 1$:

$$\mathbb{E}_{\mathcal{A}}\left[\left\|\begin{pmatrix} \mathcal{A}_{\boldsymbol{x}}(\mathcal{S}) - \mathcal{A}_{\boldsymbol{x}}(\mathcal{S}') \\ \mathcal{A}_{\boldsymbol{y}}(\mathcal{S}) - \mathcal{A}_{\boldsymbol{y}}(\mathcal{S}') \end{pmatrix}\right\|\,\bigg|\,\delta_{t_0} = 0\right]$$

$$\le \frac{2G}{n}\sum_{k=t_0}^{T-1}\eta_k^{max}\prod_{s=k+1}^{T-1}(1 + \eta_s^{min}L) + 4GL\sum_{k=t_0}^{T-1}\left(\eta_k^{max}\sum_{s=0}^{k-1}\eta_s^{max}\lambda^{k-1-s}\right)\prod_{j=k+1}^{T-1}(1 + \eta_j^{min}L)$$

(16)

where we use the fact that $\delta_{t_0} = \left\|\begin{pmatrix} \boldsymbol{x}^t - \dot{\boldsymbol{x}}^t \\ \boldsymbol{y}^t - \dot{\boldsymbol{y}}^t \end{pmatrix}\right\| = 0$.

When learning rates are fixed that $\eta_{\boldsymbol{x},t} = \eta_{\boldsymbol{x}}$ and $\eta_{\boldsymbol{y},t} = \eta_{\boldsymbol{y}}$, then we have:

$$\mathbb{E}_{\mathcal{A}}\left[\delta_t\,\bigg|\,\delta_{t_0} = 0\right]$$

$$\le \frac{2G\eta^{max}((1 + \eta^{min}L)^{T-t_0} - 1)}{n\eta^{min}L} + \frac{4GL(\eta^{max})^2((1 + \eta^{min}L)^{T-t_0} - 1)}{(1 - \lambda)\eta^{min}L}$$

Then combining with Lemma 5, we have the following result:

$$\mathbb{E}_{\mathcal{A}}[\boldsymbol{f}(\boldsymbol{x}^t, \boldsymbol{y}'; \boldsymbol{\xi}) - \boldsymbol{f}(\dot{\boldsymbol{x}}^t, \boldsymbol{y}'; \boldsymbol{\xi}) + \boldsymbol{f}(\boldsymbol{x}', \boldsymbol{y}^t; \boldsymbol{\xi}) - \boldsymbol{f}(\boldsymbol{x}', \dot{\boldsymbol{y}}^t; \boldsymbol{\xi})]$$

$$\le \sqrt{2}G(\frac{2G\eta^{max}}{n} + \frac{4GL\eta^{max2}}{1 - \lambda})(T - t_0) + \frac{Bmt_0}{n}$$

where it can obtain the optimal of $2\sqrt{2}G^2(\frac{\eta^{max}T}{n} + \frac{2L\eta^{max2}T}{1-\lambda})$ when $t_0 = 0$.

When learning rates are varying that $\eta_t^{min} = \frac{1}{t+1}$ and $\eta_t^{max} = \frac{1}{(t+1)^c}, c \leq 1$, then we can simplify the production by $1 \pm a \leq \exp\{a\}$:

$$\prod_{j=k+1}^{T-1} (1 + \eta_j^{min} L) = \prod_{j=k+1}^{T-1} (1 + \frac{L}{j+1}) \leq \prod_{j=k+1}^{T-1} \exp\left\{\frac{L}{j+1}\right\} = \exp\left\{\sum_{j=k+1}^{T-1} \frac{L}{j+1}\right\}$$

$$\leq \exp\left\{L \ln \frac{T}{k+1}\right\} = (\frac{T}{k+1})^L$$

Then back to the inequality (16), we can obtain:

$$\mathbb{E}_{\mathcal{A}}\left[\delta_t \,\Big|\, \delta_{t_0} = 0\right]$$

$$\leq \frac{2G}{n} \sum_{k=t_0}^{T-1} \frac{1}{(k+1)^c}(\frac{T}{k+1})^L + 4GL \sum_{k=t_0}^{T-1} \frac{1}{(k+1)^c}\frac{C_\lambda}{k^c}(\frac{T}{k+1})^L \tag{17}$$

$$= \frac{2GT^L}{n} \sum_{k=t_0}^{T-1} \frac{1}{(k+1)^{c+L}} + 4GLC_\lambda T^L \sum_{k=t_0}^{T-1} \frac{1}{(k+1)^{c+L}k^c}$$

Requiring $c+L > 1$ for convergence and combining with Lemma 5, we have the following inequality:

$$\mathbb{E}_{\mathcal{A}}[\boldsymbol{f}(\boldsymbol{x}^t, \boldsymbol{y}'; \boldsymbol{\xi}) - \boldsymbol{f}(\dot{\boldsymbol{x}}^t, \boldsymbol{y}'; \boldsymbol{\xi}) + \boldsymbol{f}(\boldsymbol{x}', \boldsymbol{y}^t; \boldsymbol{\xi}) - \boldsymbol{f}(\boldsymbol{x}', \dot{\boldsymbol{y}}^t; \boldsymbol{\xi})]$$

$$\leq \sqrt{2}G\left(\frac{2GT^L}{n} \sum_{k=t_0}^{T-1} \frac{1}{(k+1)^{c+L}} + 4GLC_\lambda T^L \sum_{k=t_0}^{T-1} \frac{1}{(k+1)^{c+L}k^c}\right) + \frac{Bmt_0}{n}$$

$$\leq \frac{2\sqrt{2}G^2 T^L}{(c+L-1)n} \frac{1}{t_0^{c+L-1}} + \frac{4\sqrt{2}G^2 LC_\lambda T^L}{(2c+L-1)} \frac{1}{t_0^{2c+L-1}} + \frac{Bmt_0}{n}$$

$$\leq (c+L)(c+L-1)^{\frac{1}{c+L}} \left(\frac{2\sqrt{2}G^2 T^L}{(c+L-1)n} + \frac{4\sqrt{2}G^2 LC_\lambda T^L}{2c+L-1}\right)^{\frac{1}{c+L}} (\frac{Bm}{n})^{1-\frac{1}{c+L}}$$

where for the last inequality, when $t_0 = \left(\frac{(c+L-1)(\frac{2\sqrt{2}G^2 T^L}{(c+L-1)n}+\frac{4\sqrt{2}G^2 LC_\lambda T^L}{2c+L-1})}{\frac{Bm}{n}}\right)^{\frac{1}{c+L}}$, it can obtain minimal.

Taking supremum over parameters $\boldsymbol{x}, \boldsymbol{y}$ separately and over the random sample, we can get the weak stability for D-SGDA in the NC-NC condition. $\qquad\square$

## H  Primal Metric

In addition to the primal-dual population risk (see Def. 2) as well as the corresponding generalization gap (see Def. 3), primal measure extends the relative concept in single variable minimization problem.

The excess primal population risk is defined as $\Delta^{ex}(\boldsymbol{x}) = \sup_{\boldsymbol{y}' \in \mathcal{Y}} F(\boldsymbol{x}, \boldsymbol{y}') - \inf_{\boldsymbol{x}' \in \mathcal{X}} \sup_{\boldsymbol{y}' \in \mathcal{Y}} F(\boldsymbol{x}', \boldsymbol{y}')$; and the excess primal empirical risk is $\Delta_{\mathcal{S}}^{ex}(\boldsymbol{x}) = \sup_{\boldsymbol{y}' \in \mathcal{Y}} F_{\mathcal{S}}(\boldsymbol{x}, \boldsymbol{y}') - \inf_{\boldsymbol{x}' \in \mathcal{X}} \sup_{\boldsymbol{y}' \in \mathcal{Y}} F_{\mathcal{S}}(\boldsymbol{x}', \boldsymbol{y}')$, for a randomized model $(\boldsymbol{x}, \boldsymbol{y})$.

There are two ways to define the corresponding generalization gap. One is to directly subtract the excess primal empirical risk from the excess primal population risk, defined as the excess primal generalization gap: $\epsilon_{gen}^{ex}(\boldsymbol{x}) = \Delta^{ex}(\boldsymbol{x}) - \Delta_{\mathcal{S}}^{ex}(\boldsymbol{x})$ (called primal gap in [30]). Another one is to neglect the difference between the saddle point of $F$ and $F_{\mathcal{S}}$, defined as the primal generalization gap: $\epsilon_{gen}^{pr}(\boldsymbol{x}) = \sup_{\boldsymbol{y}' \in \mathcal{Y}} F(\boldsymbol{x}, \boldsymbol{y}') - \sup_{\boldsymbol{y}' \in \mathcal{Y}} F_{\mathcal{S}}(\boldsymbol{x}, \boldsymbol{y}')$.

**Remark 11.** Our definition of strong primal-dual population risk (see Def. 2) has included the expectation inside for assistance with the definition of weak primal-dual population risk (see Def. 2). Although the definitions in [19] hold different forms, which do not contain the expectation inside, while our theoretical result aims at the same value.

Ozdaglar et al. [30] points out that the excess primal generalization gap $\epsilon_{gen}^{ex}$ can act as a better metric to characterize the generalizability in the nonconvex condition. It is our limitation that we do not

calculate the corresponding excess primal generalization gap and population risk for nonconvex case in our paper.

While we omit the primal generalization gap $\epsilon_{gen}^{pr}$ and excess primal population risk $\Delta^{ex}$ under C-C condition in the main text, for it can be derived from the corresponding proof of strong primal-dual risk. And we will illustrate them in the following as a corollary.

**Corollary 2.** *For an $\epsilon$-argument stable decentralized algorithm $\mathcal{A}$, under Assumption 1, 2, when each $f_i$ is $\mu_{\boldsymbol{y}}$-strongly concave on the second parameter, we have the primal generalization gap:*
$\mathbb{E}_{\mathcal{A},\mathcal{S}}[\epsilon_{gen}^{pr}(\mathcal{A}_{\boldsymbol{x}}(\mathcal{S}))] \leq G\sqrt{1 + \frac{L^2}{\mu_y^2}}\epsilon.$

*Proof of Corollary 2.* $\epsilon_{gen}^{pr}(\mathcal{A}_{\boldsymbol{x}}(\mathcal{S}))$ is exactly the first counterpart of the strong primal-dual generalization gap $\epsilon_{gen}^{s}(\mathcal{A}_{\boldsymbol{x}}(\mathcal{S}), \mathcal{A}_{\boldsymbol{y}}(\mathcal{S}))$ in Eq. (6). And referring to the proof for case b. in Thm. 1 (see Appendix D), we can get the result as above. $\qquad\square$

**Corollary 3.** *Under Assumption 1,2,3, when each $f_i$ is $\mu_{\boldsymbol{x}}SC$-$\mu_{\boldsymbol{y}}SC$, we have the excess primal population risk as follows, where $\eta_t^{max} \triangleq \max\{\eta_{\boldsymbol{x},t}, \eta_{\boldsymbol{y},t}\}$, $\eta_t^{min} \triangleq \min\{\eta_{\boldsymbol{x},t}, \eta_{\boldsymbol{y},t}\}$, $\mu = \min\{\mu_{\boldsymbol{x}}, \mu_{\boldsymbol{y}}\}$, and $(\boldsymbol{x}_{ave}^T, \boldsymbol{y}_{ave}^T)$ is defined in Eq. (3):*

*a. for fixed learning rates,*

$$\mathbb{E}[\Delta^{pr}(\boldsymbol{x}_{ave}^T)] \leq 2G\sqrt{1 + \frac{L^2}{\mu^2}}(2G\frac{L+\mu}{\eta^{min}L\mu}(\frac{2(\eta^{max})^2L}{1-\lambda} + \frac{\eta^{max}}{n})) + \frac{C_{\boldsymbol{x}}^2 + C_{\boldsymbol{y}}^2}{2\eta^{min}T}$$
$$+ \eta^{max}G^2 + \frac{4(C_{\boldsymbol{x}} + C_{\boldsymbol{y}})GL\eta^{max}}{1-\lambda} + \frac{2(C_{\boldsymbol{x}} + C_{\boldsymbol{y}})G}{\sqrt{T}}.$$

*b. for decaying learning rates that $\eta_t^{min} = \frac{1}{\mu(t+1)}$ and $\eta_t^{max} = \frac{1}{\mu(t+1)^c}$ with $c \leq 1$ and $2c \geq \frac{L}{L+\mu} + 1$,*

$$\mathbb{E}[\Delta^{pr}(\boldsymbol{x}_{ave}^T)]$$
$$\leq 2G\sqrt{1 + \frac{L^2}{\mu^2}}\left(\frac{2G}{\mu(1-c+\frac{L}{L+\mu})}\frac{T^{1-c}}{n} + \frac{4GLC_\lambda}{\mu^2 T^{\frac{L}{L+\mu}}}(\frac{\mathbf{1}_{2c\neq L/(L+\mu)+1}}{2c - \frac{L}{L+\mu} - 1} + \ln T \cdot \mathbf{1}_{2c=L/(L+\mu)+1})\right)$$
$$+ \frac{2G(C_{\boldsymbol{x}}+C_{\boldsymbol{y}})}{\sqrt{T}} + \frac{G^2}{2\mu}(\frac{1+\ln T}{T} + \frac{\mathbf{1}_{c\neq 1}}{(1-c)T^c} + \frac{(1+\ln T)\mathbf{1}_{c=1}}{T}) + \frac{4GLC_\lambda(C_{\boldsymbol{x}}+C_{\boldsymbol{y}})}{\mu T^c}(\frac{\mathbf{1}_{c\neq 1}}{1-c} + \ln T \cdot \mathbf{1}_{c=1}).$$

*Proof of Corollary 3.* Firstly we already know the argument stability bound for D-SGDA (denoted as $\mathcal{A}$) on the last iterate. Then the averaged output follows due to the convexity of the norm:

$$\mathbb{E}_{\mathcal{A}}\left\|\left(\begin{array}{c}\boldsymbol{x}_{ave}^T - \dot{\boldsymbol{x}}_{ave}^T \\ \boldsymbol{y}_{ave}^T - \dot{\boldsymbol{y}}_{ave}^T\end{array}\right)\right\| \leq \mathbb{E}_{\mathcal{A}}\left\|\left(\begin{array}{c}\boldsymbol{x}^T - \dot{\boldsymbol{x}}^T \\ \boldsymbol{y}^T - \dot{\boldsymbol{y}}^T\end{array}\right)\right\| \leq \epsilon_{sta}^{arg}(\mathcal{A})$$

Then we have the primal generalization gap according to Corollary 2 that $\mathbb{E}_{\mathcal{A},\mathcal{S}}[\epsilon_{gen}^{pr}(\boldsymbol{x}_{ave}^T)] \leq G\sqrt{1 + \frac{L^2}{\mu_y^2}}\epsilon_{sta}^{arg}(\mathcal{A})$.

We decompose the primal population risk for the averaged output $(\boldsymbol{x}_{ave}^T, \boldsymbol{y}_{ave}^T)$ as follows:

$$\sup_{\boldsymbol{y}'\in\mathcal{Y}} F(\boldsymbol{x}_{ave}^T, \boldsymbol{y}') - \inf_{\boldsymbol{x}'\in\mathcal{X}}\sup_{\boldsymbol{y}'\in\mathcal{Y}} F(\boldsymbol{x}', \boldsymbol{y}')$$
$$= \left(\sup_{\boldsymbol{y}'\in\mathcal{Y}} F(\boldsymbol{x}_{ave}^T, \boldsymbol{y}') - \sup_{\boldsymbol{y}'\in\mathcal{Y}} F_{\mathcal{S}}(\boldsymbol{x}_{ave}^T, \boldsymbol{y}')\right) + \left(\sup_{\boldsymbol{y}'\in\mathcal{Y}} F_{\mathcal{S}}(\boldsymbol{x}_{ave}^T, \boldsymbol{y}') - \inf_{\boldsymbol{x}'\in\mathcal{X}} F_{\mathcal{S}}(\boldsymbol{x}', \boldsymbol{y}_{ave}^T)\right)$$
$$+ \left(\inf_{\boldsymbol{x}'\in\mathcal{X}} F_{\mathcal{S}}(\boldsymbol{x}', \boldsymbol{y}_{ave}^T) - \inf_{\boldsymbol{x}'\in\mathcal{X}} F(\boldsymbol{x}', \boldsymbol{y}_{ave}^T)\right) + \left(\inf_{\boldsymbol{x}'\in\mathcal{X}} F(\boldsymbol{x}', \boldsymbol{y}_{ave}^T) - \inf_{\boldsymbol{x}'\in\mathcal{X}}\sup_{\boldsymbol{y}'\in\mathcal{Y}} F(\boldsymbol{x}', \boldsymbol{y}')\right)$$
$$\leq \left(\sup_{\boldsymbol{y}'\in\mathcal{Y}} F(\boldsymbol{x}_{ave}^T, \boldsymbol{y}') - \sup_{\boldsymbol{y}'\in\mathcal{Y}} F_{\mathcal{S}}(\boldsymbol{x}_{ave}^T, \boldsymbol{y}')\right) + \left(\sup_{\boldsymbol{y}'\in\mathcal{Y}} F_{\mathcal{S}}(\boldsymbol{x}_{ave}^T, \boldsymbol{y}') - \inf_{\boldsymbol{x}'\in\mathcal{X}} F_{\mathcal{S}}(\boldsymbol{x}', \boldsymbol{y}_{ave}^T)\right)$$
$$+ \left(\inf_{\boldsymbol{x}'\in\mathcal{X}} F_{\mathcal{S}}(\boldsymbol{x}', \boldsymbol{y}_{ave}^T) - \inf_{\boldsymbol{x}'\in\mathcal{X}} F(\boldsymbol{x}', \boldsymbol{y}_{ave}^T)\right)$$

where the inequality is due to: $\inf_{\boldsymbol{x}'\in\mathcal{X}} F(\boldsymbol{x}', \mathcal{A}_{\boldsymbol{y}}(\mathcal{S})) \leq \inf_{\boldsymbol{x}'\in\mathcal{X}}\sup_{\boldsymbol{y}'\in\mathcal{Y}} F(\boldsymbol{x}', \boldsymbol{y}')$.

The first term is the primal generalization gap. And notice that the third term is analogous to the contrast side of the primal generalization gap. So both of them can be bounded by $\epsilon_{gen}^{pr}(\boldsymbol{x}_{ave}^T)$.

While the second term is the strong primal-dual empirical risk referring to the proof of Thm. 7 (see Appendix E.2).

Overall we can get the excess primal population risk, almost the same with the strong primal-dual population risk for SC-SC case (see Appendix E.3) except for a $\sqrt{2}$-times factor in the argument stability error. And our bound analysis for the excess primal population risk is consistent with strong primal-dual population risk (see Remark 6 below Thm. 3).

$\square$

# I   Additional Experiments

In this paper, we include two experiments including solving the AUC problem on `svmguide` and `w5a` by the decentralized SGD to verify the conclusions for the Convex-Concave case and the generative adversarial network training for the Nonconvex-Nonconcave Case on `MNIST`.

## I.1   General setup.

Different from the stability and generalization analysis of the way in [19], we need to deal with learning in a decentralized manner. In our experiments, we denote the total number of clients as $N_c$ and $\mathcal{S} = \{S_1, S_2, \cdots, S_{N_c}\}$ as the set of samples, where $S_i$ represents the observations stored in the $i$-th client. And we let $N_{S_i}$ denote the size of $S_i$. We follow the same experimental setting as outlined in [13, 19] to build a neighboring/perturbing dataset $\mathcal{S}'$, which is constructed by individually changing one observation on each node. That is to say, for each $S_i$, $S_i'$ is constructed by randomly changing one element in $S_i$. Then, we deploy the totally same sub-model on each client and initialize them to the same starting point. Then, each sub-model is trained on its local data. After each iteration, each sub-model is communicated with some other clients as per a predefined communication topology. To evaluate the distance between two models trained on $\mathcal{S}$ and $\mathcal{S}'$, after finishing training, we obtain an ensemble model by averaging all sub-models collected from all clients.

## I.2   Detailed implementations.

For the AUC problem, we get two model squences $\{(w, v)\}$ and $\{(w', v')\}$. Then, we calculate the Euclidean distance $\Delta = (||w - w'||_2^2 + ||v - v'||_2^2)^{1/2}$. For training each model on each client, the algorithm we used in our experiment is `SOLAM` [41], which is the SGDA designed for the minimax AUC problem. We repeat the experiments 10 times to report the average results as well as the standard deviation.

For the generative adversarial learning problem, we just take the vanilla GAN structure, of which the generator and the discriminator comprise 4 fully connected layers, respectively. The leaky ReLU is taken before the output layer. Following [19], we ignore all forms of regularization such as the weight decay or dropout, and data augmentation tricks. We take 3 different seeds and 3 different ways to construct $\mathcal{S}'$, which means changing different observations (total 9 runs). To evaluate the model distance, we also take the Euclidean distance between the generator and discriminator separately.

Our implementation is highly based on the two source codes[2][3].

---

