# OpenReview forum: "Stability and Generalization of the Decentralized Stochastic Gradient Descent Ascent Algorithm"
_NeurIPS.cc/2023/Conference — NeurIPS 2023 poster_

### Official Review · Reviewer_Z6QM · 2023-07-05

**Soundness:** 3 good
**Presentation:** 2 fair
**Contribution:** 2 fair
**Rating:** 5
**Confidence:** 4

**Summary:**

This paper establishes the algorithmic-stability analysis of the decentralized stochastic gradient descent-ascent (D-SGDA) algorithm which is commonly used for min-max problems in distributed settings. This is turn leads to generalization bounds of D-SGDA for various assumptions on convexity-concavity of the objective. In particular, the paper focuses on two notions of generalization for min-max problems, namely weak and strong primal-dual generalization error. The bounds are expressed in terms of sample-size, training horizon and certain information about the mixing matrix such as the spectral gap. The flavor of results are closely-related to the recent analyses in literature that obtain algorithmic-stability bounds for decentralized gradient descent, however this work extend these to min-max scenarios.  The bounds (as expressed in the theorems) take complicated forms in term of the problem parameters, however the authors provide simplified interpretations of them in the remarks. In particular, the derived bounds strongly depend on the learning rate and spectral gap of mixing matrix in order to be meaningful. In particular, for strongly convex-strongly concave objectives, the generalization bounds are of order $O(\frac{\eta}{1-\lambda} + \frac{1}{n})$ and for the convex-concave case the bounds are $O(\frac{\eta T}{n} + \frac{\eta^2 T}{1-\lambda})$ for learning-rate $\eta$, sample size $n$, training time $T$ and the spectral gap $1-\lambda.$

**Strengths:**

The studied problem is new as it is the first work to obtain generalization bounds for D-SGDA. Although the generalization error of decentralized gradient descent and the gradient-descent algorithms were studied separately in existing works, this paper gives a unifying analysis. The method is largely based on the work of Hardt et al and relevant works on the gradient descent-ascent literature.

The derived bounds are in rather simple forms and might be informative on the role of sample-size or some other parameters such as the strong-convexity parameter on the generalization.

The paper is generally clear and easy to read although some aspects of the presentation can be improved as I will discuss in the next section.

Several experiments on different topologies presented in the paper seem to back the general claim on the role of spectral-gap and connectivity on generalization performance. I will discuss this more in the next section.



**Weaknesses:**

Weaknesses and some questions:

1 - One of my main concerns is that the generalization bounds derived in this paper are essentially vacuous for a large set of commonly used learning-rates or training time choices. Even for strongly convex-strongly concave setup, the generalization bounds are meaningful only when considering time-decaying learning rates. The authors do not compare their results to the state-of-the-art results for the generalization of ordinary gradient descent-ascent with fixed learning rate, therefore it is impossible to discuss the optimality of the obtained rates. This is especially important due to the role of step-size on the optimization and convergence of D-SGDA. The derived bounds also appear to be very sub-optimal for convex-concave objectives: the bounds are increasing in $T$ (because of the $\eta^2 T/(1-\lambda)$ term), therefore common early stopping choices such as $T\approx \sqrt{n}$ lead to a vacuous generalization bound unless the learning rate is decaying.

2 - The results in Table 1 are not very informative: there is no comparison with Decentralized SGD (D-SGD) and SGDA methods and the choice of step-size for each row is also not specified. It will be helpful if these comparisons are added.

3 - In line 47-49 the authors assert that *"Note that even for the decentralized minimization problem, the generalization and
 stability of decentralized SGD are adversely affected by an extra non-vanishing term [31], and the stability usually suffers from a constant term λ^2"*. While the current results on the stability of D-SGD indeed suffer from this additional term, these are just upper bounds therefore the conclusions on the role of topology/mixing matrix are not accurate until the optimality of these bounds are proved.

In the conclusions section the authors state *"Our theoretical results show that a decentralized
structure does not destroy the stability and generalization of D-SGDA"* . This is wrong as explained above and even in contradiction with the generalization rates obtained throughout the paper, as the generalization error rates is severely affected by decentralization.

4- A precise discussion on the optimization guarantees of DSGDA with the given learning rate selections seems lacking in the paper. This will be beneficial for understanding final test error rates.

5 - Some parts of the paper can be rephrased:

- The descriptions in the "Related Works" section are wrong in some places: The work of Bousquete and Elisseeff  established the connection between generalization and stability. Hardt et al. just showed that the output of SGD satisfies the stability notion. I think the lines 87-89 need to be edited.

- The sentence in line 80 "Huang [15]...." is also not clear and I think it needs to be rephrased.

- The authors refer to (Sun et al. 2021) and (Zhu et al. 2022) for previous works on generalization of decentralized methods. Some missing and relevant works that study generalization of DGD are the following:

  *"Graph-Dependent Implicit Regularisation for Distributed Stochastic Subgradient Descent"* by Richards et al. 2020.

  *"On Generalization of Decentralized Learning with Separable Data"* by Taheri et al. 2023.

  *"High-dimensional inference over networks: Linear convergence and statistical guarantees."* by Sun et al 2022.


- In the abstract (lines 11-13), the purpose of the sentence is not clear, since the bounds essentially are in terms of sample-size, learning rate and iterations.

- The sentence in line 35 is not clear. Can the authors please rephrase this sentence?

- Line 150: the sentence starting with ".And therefore" needs to be rephrased. Also see Line 235 and the sentence ".So...".


**Questions:**

Please see the last section for questions and suggestions.

**Limitations:**

The limitations of the work are clear and I see no potential negative societal impact related to this work.

---

> ### Author Rebuttal · Authors · 2023-08-09
>
> # Response to Reviewer Z6QM
>
> Thank you for your careful reading and constructive suggestions! We have answered your questions in the following comment.
>
> ***Q1: Vacuous results for generalization bounds.***
> 1.Our results of generalization are not vacuous even for fixed learning rate conditions. In SC-SC case, the stability and corresponding generalization gap bound is $\mathcal{O}(\frac{\eta}{1-\lambda}+\frac{1}{n})$ for fixed learning rates as we have discussed in Remark 5, where we can choose the learning rates as $\eta\sim\frac{1}{T}$ to get a non-vacuous bound. And in C-C condition, the generalization bound is $\mathcal{O}(\frac{\eta T}{n}+\frac{\eta^2 T}{1-\lambda})$ for fixed learning rates as shown in Remark 7, where we can also choose $\eta\sim\frac{1}{T}$ to obtain a non-vacuous bound.
> 2.Compared with vanilla SGDA, their generalization bound under SC-SC tis $\widetilde{\mathcal{O}}(\frac{1}{\sqrt{T}}+\frac{1}{N})$ (see Theorem 2.(e) in [18]). Our results can almost match it except for the influence of $C _{\lambda}$. Under C-C condition for fixed learning rates their generalization bound is $\mathcal{O}(\frac{\sqrt{T}}{n}+\frac{1}{\sqrt{n}})$ (see Theorem 2.(b) in [18]), our result can approach it when $\eta\sim\frac{1}{T^{\frac{3}{4}}}$ and $n\sim T^{\frac{3}{4}}$ except for the influence of $\frac{1}{1-\lambda}$. As for the influence of $\lambda$, i.e., the topology effect. We have made a thorough investigation in Remark 5.(3) with proof in Lemma 3 and Remark 10 in Appendix C for detail.
>
> ***Q2: The results in Table 1 are not very informative: there is no comparison between Decentralized SGD (D-SGD) and SGDA methods and the choice of step size for each row is also not specified. It will be helpful if these comparisons are added.***
> Thank you for your kind suggestions! In fact, our work focuses on the minimax problem, and we revise the Table and make comparisons with vanilla SGDA in generalization gap and population risks with respect to specified learning rates in the table presented in the PDF in the common response which you can refer to.
> However, compared with D-SGD which solves the minimization problem, our results can in a certain sense be tighter for the reason that our results do not contain a constant term concerned with $\lambda$. The $\lambda$-concerned term in all settings, SC-SC($\frac{C_{\lambda}}{T^{\frac{L}{L+\mu}}}$), C-C($\frac{1}{(1-\lambda)T}$), NC-NC($(C_{\lambda}T^L)^{\frac{1}{c+L}}(\frac{m}{n})^{1-\frac{1}{c+L}}$) can all vanish as iterations $T$ and sample amount $n$ increase.
>
>
> ***Q3: Questions about the statement.***
> Thank you for your constructive comments. We should revise our statement. Actually, we draw this conclusion due to the fact that our results containing $\lambda$-term can vanish as we have illustrated in the answer to ***Q2***. While in [31], the term concerned with $\lambda$ can not eliminate no matter how we choose the parameters.
>
>
> ***Q4: More discussions on the optimization guarantees.***
> Thank you for your kind suggestions. We have put the discussions about the optimization error in the appendix due to the space limit. Specifically, the optimization error for SC-SC is $\frac{C_{x}^2+C_{y}^2}{2\eta^{min}T}+\eta^{max}G^2+\frac{4(C_{x}+C_{y})GL\eta^{\max}}{1-\lambda}+\frac{2(C_{x}+C_{y})G}{\sqrt{T}}$ for fixed learning rates, and $\frac{2G(C_{x}+C_{y})}{\sqrt{T}}+T_{x}+T_{y}+T_{max}$ for decaying learning rates $\eta_{x,t}=\frac{1}{\mu_{x}(t+1)^{c_{x}}}$ and $\eta_{y,t}=\frac{1}{\mu_{y}(t+1)^{c_{y}}}$, where $T _ {\alpha}=\left\\{\begin{array}{cc}
>                 \frac{G^2}{2\mu_{\alpha}}\frac{1+\ln{T}}{T} & c_{\alpha}=1\\\\
>                 \frac{G^2}{2\mu_{\alpha}(1-c_{\alpha})T^{c_{\alpha}}} & 0< c_{\alpha}<1
>             \end{array}
>             \right.$ and $T _ {max}=\left\\{\begin{array}{cc}
>                 \frac{4GLC_{\lambda}(C_{x}+C_{y})\ln{T}}{\mu T} & k_{min}=1\\\\
>                 \frac{4GLC_{\lambda}(C_{x}+C_{y})}{\mu(1-k_{min})T^{k_{min}}}  &  0< k_{min}<1
>             \end{array}.
>             \right.$
> And the optimization error for C-C is $\frac{C_{x}^2+C_{y}^2}{2\eta^{min}T}+\eta^{max}G^2+\frac{4(C_{x}+C_{y})GL\eta^{\max}}{1-\lambda}+\frac{2(C_{x}+C_{y})G}{\sqrt{T}}$ for fixed learning rates.
> The details are presented in Theorem 7 in Appendix E.2 and Corollary 1 in Appendix F.2 respectively.
> Furthermore, considering the population risk is a sum of the generalization gap and optimization error. We have to re-choose the learning rates for lower population risk. For SC-SC case, we should choose $\eta\sim\frac{1}{\sqrt{T}}$ to also guarantee the optimization convergence, as we discussed in Remark 6. For C-C case, we have to re-choose $\eta\sim\frac{1}{T^{\frac{2}{3}}}$ to also guarantee the optimization convergence, as we discussed in Remark 8.
>
>
>
>
> ***Suggestions on the rephrased parts.***
> Thank you for your careful reading and constructive suggestions. We will revise our paper and polish our writing.
> - Line 87-89: Bousquete and Elisseeff[4] established the connection between generalization and stability. Elisseeff[8] extends the concept to randomized algorithms. Hardt[13] followed the theory and proved the stability of SGD.
> - Line 80: Huang[15], Luo and Ye[24] proposes accelerating algorithms by methods of variance reduction.
> - We will discuss these related works in proper positions in our paper.
> - Line 11-13: Yes indeed, what we mean is our results also analyze the impact of different topologies on the generalization bound except the trivial factors. - Line 35: However, it is important to consider the generalization performance for the stochastic algorithm, which is quantitatively evaluated by the value difference between Eq.(1) and Eq.(2).
> - Line 150: Hence, the difference between the empirical and population gaps can reflect the generalization performance.
> - Line 235: The major difference with vanilla SGDA lies in $C_{\lambda}$ and the number of nodes $m$.

---

> > ### Comment · Reviewer_Z6QM · 2023-08-18
> >
> > Thank you for your response. My score remains the same.

---

### Official Review · Reviewer_CPGp · 2023-07-06

**Soundness:** 2 fair
**Presentation:** 3 good
**Contribution:** 2 fair
**Rating:** 5
**Confidence:** 1

**Summary:**

this paper provides an analysis of the stability and generalization of decentralized GDA algorithms in SCSC, CC, and NCNC settings. the main results indicate that a decentralized setting GDA has similar error bounds as centralized settings. numerical experiments on AUC and GAN problems are reported.

**Strengths:**

1. the analysis looks solid to me and the results build the bridge between GDA and DGDA for generalization bound.
2. the numerical experiments are strong and convinsing.

**Weaknesses:**

1. the figures are too small, need to zoom in to read the content.

**Questions:**

1. what is the main technical difficulty for DGDA compared with the centralized case?

**Limitations:**

see weakness

---

> ### Author Rebuttal · Authors · 2023-08-10
>
> Thank you for your valuable comments! We have answered your questions in the following comment.
>
> ***Q1: Size of the figures.***
> Thank you for your valuable suggestions. We will zoom in on Figure 1 and Figure 2 for better readability.
>
> ***Q2: Technical difficulty of D-SGDA.***
> Thank you for your insightful question. We answer this question in the common response. Please refer to the part ***Technical challenges*** in the common response.

---

### Official Review · Reviewer_ipno · 2023-07-06

**Soundness:** 3 good
**Presentation:** 3 good
**Contribution:** 3 good
**Rating:** 6
**Confidence:** 4

**Summary:**

The paper investigates the primal-dual generalization gap bound of the decentralized stochastic gradient descent ascent (D-SGDA) algorithm. The authors start with a general decentralized minimax stochastic optimization problem, and its empirical counterpart calculated by the training dataset consisting of local samples. They model the network as an undirected graph with symmetric double stochastic matrix W. They assume Lipschitz-continuity and Lipschitz-smoothness on the local functions for the algorithm. They define weak and strong population risks, weak and strong generalization gaps -between the training dataset and original unknown distribution-, and algorithmic stability. The main purpose of the paper is the connection between and e-argument stable algorithm to its generalization gaps, both weak and strong. The authors show that the weak and strong -for strong convexity- strong concavity satisfied- generalization gaps are bounded by the argument stability error of an e-argument stable algorithm. The authors further show the boundedness of the argument and weak stability errors for strongly convex-strongly concave, convex-concave, and nonconvex-nonconcave settings. They analyze the generalization gaps for various parameters, including network topology, learning rates, and the number of nodes to show that the results are bounded as proposed.

**Strengths:**

The paper aims to investigate the connection between argument and weak stability with strong and weak generalization gaps for a training dataset for D-SGDA algorithm. The authors claim to propose this connection not only for D-SGDA algorithm, but for decentralized minimax algorithms in general. Their generalization and stability bounds for decentralized SGDA extent the stability and generalization for vanilla SGDA, so it can be said that the study is original. The overall quality of the paper is well, the problem is well-defined in a clear way and the approach is shown clearly. The results are significant at an acceptable level, since the study proposes the decentralized structure does not violate stability and generalization of SGDA algorithms, and decentralized minimax algorithms in general, as the bounds are shown by the authors.

**Weaknesses:**

In general, the paper is well-structured, but I believe a few things could be more elaborate. For example:

1. The mixing matrix and its purpose, how to select λ, what the W corresponds for a network topology could be explained better.

2. Proof of Theorem 1 in Appendix could show some intermediate steps where 𝑆(𝑙) is included.

3. Typos: Theorem 2, part b. wording, Ref. 6, sizes of parentheses in equations

**Questions:**

None

**Limitations:**

As the authors state, the analysis requires Lipschitz continuity and smoothness, and the stability and generalization analysis of D-SGDA for heterogeneous data distribution is missing. How would the analysis on stability and generalization vary for a directed network? In this case, W wouldn’t be necessarily symmetric, how would that affect the analysis?

---

> ### Author Rebuttal · Authors · 2023-08-09
>
> # Response to Reviewer ipno
>
>
> Thank you for your careful reading and constructive suggestions! We have answered your questions in the following comment.
>
> ***Q1: More explanations on the mixing matrix $W$ and the crucial constant $\lambda$.***
> Thank you for your kind suggestions. We will provide more details about the background and preliminary of the network and its associated mixing matrix in Section 3.2. Actually each communication network $G=(V,E)$ is naturally associated with an adjacency matrix, which is usually called a mixing matrix in the context of decentralized learning. Each element in the mixing matrix represents whether these two nodes communicate and the non-zero value will estimate the probability of one node choosing to communicate among its neighborhood. As for the crucial constant $\lambda$, it is naturally defined with a given matrix that is the second largest eigenvalue. So there exists a correspondence between specific network topology and the mixing matrix also and the constant $\lambda$.
> I can further illustrate the relationship by an example in the PDF in the common response which you can refer to.
>
>
> ***Q2: More intermediate steps in the proof of Theorem 1.***
> Actually $S^{(l)}$ is only a symbol for neighboring dataset in a decentralized manner (see Definition 4). In the proof of Theorem 1, we specify the detailed setting of $S^{(l)}$ as illustrated in Line 527-530 that $l=(l_1,l_2,...,l_m)$ with each $l_k$ representing the $l$-th local dataset differs by the $k$-th local sample. As for the formula begins at Line 536, the first equation is because there are $n^m$ kinds of permutation of $S^{(l)}$ and the symmetric distribution between $\xi_{i,l_i}$ and $\xi'_{i,l_i}$. And the subsequent steps do not involve the derivation of $S^{(l)}$.
>
>
> ***Typos: Theorem 2, part b. wording, Ref. 6, sizes of parentheses in equations.***
> Thank you for your careful reading. We will revise these typos in our paper and polish our writing later.
>
>
> ***Limitations: directed network.***
> Thank you for your insightful comment! Our theoretical result does rely on the property of the undirected graph, where the mixing matrix needs to be symmetric. We have also made experimental attempts on the *exponential* network and the experiment shows similar results, which offers us a clue for further theoretical research on directed networks. The research for directed graphs is nowadays a novel research avenue with notable works of *push-sum distributed algorithm* and we will attempt to address this issue in future work.

---

### Official Review · Reviewer_FcPE · 2023-07-25

**Soundness:** 3 good
**Presentation:** 4 excellent
**Contribution:** 3 good
**Rating:** 6
**Confidence:** 4

**Summary:**

The paper studied the generalization analysis of the decentralized stochastic gradient descent ascent algorithm through the lens of argument stability for solving minimax problems. Strong/weak primal-dual population risks are established for both convex-concave, strongly convex-strongly concave and nonconvex-nonconcave cases.


**Strengths:**

1. The paper provided a comprehensive stability and generalization analysis of D-SGDA for solving the minimax problem. Their results imply that the decentralized structure does not destroy the stability and generalization of SGDA.

2. The impact of different topologies of the decentralized structure on the generalization bound is observed, which is very interesting.

3. Experiments validate the theoretical results.


**Weaknesses:**

1. The results in Theorem 1 might be improved. Specifically, [1] established the connection between on-average argument stability and the weak primal-dual generalization gap for Markov chain SGDA (Here, SGDA is a special case of Markov chain SGDA) only with the Lipschitz assumption. Also, [2] provided this connection for Lipschitz losses. However, Theorem 1 requires the loss to satisfy both Lipschitz continuous and smooth conditions. For strong primal-dual generalization gap, [2] established the connection under the assumption $f$ is $\mu_y$ strongly-concave, while Theorem 1 assumes that $f$ is $\mu_x$ strongly-convex $\mu_y$ strongly-concave. In addition, $\mu$ is not defined in the theorem.


2. The results for both weak primal-dual and strong primal-dual population risks depend on $T$, $\eta$ and $n$. The authors might give some discussion on the choices of $\eta$ and $T$, and establish the explicit population rates as provided in [1] and [2], and compare with their results.

[1] Wang, P., Lei, Y., Ying, Y., and Zhou, D. X. (2022). Stability and generalization for markov chain stochastic gradient methods. Advances in Neural Information Processing Systems, 35, 37735-37748.

[2] Lei, Y., Yang, Z., Yang, T., and Ying, Y.(2021). Stability and generalization of stochastic gradient methods for minimax problems. In International Conference on Machine Learning, pages 6175–6186.

3. The experiment setup is unclear, I would suggest the authors add some details in the appendix.


**Questions:**

1. Can the stability and generalization results be generalized to the directed graph?

2. As mentioned in Weakness 2, could the authors provide some discussions or corollaries for discussing the choices of stepsize $\eta$ and iteration number $T$ to establish the explicit population rates?

3. A small question: in table 2, for fully connected graph, why $\frac{1}{1-\lambda}$=0?


**Limitations:**

yes

---

> ### Author Rebuttal · Authors · 2023-08-09
>
> # Response to Reviewer FcPE
>
> Thank you for your positive feedback, constructive suggestions, and insightful questions! We have answered your questions in the comment below.
>
> ***Q1: The results in Theorem 1 might be improved. Specifically, [1] established the connection between on-average argument stability and the weak primal-dual generalization gap for Markov chain SGDA (Here, SGDA is a special case of Markov chain SGDA) only with the Lipschitz assumption. Also, [2] provided this connection for Lipschitz losses. However, Theorem 1 requires the loss to satisfy both Lipschitz continuous and smooth conditions. For strong primal-dual generalization gap, [2] established the connection under the assumption $f$ is $\mu_y$ strongly-concave, while Theorem 1 assumes that $f$ is $\mu_x$strongly-convex$\mu_y$strongly-concave. In addition, $\mu$ is not defined in the theorem.***
> Thank you for your kind suggestions and we will add a detailed discussion on these two referred papers in our revision.
> 1.When establishing the connection between argument stability and weak primal-dual generalization gap, we are careless when checking the required conditions which strictly only need Lipschitz continuity. We will revise Theorem 1 for this part of our paper.
> 2.When establishing the connection between argument stability and strong primal-dual generalization gap, we check our proof process again and conclude that it does require $\mu_x$SC-$\mu_y$SC that strong convexity cannot be omitted. Because the deduction on the term $\inf_{x'\in\mathcal{X}}F(x',y)$ also relies on the property of strong convexity to derive $F(x^*(y),y)$ and Lipschitz property of $x^*(y)$ with respect to $y$. This deduction is just symmetric to the term $\sup_{y'\in\mathcal{Y}}F(x,y')$ which requires strong concavity as well. And we look into [2] and find that part c in Theorem 1 also requires strong convexity.
> Besides, we apologize for the negligence of definitions here $\mu\triangleq \min \lbrace\mu_x,\mu_y\rbrace$ which we will revise in our paper later.
>
>
> ***Q2: More discussion on the results of population risks and the choice of $\eta$, $T$, and $n$.***
> Thank you for your kind suggestions.
> For strong primal-dual population risk, for fixed learning rates, when we choose $\eta\sim\frac{1}{\sqrt{T}}$, our population rate is $\mathcal{O}(\frac{1}{n}+\frac{1}{(1-\lambda)\sqrt{T}})$. For decaying learning rates that $\eta^{min} _ t=\frac{1}{\mu(t+1)}$ and $\eta^{max} _ t=\frac{1}{\mu(t+1)^c}$ , the population rate is $\widetilde{\mathcal{O}}(\frac{T^{1-c}}{n}+\frac{C_ {\lambda}}{T^{\min\lbrace\frac{1}{2},\frac{L}{L+\mu}\rbrace}})$. Compared with vanilla SGDA[2], who reaches $\mathcal{O}(\frac{ln N}{\mu N})$ for decaying learning rates that $\eta_t=\frac{1}{\mu(t+t_0)}$ and $T\sim N$, our results can matach when $n\sim T^{\min\lbrace\frac{1}{2},\frac{L}{L+\mu}\rbrace}$.
> For weak primal-dual population risk, when $\eta\sim\frac{1}{T^{\frac{2}{3}}}$, the population rate is $\mathcal{O}(\frac{T^{\frac{1}{3}}}{n}+\frac{1}{(1-\lambda)T^{\frac{1}{3}}})$. Compared with vanilla SGDA[2], whose result is $\mathcal{O}(\frac{1}{\sqrt{n}})$ when $T\sim n$ and $\eta\sim\frac{1}{\sqrt{T}}$. And our results can match when $n^{\frac{1}{2}}\sim T^{\frac{1}{3}}$. But there is no need to compare with the results in [1] of $\mathcal{O}(\frac{log n}{\sqrt{n}log(\frac{1}{\lambda(P)})})$ when $T\sim n$ and $\eta\sim\frac{1}{\sqrt{T log(T)}}$ , since their results are under the Markovian assumptions.
> We have included the discussions in Remark 6 and Remark 8 respectively. And further proof details can be referred to Appendix E.3 and F.3 respectively. We will revise our paper to include more detailed comparisons with the referred [1][2].
>
>
> ***Q3: More details about the experiment setup.***
> We simplify the descriptions of the experiment set up in the paper due to the space limit, but we have implemented the details in Appendix I. We will make the descriptions more clear later.
>
>
>
>
>
> ***Q4: Can our results be generalized to the directed graph?***
> We have to admit that our current methods have restrictions that the mixing matrix should be symmetric, i.e., the network is limited to undirected graphs. However, we have tried some experiments on exponential topology which is a directed graph. The results do not exhibit contradictions with undirected graphs, which may be potential for future theoretical work on directed graphs. And we are thinking of the decentralized distributed method of push-sum which may help us analyze the directed network.
>
>
>
>
> ***Q5: Why $\frac{1}{1-\lambda}=0$ for fully connected graph?***
> We are sorry for the vague expressions in Table 2. The first row of $\lambda$ represents the value of the crucial constant, while the last two rows of $C_{\lambda}$ and $\frac{1}{1-\lambda}$ mean the value concerned about $\lambda$ appeared in our bound result respectively. For a fully connected graph, $\lambda=0$ so that $\lambda$ will not appear in the bound result. Therefore we remark it as $0$ which means will disappear in the term.

---

> > ### Comment · Reviewer_FcPE · 2023-08-16
> > **Thanks for the rebuttal**
> >
> > Thanks for the comments and I would like to keep my score.

---

### Official Review · Reviewer_5Agt · 2023-07-27

**Soundness:** 3 good
**Presentation:** 3 good
**Contribution:** 2 fair
**Rating:** 5
**Confidence:** 3

**Summary:**

This paper analyzes the generalization of the Decentralized Stochastic Gradient Descent Ascent (D-SGDA) Algorithm for min-max problem using the algorithm stability framework. Specifically, the paper analyzes the weak and strong primal-dual generalization gap under both convex-concave (C-C) and nonconvex-nonconcave (NC-NC) settings.

The key result is that the decentralized structure with undirected graph does not harm the generalization upper bound of D-SGDA.
Empirical experiments are performed to further validate the theoretical findings.

**Strengths:**

1. The paper is well written and easy to follow.

2. The paper studies an important problem of the generalization of the DSGDA algorithm for minmax problems in the decentralized settings.

**Weaknesses:**

1. Technical novelty is limited.

The algorithm stability for decentralized SGD algorithm has been analyzed in prior work [31]. And the algorithm stability for the SGDA algorithm for minmax problem has also been analyzed in prior work [18]. The techniques used in this paper mainly combine the two papers [18, 31].

2. The tightness of the generalization upper bound obtained in this paper is not discussed.

3. Some results are not clearly described. See **Questions**.


### Minor:

1. Unify the notations for projection operator. e.g. Algorithm 1 uses $P_{\cal X}$, while line 564-567 in the appendix uses $Proj_{\cal X}$.

2. Line 109, "differential" -> "differentiable".



**Questions:**

1. In Table 1, why only the NC-NC case has a bound depending on $m$, the number of agents while other results do not depend on $m$?

2. Please also include results on the optimization error and population risk in Table 1 or in the appendix.

3. One theoretical result shows that the decentralized structure does not harm stability and generalization of D-SGDA. This is different from the conclusion of [31], which shows the decentralized SGD are adversely affected by an extra non-vanishing term compared to SGD. What leads to the difference in the conclusions? Is it because of specific assumptions on the structure of the undirected graph, or improved / tighter analysis? This needs to be discussed carefully since it is the main contribution of this work.

4. In Theorem 6, what are the step size choices to achieve the corresponding bound? This needs to be stated as in other theorems.

**Limitations:**

The authors have discussed the limitations in Section 6, conclusion.

---

> ### Author Rebuttal · Authors · 2023-08-09
>
> # Response to Reviewer 5Agt
>
> Thank you for your careful reading and constructive suggestions! We have answered your questions in the comment below.
>
>
> ***Q1: Limited technical novelty.***
> We answer this question on the common response. Please refer to part ***Technical novelty*** in the common response.
>
>
> ***Q2: The tightness of upper bound.***
> Thank you for your constructive suggestions! We have not discussed the tightness of the generalization bound in our paper due to the fact that we are the first work to analyze the stability and generalization of decentralized minimax problems and we can not make comparisons. And we have to admit that we do not analyze the lower bound of our generalization gap, which is valuable for generalization analysis. We are thinking about this problem and will continue on this as future work inspired by [ref].
> However, in a certain sense, we have a tighter bound compared with the existing stability and generalization analysis on D-SGD, our upper bound does not contain a *constant* term concerned with $\lambda$. Specifically, as we have illustrated in the ***Technical novelty*** part in the common response, results in [31] contain an extra constant term concerned with $\lambda$ which means the upper bound suffers from a nonvanishing influence of the communication network. While in our work, the term concerned with $\lambda$ in the upper bound, in all cases of SC-SC ($\frac{C_{\lambda}}{T^{\frac{L}{L+\mu}}}$),C-C ($\frac{1}{(1-\lambda)T}$), and NC-NC ($(C_{\lambda}T^L)^{\frac{1}{c+L}}(\frac{m}{n})^{1-\frac{1}{c+L}}$), can vanish as iterations $T$ or the sample amount $n$ increase.
> We also implement comparisons with vanilla SGDA in the table presented in the PDF in the common response.
>
> [ref] *Stability of SGD: Tightness Analysis and Improved Bounds*
>
> ***Typos***
> > Minors:
> > 1. Unify the notations for projection operator. e.g. Algorithm 1 uses $P_{\cal X}$, while line 564-567 in the appendix uses $Proj_{\cal X}$.
> > 2. Line 109, "differential" -> "differentiable".
>
> Thank you for your careful reading and we will revise in the paper as suggested.
>
> ***Q3: Why only the NC-NC case has a bound depending on $m$?***
> Thank you for your insightful questions! In the NC-NC condition, we are using different deduction methods due to a lack of convexity. When meeting with the "different" sample for each local loss function, the cross term of the inner product between different local loss functions cannot be eliminated by the inequality property of convexity-concavity. Thus these cross terms can only be bounded by the Lipschitz property, and the number of cross terms will remain as a factor in the upper bound, which is related to the number of agents $m$.
>
> ***Q4: More results on the optimization error and population risk in Table 1 or in the appendix.***
> Thanks for your kind suggestions. We present our main results as a summary in Table 1, including the generalization gap and population risk under SC-SC and C-C cases. We omit the optimization error here since we are mainly concerned about the generalization performance of D-SGDA and the population risk acts as a sum of generalization gap and optimization error. For more details, we present the optimization error (empirical risk) in Theorem 7 in Appendix E.2 and Corollary 1 in Appendix F.2 for SC-SC and C-C respectively. As for NC-NC conditions, we only obtain results on its generalization gap without its optimization error or even population risk. It's a great challenge for minimax stochastic optimization problems under NC-NC conditions, where the Nash equilibrium saddle point may not exist, and finding it is NP-hard.
> Besides, we implement some comparisons with vanilla SGDA in the table presented on the PDF in the common response which you can refer to.
>
>
> ***Q5: More discussion on the result that the decentralized structure does not harm the stability and generalization of D-SGDA.***
> Yes indeed, we draw the conclusion that the decentralized structure does not harm the stability and generalization of D-SGDA from the observation that our algorithmic stability bound does not contain a non-vanishing term concerned with $\lambda$ (see Theorem 2, Remark 5 and Theorem 4, Remark 7). While in [31], as their Theorem 1 indicates that the second term in the algorithmic stability bound can not vanish with decaying learning rates and Theorem 2 shows that the second term does not disappear with iteration $T$ increasing.
> Actually, this difference does not come from extra assumptions. We can obtain a tighter bound due to the several preliminary steps in Line 563-567 in Appendix while [31] just directly comes to the norm of the differences, which result in a looser bound.
>
> At last, thank you for your kind suggestion to stress this contribution. We will revise our paper to stress our technical contribution.
>
>
> ***Q6: More statement on the choice of the step size in Theorem 6.***
> Thank you for your kind suggestions. We miss the discussion about the choice of step size in Theorem 6. Here we complement the following discussion which will later be revised into our paper in the remark below Theorem 6.
> For fixed learning rates, the weak stability can be bound: $\epsilon^w_{sta}(A)\leq2\sqrt{2}G^2(\frac{{\eta^{max}} T}{n}+\frac{2L{\eta^{max}}^2T}{1-\lambda})$ which can reach $\mathcal{O}(\frac{1}{n}+\frac{1}{(1-\lambda)T})$ when $\eta\sim\frac{1}{T}$. For deacaying learning rates that $\eta_t^{min}=\frac{1}{t+1}$, and $\eta_t^{max}=\frac{1}{(t+1)^c}, c\leq1$, the stability and generalization gap is bounded by $\small \mathcal{O}((C_{\lambda})^{\frac{1}{c+L}}T^{\frac{L}{c+L}}(\frac{m}{n})^{1-\frac{1}{c+L}})$.

---

> > ### Comment · Reviewer_5Agt · 2023-08-13
> > **Thanks for the rebuttal**
> >
> > Thanks for the detailed rebuttal. It addresses my concerns and I have updated my score.

---

> > > ### Author Response · Authors · 2023-08-14
> > > **Thanks for your support**
> > >
> > > Dear Reviewer 5Agt
> > >
> > > We sincerely thank you for raising your score. Your support means a lot to us. We really appreciate it!
> > > If you possess further insights or advice, your input is warmly welcomed.
> > >
> > > Best,
> > > Authors

---

### Author Rebuttal · Authors · 2023-08-09

# Common Response

***Technical challenges***
The major difference with vanilla SGDA lies in the communication through different local agents. The vanilla SGDA can be seen as a special case where there is only one agent and it can train on its own dataset and update its parameters all by itself. While in the decentralized setting, after each local agent updates by each local gradient sampled on the local dataset, they have to communicate with each other to complete the iterations.
So the key challenge is how we deal with the communication process. By means of the mixing matrix $W$ and its associated crucial constant $\lambda$, we can quantitatively characterize the communication between agents and observe from the bound results to figure out the impact of different communication ways (or associated network topology) on the generalization performance. In a decentralized setting, we need to measure the impact of consistency on generalization due to the fact that gradients are computed locally. Compared with the centralized approach, the decentralized setting lacks the aggregation step involving all workers, which can lead to a certain degree of model inconsistency.


***Technical novelty***
We are the first work to analyze the stability and generalization of D-SGDA for decentralized minimax problems, which is nontrivial considering the complex structure of minimax problems combined with the communication between agents.
Firstly, we propose the decentralized framework for analyzing stability and generalization based on the thought of permutation, which is not discussed in existing decentralized work. While under the new definitions of the decentralized neighboring dataset and corresponding decentralized algorithmic stability (see Definition 4 and 5, where we allow each local dataset to hold at most one different sample), we can see more clearly how the number of agents and local samples could influence the generalization performance. Furthermore, our proposed definitions are well-defined that when $m=1$, the decentralized definitions can be naturally degraded into vanilla one. When $m>1$, it can characterize the isolated influence of the number of the agent on the generalization performance.
Secondly, we have derived a tighter generalization bound compared with existing decentralized results. There are only two works concentrating on the generalization of decentralized algorithms. [43]requires a rather strong assumption that the weight difference obeys Gaussian distribution, and results in [31] contain an extra constant term concerned with $\lambda$ which means the upper bound suffers from a nonvanishing influence of the communication network. While in our work, the term concerned with $\lambda$ in the upper bound, in all cases of SC-SC ($\frac{C_{\lambda}}{T^{\frac{L}{L+\mu}}}$),C-C ($\frac{1}{(1-\lambda)T}$), and NC-NC ($(C_{\lambda}T^L)^{\frac{1}{c+L}}(\frac{m}{n})^{1-\frac{1}{c+L}}$), can vanish as iterations $T$ or the sample amount $n$ increase.


To summarize our ***contribution***, we are the first work to investigate the stability and generalization of D-SGDA for minimax problems. When meeting with the above challenges, we come up with the method of permutation and eventually draw the conclusion with a tightly bound to reveal the topology effect on the generalization performance.

---

### Decision · Program_Chairs · 2023-09-21

**Decision:**

Accept (poster)

**Comment:**

The paper presents generalization bounds for decentralized SGDA for min-max problems using the algorithmic stability approach. Reviewers appreciate the pioneering and comprehensive exploration of this important problem. The novel finding that the decentralized structure doesn't compromise SGDA's stability and generalization, along with its enhancement of existing work by relaxing assumptions on $\lambda$, is well-received. Although there were initially raised concerns about technical novelty, these were addressed in the rebuttal, leading to a revised score. Therefore, I recommend its acceptance.

I also strongly encourage the authors to thoroughly address all reviewers' comments in the revised version, particularly concerning the relationship with relevant cited papers.